# Apoptotic stress causes mtDNA release during senescence and drives the SASP

Stella Victorelli[1,2,21], Hanna Salmonowicz[1,2,3,4,21], James Chapman[3,21], Helene Martini[1,2], Maria Grazia Vizioli[1,2], Joel S. Riley[5,6,7], Catherine Cloix[5,6], Ella Hall-Younger[5,6], Jair Machado Espindola-Netto[2], Diana Jurk[1,2], Anthony B. Lagnado[1,2], Lilian Sales Gomez[1,2], Joshua N. Farr[1,2], Dominik Saul[2], Rebecca Reed[3], George Kelly[3], Madeline Eppard[1,2], Laura C. Greaves[8], Zhixun Dou[9,10], Nicholas Pirius[1,2], Karolina Szczepanowska[4], Rebecca A. Porritt[11], Huijie Huang[12], Timothy Y. Huang[12], Derek A. Mann[13,14], Claudio Akio Masuda[1,2,15], Sundeep Khosla[2], Haiming Dai[16], Scott H. Kaufmann[16], Emmanouil Zacharioudakis[17], Evripidis Gavathiotis[17], Nathan K. LeBrasseur[2], Xue Lei[18], Alva G. Sainz[19,20], Viktor I. Korolchuk[3], Peter D. Adams[18], Gerald S. Shadel[19], Stephen W. G. Tait[5,6]✉ & João F. Passos[1,2]✉

Senescent cells drive age-related tissue dysfunction partially through the induction of a chronic senescence-associated secretory phenotype (SASP)[1]. Mitochondria are major regulators of the SASP; however, the underlying mechanisms have not been elucidated[2]. Mitochondria are often essential for apoptosis, a cell fate distinct from cellular senescence. During apoptosis, widespread mitochondrial outer membrane permeabilization (MOMP) commits a cell to die[3]. Here we find that MOMP occurring in a subset of mitochondria is a feature of cellular senescence. This process, called minority MOMP (miMOMP), requires BAX and BAK macropores enabling the release of mitochondrial DNA (mtDNA) into the cytosol. Cytosolic mtDNA in turn activates the cGAS–STING pathway, a major regulator of the SASP. We find that inhibition of MOMP in vivo decreases inflammatory markers and improves healthspan in aged mice. Our results reveal that apoptosis and senescence are regulated by similar mitochondria-dependent mechanisms and that sublethal mitochondrial apoptotic stress is a major driver of the SASP. We provide proof-of-concept that inhibition of miMOMP-induced inflammation may be a therapeutic route to improve healthspan.

Cellular senescence refers to the irreversible growth arrest that occurs as a response to different stressors[1,4]. Senescent cells secrete multiple factors, collectively known as the SASP[5]. Senescent cells accumulate during ageing and chronic diseases and clearance of senescent cells alleviates several age-related pathologies in mice[6]. These cells therefore represent promising therapeutic targets to prevent age-related disorders.

Mitochondrial dysfunction is a hallmark of cellular senescence[2,7,8]. Our earlier data revealed that clearance of mitochondria in senescent cells suppresses the SASP while preserving the cell-cycle arrest[2], which led us to propose that mitochondria may be promising targets for anti-senescence therapies[9]. Mitochondria also have a major role in apoptosis, a process that involves MOMP, dependent on BAX or BAK, causing rapid cell death[3]. Apoptosis is also accompanied by BAX/BAK-dependent release of mtDNA into the cytosol[10,11].

Notably, we found that MOMP occurring in a small subset of mitochondria without inducing cell-death, an event called miMOMP[12], is a feature of cellular senescence. During senescence, miMOMP releases mtDNA into the cytosol, which activates the cGAS–STING pathway—a major regulator of the SASP[13,14]. Finally, we found that inhibition of miMOMP in vivo decreases inflammatory markers and improves multiple healthspan parameters in aged mice.

[1]Department of Physiology and Biomedical Engineering, Mayo Clinic, Rochester, MN, USA. [2]Robert and Arlene Kogod Center on Aging, Mayo Clinic, Rochester, MN, USA. [3]Biosciences Institute, Faculty of Medical Sciences, Campus for Ageing and Vitality, Newcastle University, Newcastle upon Tyne, UK. [4]ReMedy International Research Agenda Unit, IMol Polish Academy of Sciences, Warsaw, Poland. [5]Cancer Research UK Scotland Institute, Glasgow, UK. [6]School of Cancer Sciences, University of Glasgow, Glasgow, UK. [7]Institute of Developmental Immunology, Biocenter, Medical University of Innsbruck, Innsbruck, Austria. [8]Wellcome Centre for Mitochondrial Research, Biosciences Institute, Newcastle University, Newcastle upon Tyne, UK. [9]Center for Regenerative Medicine, Department of Medicine, Massachusetts General Hospital, Boston, MA, USA. [10]Harvard Stem Cell Institute, Harvard University, Cambridge, MA, USA. [11]Sanford Burnham Prebys Medical Discovery Institute, La Jolla, CA, USA. [12]Degenerative Diseases Program, Sanford Burnham Prebys Medical Discovery Institute, La Jolla, CA, USA. [13]Newcastle Fibrosis Research Group, Biosciences Institute, Newcastle University, Newcastle upon Tyne, UK. [14]Department of Gastroenterology and Hepatology, School of Medicine, Koç University, Istanbul, Turkey. [15]Instituto de Bioquímica Médica Leopoldo de Meis, Universidade Federal do Rio de Janeiro, Rio de Janeiro, Brazil. [16]Division of Oncology Research and Department of Molecular Pharmacology and Experimental Therapeutics, Mayo Clinic, Rochester, MN, USA. [17]Department of Biochemistry, Department of Medicine, Montefiore Einstein Cancer Center, Wilf Family Cardiovascular Research Institute, Institute for Aging Research, Albert Einstein College of Medicine, New York, NY, USA. [18]Cancer Genome and Epigenetics Program, Sanford Burnham Prebys Medical Discovery Institute, La Jolla, CA, USA. [19]Salk Institute for Biological Studies, La Jolla, CA, USA. [20]Department of Pathology, Yale University School of Medicine, New Haven, CT, USA. [21]These authors contributed equally: Stella Victorelli, Hanna Salmonowicz, James Chapman. ✉e-mail: Stephen.Tait@glasgow.ac.uk; passos.joao@mayo.edu

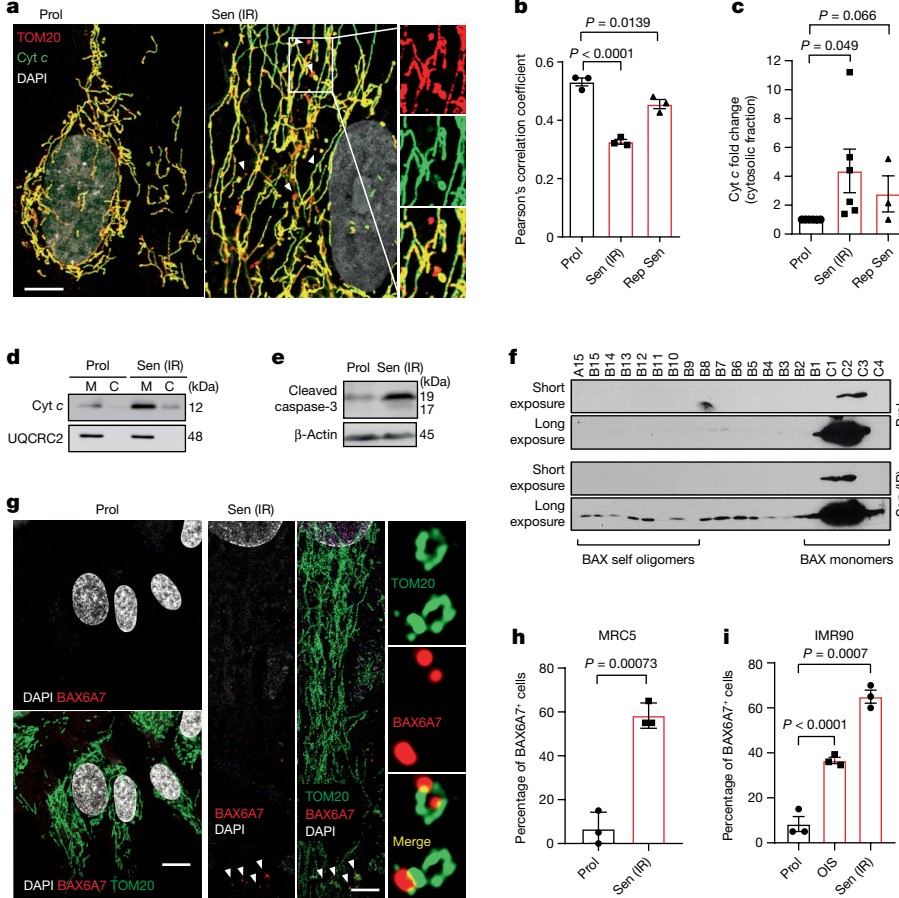

**Fig. 1 | Sublethal mitochondrial apoptotic signalling is a feature of cellular senescence. a**, Representative super-resolution SIM microscopy images of Cyt *c* (green) and TOM20 (red) in proliferating (prol.) and radiation-induced senescent (Sen IR) human fibroblasts. Scale bar, 10 μm. The magnified images on the right show areas in which Cyt *c* (green) does not co-localize with TOM20 (red) in senescent cells. **b**, Quantification of the co-localization between Cyt *c* and TOM20 using Pearson's correlation coefficient. *n* = 3 independent experiments. **c**, The relative levels of cytosolic Cyt *c* in irradiation-induced (IR) and replicative (Rep Sen) senescent cells expressed as the fold change compared with the proliferating control. *n* = 6 (Prol and Sen IR) and *n* = 3 (replicative senescent) independent experiments. **d**, Representative western blot of **c**, showing Cyt *c* enriched in the cytosolic fraction of senescent cells. UQCRC2 (complex III mitochondria protein) shows that the cytosolic fraction lacks mitochondria. **e**, Western blot analysis showing the increase in cleaved caspase-3 in irradiation-induced senescent cells. Data are representative of *n* = 4 independent experiments. **f**, Western blot analysis of BAX in FPLC fractions of proliferating and senescent cells. Fractions of decreasing protein molecular mass are shown from left to right. Films were exposed for 2 min (short exposure) and overnight (long exposure). Representative blot of *n* = 2 independent experiments. **g**, Representative super-resolution SIM microscopy images of BAX(6A7) (activated form of BAX; red) and TOM20 (green). Scale bars, 10 μm. **h,i**, Quantification of the percentage of MRC5 (**h**) and IMR90 (**i**) fibroblasts positive for BAX(6A7) co-localizing with TOM20. For **h** and **i**, *n* = 3 independent experiments. Data are mean ± s.e.m. Statistical significance was assessed using one-way analysis of variance (ANOVA) followed by Tukey's multiple-comparison test (**b**, **c** and **i**) and two-sided Student's unpaired *t*-tests (**c** and **h**). Gel source data for **d**–**f** are provided in Supplementary Fig. 1.

Our data support the concept that apoptosis and senescence are regulated by similar mitochondrial-dependent mechanisms and miMOMP is a major contributor to senescence and the SASP. Our results suggest that targeting miMOMP-induced inflammation may be a therapeutic route to improve healthspan.

## miMOMP occurs in cellular senescence

To investigate whether miMOMP is a feature of cell senescence, we conducted 3D structured illumination microscopy (SIM) analysis of proliferating or senescent human fibroblasts and analysed the co-localization between the outer mitochondrial membrane protein TOM20 and cytochrome *c* (Cyt *c*). While Cyt *c* and TOM20 co-localized in proliferating cells, in senescent cells, we observed a subset of peripheral mitochondria (dissociated from the network and exhibiting a globular structure) that were positive for TOM20, but not for Cyt *c* (Fig. 1a). Pearson's correlation coefficient analysis revealed decreased co-localization between Cyt *c* and TOM20 in radiation-induced senescent cells (Sen IR) and replicatively senescent cells (Fig. 1b). Supporting this, we found increased cytosolic Cyt *c* (Fig. 1c,d) and cleaved caspase-3 (Fig. 1e) in senescent cells. We next investigated whether miMOMP was accompanied by BAX oligomerization (indicative of BAX activation). BAX oligomerization was analysed using size-exclusion chromatography, which revealed that BAX forms oligomers in senescent, but not proliferating, cells (Fig. 1f). Using the antibody BAX6A7, which detects an active form of BAX[15], we found using 3D-SIM microscopy that, in senescent cells, peripheral globular mitochondria dissociated from the main network show increased expression of activated BAX (Fig. 1g). Increased BAX activation was observed in two independent human fibroblast strains (MRC5 and IMR90) and in irradiation-induced (Sen IR) and ER-RAS-oncogene-induced (OIS) senescence (Fig. 1h,i). These data demonstrate that miMOMP, a consequence of sublethal apoptotic stress, occurs during senescence.

## Cytosolic mtDNA increases in senescence

During apoptosis, MOMP leads to mtDNA release into the cytosol, but the inflammatory effects of this phenomenon are inhibited by caspase activity[10,11]. We hypothesized that mitochondria undergoing miMOMP may also release mtDNA into the cytosol of senescent cells. Dual immunostaining of TOM20 and DNA using 3D super-resolution AiryScan confocal microscopy revealed that senescent fibroblasts contain an increased number of DNA nucleoids in the cytosol. This observation was consistent between different cell types and irrespective of senescence-inducing stimulus (Fig. 2a,b). Similarly, we observed increased cytosolic nucleoids in senescent cells by immunogold labelling transmission electron microscopy (Fig. 2c). To determine whether the DNA nucleoids were of mitochondrial origin, we performed quantitative PCR (qPCR) to specifically detect mtDNA (D-loop region) and found increases in the cytosolic fractions of Sen IR, OIS and replicative senescent cells (Fig. 2d). Further supporting the mitochondrial origin of cytosolic DNA, we found that the majority of cytosolic DNA co-localized with transcription factor A (TFAM, mitochondrial), one of the core components of the mtDNA nucleoid (Fig. 2e,f). Cytosolic fractionation followed by western blotting confirmed an enrichment of TFAM in the cytosol of senescent cells (Fig. 2g). These results demonstrate that TFAM-bound mtDNA nucleoids, which are a preferred substrate for cGAS[16], are present in the cytosol of senescent cells.

## miMOMP is a driver of the SASP

We next investigated whether miMOMP is sufficient to drive senescence and the SASP. Proliferating human fibroblasts were treated with a known inducer of miMOMP, ABT-737, the prototypic BH3-mimetic compound that inhibits anti-apoptotic BCL-2 proteins. We used low concentrations that cause caspase activation but do not induce cell death (Extended Data Fig. 1a,b). Chronic treatment with ABT-737 resulted in a significant increase in the secretion of IL-6 and IL-8, as well as increased expression of *IL6*, *CXCL8* (encoding IL-8), *IL1A*, *IL1B*, *IFNA* and *IFNB* mRNAs in MRC5 and IMR90 human fibroblasts (Extended Data Fig. 1c–g). Notably, repeated ABT-737 treatment did not affect the rate of cell division, but cells reached replicative senescence a few population doublings earlier compared with the controls and showed a small, but significant, increase in the senescence-associated markers γH2A.X foci and SA-β-Gal (Extended Data Fig. 1h–j). This is consistent with previous findings that BH3-mimetic induced DNA damage can promote senescence[17]. Consistent with the reported role of MOMP in cytosolic mtDNA release, ABT-737 treatment resulted in an increased number of extramitochondrial nucleoids (Extended Data Fig. 1k).

The release of mtDNA can occur through BAX and BAK macropores[10,11]. Given our observation that senescent cells display BAX activation, we sought to address whether BAX and BAK (two proteins that are essential for MOMP) can facilitate mtDNA release in senescent cells. First, we observed that BAK (but not BAX) protein levels significantly increased in senescent cells when normalized to α-tubulin, but not when normalized to the mitochondrial protein VDAC. Thus, it is possible that the increase in BAK in senescent cells is due to changes in mitochondrial content (Supplementary Fig. 2). We used CRISPR–Cas9 gene editing to generate human fibroblasts that are deficient for both BAX and BAK (Fig. 3a). Combined deletion of *BAX* and *BAK* suppressed mtDNA release in senescent cells induced by DNA damage (Fig. 3b). To investigate whether BAX and BAK impacted the SASP, we performed RNA-sequencing (RNA-seq) analysis and a cytokine array using conditioned medium and found that commonly expressed SASP genes were decreased after *BAX* and *BAK* deletion (Fig. 3c,d). We confirmed the BAX/BAK-dependent decrease in different pro-inflammatory cytokines by independent enzyme-linked immunosorbent assays (ELISAs) and qPCR analysis of relevant targets (Extended Data Fig. 2a–f). We investigated whether deletion of *BAX* and *BAK* impacted senescence-related

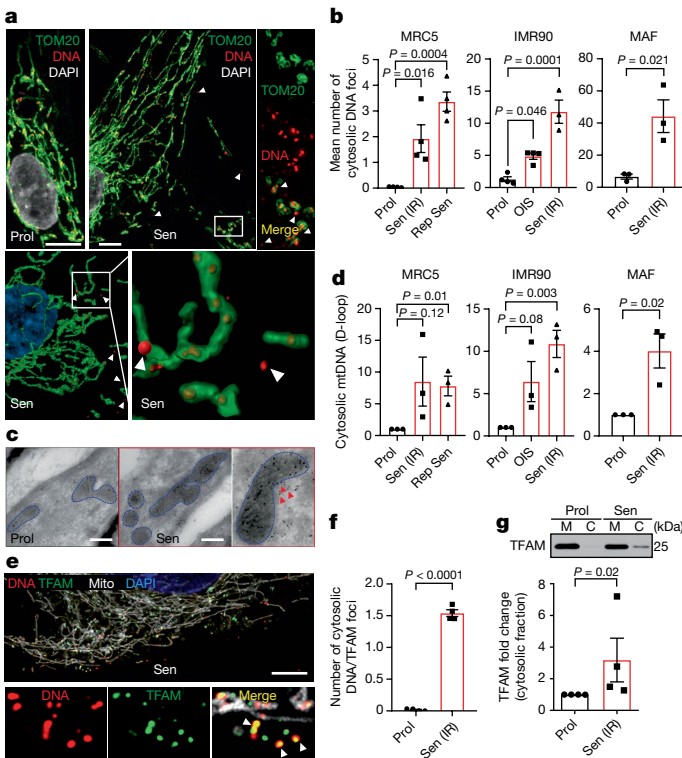

**Fig. 2 | Senescent cells have increased levels of cytosolic mtDNA.**
**a**, Representative super-resolution AiryScan microscopy images of DNA (red) and TOM20 (green) in proliferating and senescent cells (top). Scale bars, 5 μm. Bottom, 3D representations of the mitochondrial network (green) and DNA (red), showing that most DNA foci are located within mitochondria with some foci in the cytoplasm of senescent cells. **b**, The mean number of DNA foci present in the cytoplasm of proliferating and senescent human MRC5 and IMR90 fibroblasts and mouse adult fibroblasts (MAFs). *n* = 3 for IMR90 Sen (IR) and MAF, *n* = 4 independent experiments for all the other conditions. **c**, Representative electron microscopy images of DNA immuno-gold labelling in proliferating and senescent cells. Representative image of *n* = 2 independent experiments. Scale bars, 200 nm. **d**, qPCR quantification of the levels of mtDNA (D-loop region) present in the cytosolic fraction of proliferating and senescent human MRC5 and IMR90 fibroblasts and MAFs, normalized to the levels of total cellular mtDNA. *n* = 3 independent experiments. **e**, Representative super-resolution AiryScan microscopy image of DNA (red), TFAM (green) and mitochondria (Mito, white; labelled with BacMam 2.0 RFP) in senescent cells. Scale bar, 10 μm. The magnified images show TFAM co-localizing with DNA outside the mitochondrial network, representing mtDNA leakage. **f**, Quantification of the mean number of DNA + TFAM foci present outside of the mitochondrial network in proliferating and senescent cells. *n* = 4 independent experiments. **g**, Representative western blot (top) and quantification (bottom) of TFAM present in the cytosolic fraction of proliferating and senescent cells. *n* = 4 independent experiments. Values are normalized to the mitochondrial protein UQCRC2 (shown in Fig. 1d; the samples in **g** and Fig. 1d were probed for Cyt *c* and TFAM, respectively, on the same blot) and expressed as the fold change. Gel source data for **g** are provided in Supplementary Fig. 1. Data are mean ± s.e.m. Statistical significance was assessed using one-way ANOVA followed by Tukey's multiple-comparison test (**b**) and two-sided Student's unpaired *t*-tests (**b**, **d**, **f** and **g**). Individual data points are from biological replicates.

cell-cycle arrest pathways. Deletion of *BAX* and *BAK* did not alter the expression of the cyclin-dependent-kinase inhibitors p21 and p16^INK4a (Fig. 3e–g,m) or increase SA-β-Gal activity (Fig. 3h,m) or the number of γH2A.X foci (Fig. 3i,m), or impact the senescence-associated loss of lamin B1 and HMGB1 (Fig. 3j). No changes in the expression of the proliferation marker Ki-67 (Fig. 3k,m) or proliferation genes that are normally downregulated in senescent cells were observed (Fig. 3l).

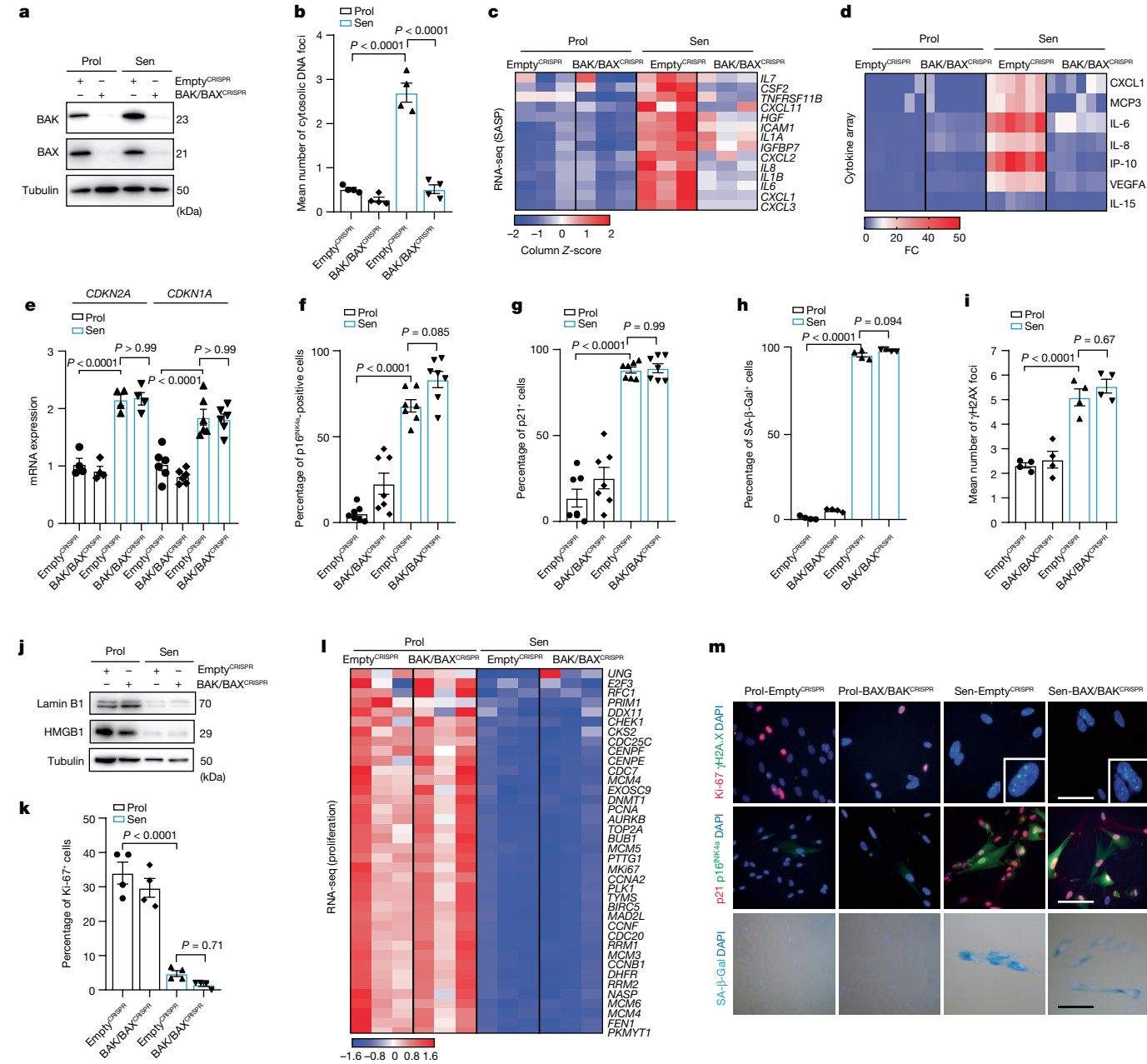

**Fig. 3 | BAX and BAK macropores mediate mtDNA release and the SASP in senescent cells. a**, CRISPR–Cas9 gene editing was used to generate human fibroblasts deficient in *BAX* and *BAK* (*BAX*⁻/⁻ *BAK*⁻/⁻). Western blot showing successful CRISPR–Cas9-mediated deletion of *BAK* and *BAX* in proliferating and senescent (IR) cells. BAX/BAK^CRISPR indicates cells negative for *BAX* and *BAK*. Western blot is representative of *n* = 3 independent experiments. **b**, The mean number of cytosolic DNA foci in proliferating and senescent *BAX*⁻/⁻ *BAK*⁻/⁻ cells. *n* = 4 independent experiments. **c**, Column-clustered heat map of SASP genes that are differentially expressed in senescence and rescued by deletion of *BAX* and *BAK*. The colour intensity represents the column *Z*-score; red and blue indicate high and low expression, respectively. **d**, The levels of secreted cytokines in proliferating and senescent empty vector (EV) and *BAX*⁻/⁻ *BAK*⁻/⁻ cells. *n* = 6 independent experiments. **e**, mRNA levels of *CDKN2A* (*n* = 4 independent experiments) and *CDKN1A* (*n* = 6 independent experiments) in proliferating and senescent EV and *BAX*⁻/⁻ *BAK*⁻/⁻ cells. **f,g**, The percentage p16^INK4a-positive (**f**) and p21-positive (**g**) proliferating and senescent EV and *BAX*⁻/⁻ *BAK*⁻/⁻ cells. *n* = 7 independent experiments. **h,i**, Quantification of the percentage of senescence-associated β-galactosidase (Sen-β-Gal)-positive cells (**h**) and the mean number of γH2AX foci in proliferating and senescent EV and *BAX*⁻/⁻ *BAK*⁻/⁻ cells (**i**). *n* = 4 independent experiments. **j**, Western blot analysis of lamin B1 and HMGB1 levels in proliferating and senescent EV and *BAX*⁻/⁻ *BAK*⁻/⁻ cells. Representation of *n* = 3 independent experiments. **k**, The percentage of Ki-67-positive proliferating and senescent EV and *BAX*⁻/⁻ *BAK*⁻/⁻ cells. *n* = 4 independent experiments. **l**, Column-clustered heat map of proliferation genes that are differentially expressed in senescent cells and are not rescued by deletion of *BAX* and *BAK*. The colour intensity represents the column *Z*-score. **m**, Representative microscopy images of Ki-67 (red) and γH2AX (green) (top); p21 (red) and p16^INK4a (green) (blue is DAPI) (middle); and Sen-β-Gal (bottom) in proliferating and senescent EV and *BAX*⁻/⁻ *BAK*⁻/⁻ cells. Scale bars, 100 μm. Representative images of *n* = 4 independent experiments. Data are mean ± s.e.m. Statistical significance was assessed using one-way ANOVA followed by Tukey's multiple-comparison test (**b**, **e**–**i** and **k**). Individual data points are from biological replicates. Gel source data for **j** are provided in Supplementary Fig. 1.

These data suggest that BAX and BAK regulate the SASP, but not senescence-associated cell-cycle arrest.

The mitochondrial permeability transition pore (MPTP) can also be involved in the release of mtDNA[18]. To investigate the role of the MPTP, we treated senescent cells with cyclosporin A, which inhibits the MPTP by binding to mitochondrial cyclophilin D, and then measured mtDNA release and secretion of the SASP factors. Cyclosporin A treatment did not alter the percentage of cells containing cytoplasmic mtDNA and did not suppress the secretion of SASP components IL-6 and IL-8 (Extended Data Fig. 2g,h).

BAX and BAK can trigger other mechanisms that may contribute to the SASP, besides enabling mtDNA release. For example, dysfunctional mitochondria in senescent cells lead to the formation of cytosolic chromatin fragments (CCFs) and the SASP[19]. However, loss of BAX and BAK did not affect mitochondrial respiration, generation of reactive oxygen species or the formation of CCF in senescent cells (Extended Data Fig. 2i–k). Moreover, there were no significant differences in mRNA expression of OXPHOS components after *BAX* and *BAK* deletion (Extended Data Fig. 2l). Finally, we investigated whether *BAX* and *BAK* deletion impacted the SASP in therapy-induced senescence. Combined *BAX* and *BAK* deletion reduced the SASP, but not expression of the cyclin-dependent-kinase inhibitors p21 and p16[INK4a] when senescence was induced by chemotherapeutic agents doxorubicin and etoposide (Extended Data Fig. 3a–l).

We addressed the possibility that SASP may be influenced by BAX- and BAK-dependent caspase activation, which, during apoptosis, dampens mtDNA-driven inflammation[20–22]. As APAF1 is required for mitochondria-dependent caspase activation (Extended Data Fig. 4a), we used CRISPR–Cas9 to delete *APAF1* in human fibroblasts (Extended Data Fig. 4b). APAF1-deficient cells reach replicative senescence at the same population doubling as control cells (Extended Data Fig. 4c). Moreover, *APAF1* deletion did not affect the expression of cyclin-dependent-kinase inhibitors, the proliferation marker Ki-67 or SASP components when senescence was induced by irradiation (Extended Data Fig. 4d–k). These results indicate that miMOMP-dependent caspase activation does not affect SASP activation under these conditions.

## BAX and BAK promote the SASP in vivo

To investigate a role of BAX and BAK in senescence in vivo, we used *Bax*[fl/fl]*Bak*[−/−] mice and performed tail-vein injection with AAV-TBG-Cre virus to delete *Bax* in the liver. We exposed mice to 4 Gy of ionizing irradiation (IR), a dose that was previously shown to induce senescence and the SASP in the liver[13,19] (Fig. 4a). The absence of BAX in the liver was confirmed using immunohistochemistry (Fig. 4a). We next characterized the mRNA expression of different pro-inflammatory components 6 days after IR and found that, while these were significantly increased in *Bak*[−/−] mice, deletion of both *Bax* and *Bak* suppressed their induction (Fig. 4b). Nonetheless, when we analysed the senescence marker telomere-associated foci, which denotes co-localization between the DNA damage response protein γH2A.X and telomeres, we found that it was induced by irradiation irrespectively of the genotype (Supplementary Fig. 3a,b). IR did not affect the percentage of caspase-3[high] liver cells, which suggests that senescence—rather than apoptosis—was induced under these conditions (Supplementary Fig. 3c). To further test the hypothesis that BAX and BAK regulate the SASP, we aged *Bax*[fl/fl]*Bak*[−/−] mice to 20 months and administered either an AAV9-CAG-eGFP or AAV9-CAG-iCre/eGFP virus through tail-vein injection to delete *Bax* (Fig. 4c,d). Expression of pro-inflammatory factors known to significantly increase in the liver during mouse ageing were significantly decreased in the absence of both BAX and BAK (Fig. 4e). In agreement with our in vitro data, no differences in the mRNA expression of the cell-cycle-dependent-kinase inhibitors p21 and p16[INK4a] were evident after *Bax* and *Bak* deletion (Fig. 4f), supporting a central role for BAX and BAK in regulating the SASP, but not the cell-cycle-arrest

component of senescence. Consistent with attenuated inflammation, we observed a significant decrease in the number of infiltrating CD68[+] and CD45[+]CD38[+] immune cells in the liver of *Bak*[−/−]*Bax*[−/−] mice (Fig. 4g,h). Notably, *Bax* mRNA expression was positively correlated with the expression of the majority of proinflammatory factors analysed (Fig. 4i). We also examined the impact of *Bax* and *Bak* deletion in the bone of aged mice. *Bax* and *Bak* deletion in the bone also resulted in decreased mRNA expression of several pro-inflammatory factors that are known to increase during ageing (Supplementary Fig. 3d,e), many of which were positively correlated with *Bax* expression (Supplementary Fig. 3f). These data demonstrate that miMOMP contributes to the SASP in vitro and in vivo.

## mtDNA drives the SASP through cGAS–STING

mtDNA activates cGAS–STING signalling during apoptosis and after other stresses[10,11] and cGAS–STING signalling regulates the SASP[13,14]. However, investigation of the link between cytoplasmic mtDNA release, cGAS–STING and the regulation of the SASP is lacking. To study this, we generated cells lacking mitochondria[23]. Human fibroblasts were stably transduced with YFP-Parkin and senescence was induced by X-ray irradiation. Parkin-expressing senescent cells were treated with the mitochondrial uncoupler CCCP, which triggers widespread mitophagy and generates mitochondria-depleted cells[23]. Cells were then transfected with isolated mtDNA (Fig. 5a). We confirmed by western blotting that the mitochondrial proteins NDUFB8 and UQCRC2 were absent after Parkin-mediated mitophagy and were not affected by transfection with mtDNA (Fig. 5b). mtDNA was undetectable by qPCR in mitochondria-depleted senescent cells and significantly increased after mtDNA transfection (Fig. 5c). Secretion of the common SASP factors IL-6 and IL-8 was suppressed after Parkin-mediated mitochondrial clearance and partially restored after reintroduction of mtDNA (Fig. 5d). RNA-seq analysis confirmed that Parkin-mediated mitochondrial clearance suppressed the expression of commonly known SASP components and that these were subsequently upregulated after transfection with mtDNA (Extended Data Fig. 5a–c).

Aberrant mtDNA packaging due to TFAM deficiency enhances the expression of a subset of interferon-stimulated genes (ISGs)[24]. We used our RNA-seq dataset to investigate whether this subset of mtDNA-stress-induced ISGs was upregulated during senescence and whether its expression was dependent on the presence of mitochondria or mtDNA. The majority of mtDNA-stress-induced ISGs was upregulated in senescence, reduced after mitochondrial clearance and restored after reintroduction of mtDNA (Fig. 5e). Using a more comprehensive database of NF-κB-pathway-regulated genes and ISGs, we found a similar pattern (Extended Data Fig. 5b). To further investigate whether mtDNA could have a role in the regulation of the SASP, we induced senescence by X-ray irradiation in Rho[0] cells lacking mtDNA by prolonged treatment with ethidium bromide. The absence of mtDNA-encoded *MT-ND1* and *MT-ND5* confirmed that the Rho[0] cells do not have any mtDNA (Fig. 5f). Aligned with our hypothesis, we found that the absence of mtDNA led to a significant reduction in the secretion of major SASP components IL-6 and IL-8 (Fig. 5g). To corroborate the involvement of mtDNA in the regulation of the SASP, we modified MRC5 fibroblasts to express a tamoxifen-inducible viral DNase (HSV-1 UL12.5) that specifically targets mitochondria, causing mtDNA depletion[25]. We found that the induction of HSV-1 UL12.5 expression significantly reduced the amount of mtDNA and suppressed the SASP in senescent cells (Extended Data Fig. 6a–e).

Given our data supporting a role for cytosolic mtDNA in cellular senescence, we cultured both wild-type and *Tfam*[+/−] mouse embryonic fibroblasts (MEFs), the latter of which is enriched for cytosolic mtDNA[24], until they reached senescence. We found that *Tfam*[+/−] MEFs reached replicative senescence earlier than wild-type MEFs (Supplementary Fig. 4a) and had increased SA-β-Gal activity and mRNA expression of *Cdkn2a* (encoding p16[INK4a]; Supplementary Fig. 4b,c). *Tfam*[+/−] MEFs also

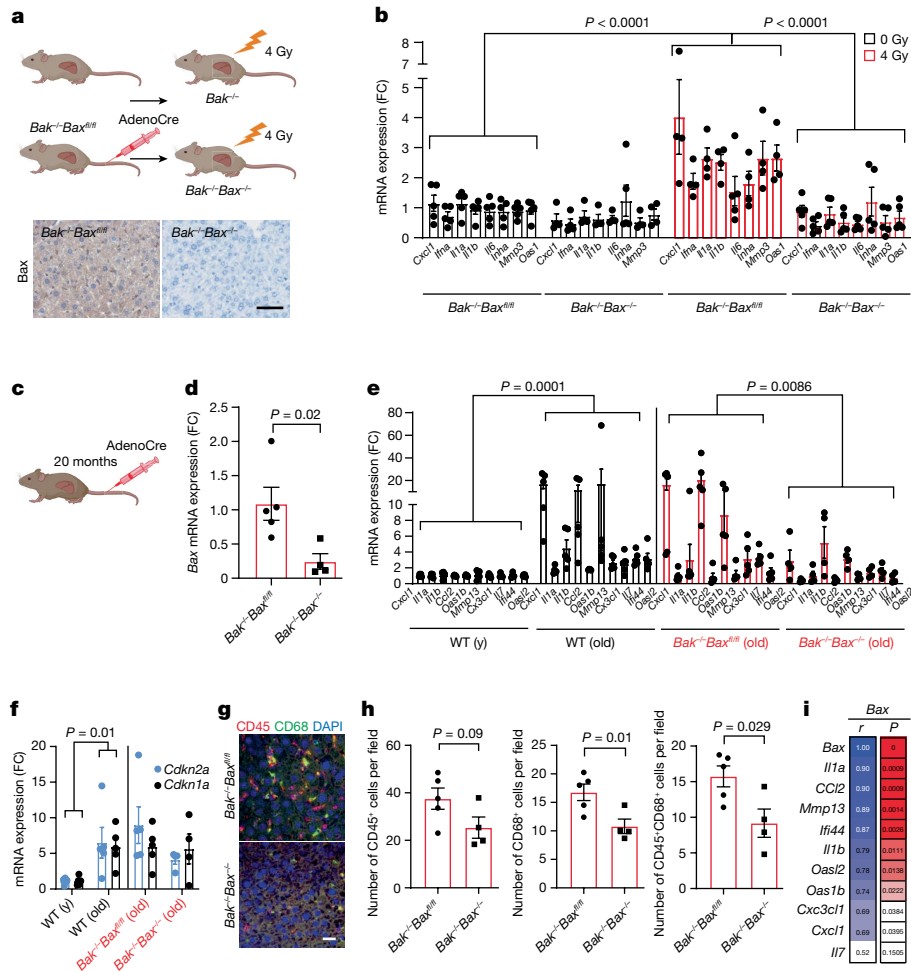

**Fig. 4 | Deletion of *Bax* and *Bak* reduces the SASP in vivo. a**, Schematic of the experimental procedure (top). Bottom, representative immunohistochemical image showing successful deletion of *Bax* in the liver after AAV injection. Scale bar, 100 μm. **b**, Quantification of mRNA levels of the indicated SASP genes in the livers of Sham- and 4-Gy-irradiated *Bax^fl/fl^Bak^−/−* and *Bak^−/−^Bax^−/−* mice. *n* = 5 (sham-IR *Bax^fl/fl^Bak^−/−* and 4Gy-IR *Bak^−/−^Bax^−/−*) and *n* = 4 (sham-IR *Bak^−/−^Bax^−/−* and 4-Gy-IR *Bax^fl/fl^Bak^−/−*) mice. Values are expressed as the fold change compared with sham-irradiated *Bax^fl/fl^Bak^−/−* mice. **c**, Schematic of the experimental procedure. **d**, Quantification of mRNA levels of *Bax* in the livers of aged *Bax^fl/fl^Bak^−/−* mice after tail-vein injection of AAV-Cre virus. *n* = 5 (*Bax^fl/fl^Bak^−/−*) and *n* = 4 (*Bak^−/−^Bax^−/−*) mice. **e**,**f**, Quantification of mRNA expression of the indicated SASP genes (**e**) and of *Cdkn2a* and *Cdkn1a* (**f**) in young (y; *n* = 5) and old (*n* = 5) wild-type mice and aged *Bax^fl/fl^Bak^−/−* mice (*n* = 5) after AAV-Cre virus injection (*n* = 4). **g**, Representative immunofluorescence image of CD45 (red) and CD68 (green) in the livers of aged *Bax^fl/fl^Bak^−/−* (*n* = 5) and *Bax^−/−^Bak^−/−* mice (*n* = 4). Scale bar, 30 μm. **h**, Quantification of **g**. **i**, The correlation coefficient between expression levels of *Bax* and different SASP factors in the livers of aged *Bax^fl/fl^Bak^−/−* and *Bax^−/−^Bak^−/−* mice. Data are mean ± s.e.m. Statistical significance was assessed using two-way ANOVA followed by Tukey's multiple-comparison test (**b**, **e** and **f**), two-sided Student's unpaired *t*-tests (**d** and **h**) and Pearson's correlation coefficient (**i**); *$P < 0.05$, **$P < 0.01$, ***$P < 0.001$.

showed a passage-dependent increase in both NF-κB-dependent activation of pro-inflammatory cytokines *Il6* and *Cxcl15* (Supplementary Fig. 4d,e) and genes associated with a type I interferon response (Supplementary Fig. 4f–h). This is consistent with a recent report indicating that *Tfam* deficiency in T cells accelerates senescence, inflammation and ageing in mice[26]. Administration of the STING inhibitor SN011 did not prevent the senescence-associated growth arrest in *Tfam^+/−^* MEFs nor the expression of the senescence-associated markers SA-β-Gal and p16^INK4a^ (Supplementary Fig. 4i–k); however, it prevented the expression of pro-inflammatory factors (Supplementary Fig. 4l–n), consistent with the established role of cGAS–STING signalling in the regulation of the SASP[13,14].

Further strengthening our hypothesis that mtDNA regulates the SASP through cGAS and STING, we found increased co-localization between cGAS–GFP fusion protein and cytosolic TFAM in senescent human fibroblasts (Extended Data Fig. 7a,b). Moreover, we confirmed that CRISPR–Cas9-mediated deletion of cGAS or STING significantly

reduced the secretion of SASP proteins IL-6 and IL-8 (Extended Data Fig. 7c–h). Lastly, when we transfected mtDNA into WT human fibroblasts, we observed increased secretion of the SASP components IL-6 and IL-8. However, this increase was significantly reduced in cells in which STING was absent (Extended Data Fig. 7i,j).

## Mitochondrial dynamics drive miMOMP

Having established that miMOMP results in cytosolic mtDNA release and contributes to the SASP, we next examined why MOMP is limited to a subset of mitochondria in senescent cells. Activated BAX was immunoprecipitated from mitochondrial fractions of senescent and proliferating cells followed by mass spectrometry (MS) analysis to identify regulators of miMOMP. Supporting earlier data, we were able to immunoprecipitate activated BAX in senescent cells, but not in proliferative cells, indicating that miMOMP is increased in senescent cells. MS analyses uncovered multiple BAX6A7-interacting

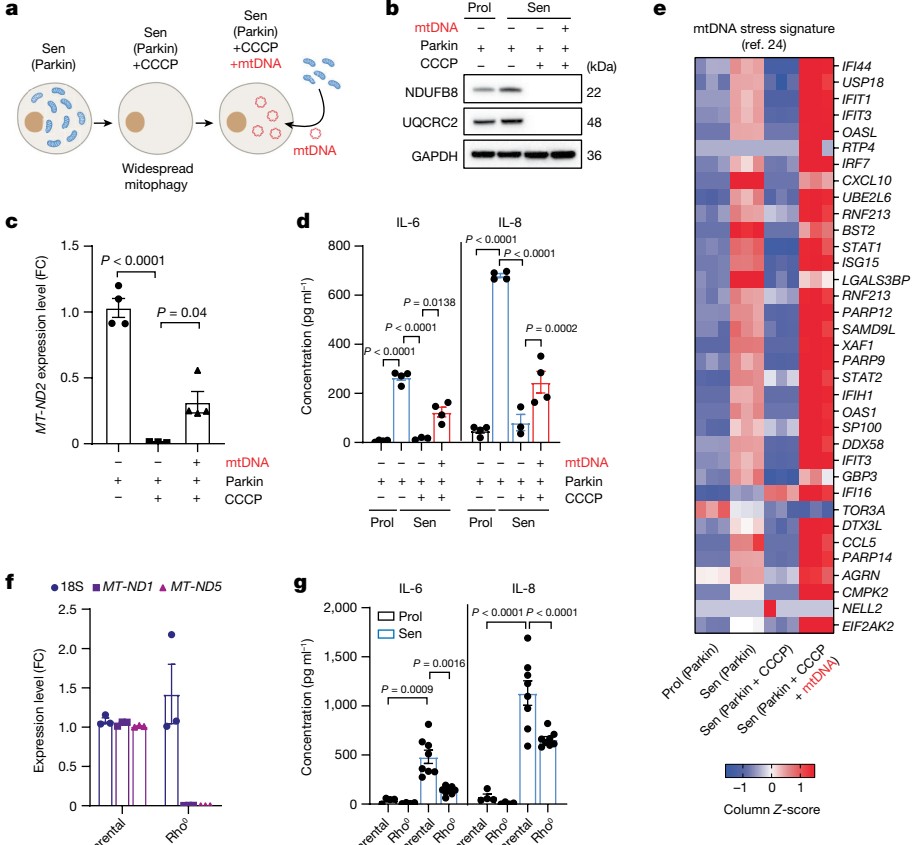

**Fig. 5 | Cytosolic mtDNA drives the SASP in senescent cells. a**, Schematic of the experimental approach. **b**, Western blot analysis of the expression levels of the mitochondrial proteins NDUFB8 and UQCRC2, demonstrating that mitochondrial proteins are absent after Parkin-mediated clearance and are not restored after mtDNA transfection in IMR90 human fibroblasts. **c**, qPCR quantification showing the levels of the mitochondrial gene *MT-ND2* in Parkin-expressing IMR90 fibroblasts after widespread mitophagy and after mtDNA transfection. $n = 3$ (Parkin + CCCP) and $n = 4$ (Parkin control and Parkin + CCCP + mtDNA) independent experiments. **d**, The secretion levels of IL-6 and IL-8 in Parkin cells after mitochondria clearance and after mtDNA transfection. $n = 3$ (Parkin + CCCP) and $n = 4$ (Parkin control and Parkin + CCCP + mtDNA)

independent experiments. **e**, Heat map revealing that the mtDNA stress signature identified previously[24] was induced at the mRNA level in senescent cells, while the addition of CCCP reversed this phenotype. Reintroduction of mtDNA was able to restore this stress signature. **f**, mRNA expression levels of the nuclear-encoded gene 18S and the mitochondrial-genome-encoded genes *MT-ND1* and *MT-ND5* in parental and Rho[0] cells. $n = 3$ independent experiments. **g**, The secreted levels of IL-6 and IL-8 in proliferating ($n = 4$) and senescent ($n = 8$) Rho[0] cells. Data are mean ± s.e.m. Statistical significance was assessed using one-way ANOVA followed by Tukey's multiple-comparison test (**c**, **d** and **g**). Individual data points are from biological replicates. Gel source data for **b** are provided in Supplementary Fig. 1.

mitochondrial proteins with Gene Ontology term functions in mitochondrial dynamics, energy metabolism, lipid homeostasis and other functions (Supplementary Fig. 5). This prompted us to analyse mitochondrial dynamics in senescent cells, finding that most senescent cells contain hyperfused mitochondrial networks, with relatively few dissociated/fragmented mitochondria (Supplementary Fig. 6a). Confocal live-cell imaging of cells labelled using the CellLight Mitochondria-RFP system confirmed previous reports that the rates of mitochondrial fission are lower in senescent cells compared with in proliferating cells[27] (Supplementary Fig. 6b). To investigate whether the few fragmented mitochondria in senescent cells are the ones undergoing MOMP, we immunostained proliferating and senescent cells for TOM20/Cyt *c* and BAX6A7. We observed that only circular, fragmented mitochondria (TOM20+) in senescent cells were negative for Cyt *c* and positive for BAX6A7 (Supplementary Fig. 6c,d) consistent with the hypothesis that this is the subset of mitochondria undergoing MOMP.

Our observations, as well as our recent research demonstrating a key role for mitochondrial dynamics in regulating miMOMP[28], led us to hypothesize that mitochondrial hyperfusion during senescence could be a mechanism to prevent miMOMP and mtDNA release.

To test this, we used shRNA to knockdown mitofusin-2 (*MFN2*) in MRC5 human fibroblasts. Knockdown of *MFN2* alone was sufficient to induce significant mitochondrial fragmentation in proliferating and senescent cells (Extended Data Fig. 8a–d). We induced senescence through X-ray irradiation in cells expressing control shRNA or *MFN2* shRNA. *MFN2* knockdown significantly increased the frequency of mitochondria displaying activated BAX (BAX6A7+) (Extended Data Fig. 8a,e) and the number of cytosolic mtDNA nucleoids (Extended Data Fig. 8b,f) in senescent cells. Senescent cells expressing *MFN2* shRNA had increased expression of the SASP components IL-6, IL-8 and IL-1β (Extended Data Fig. 8g), but no change in the expression of the cyclin-dependent-kinase inhibitors p16[INK4a] and p21 (Extended Data Fig. 8h,i). To validate these findings, we generated *MFN2*-depleted IMR90 fibroblasts using CRISPR–Cas9 genome editing. As before, we observed hyperfragmentation of mitochondria in proliferating and senescent *MFN2*-depleted cells (Supplementary Fig. 7a–c). However, only after induction of senescence did we observe a significant increase in the frequency of mitochondria positive for BAX6A7 (Supplementary Fig. 7d), the number of cytosolic DNA nucleoids (Supplementary Fig. 7e,f) and increased SASP (Supplementary Fig. 7g) in *MFN2*-depleted cells compared with in the controls. To investigate a role

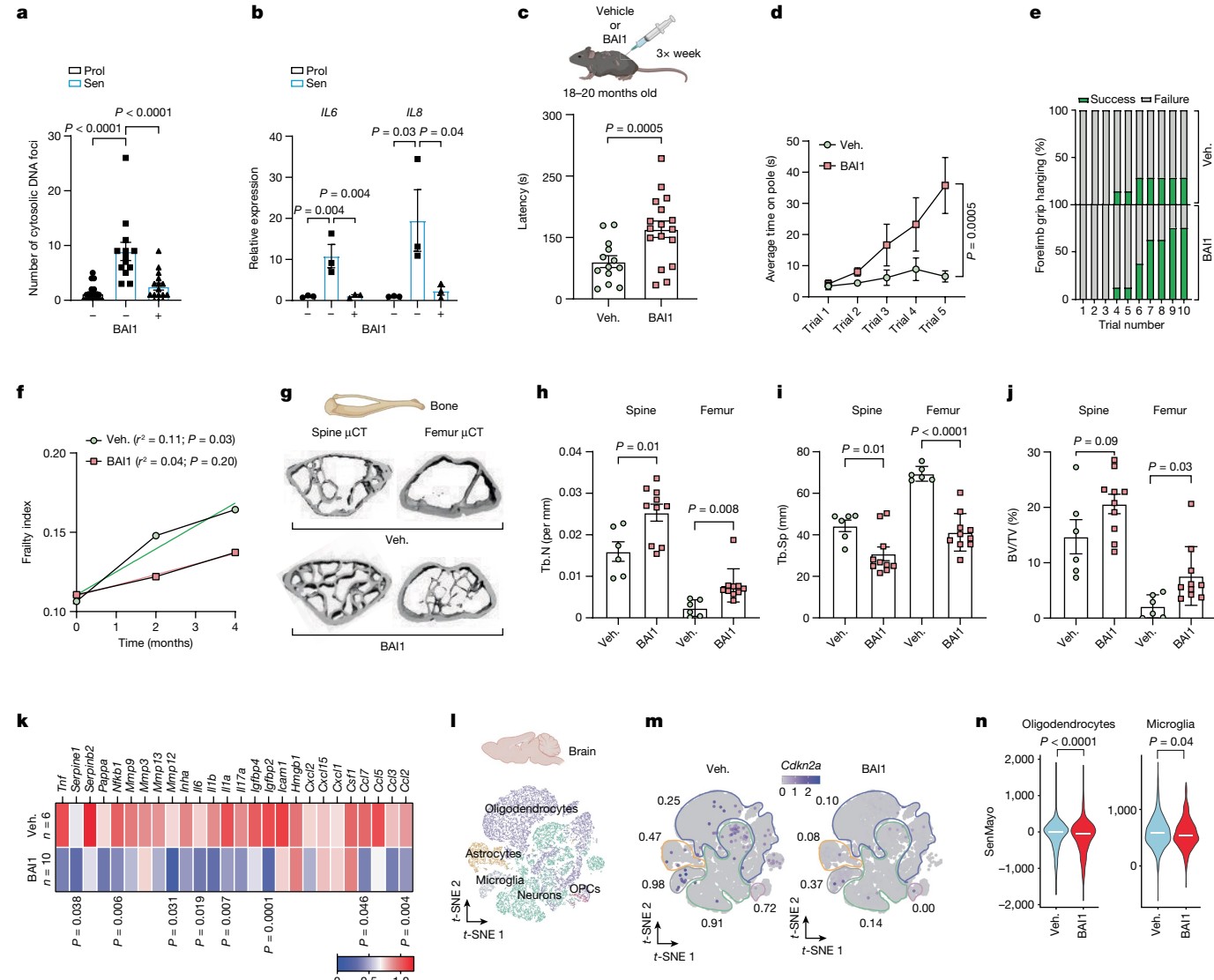

**Fig. 6 | Pharmacological inhibition of BAX improves healthspan in aged mice. a**, Mean cytosolic DNA foci in human fibroblasts treated with BAI1. $n = 20$ (proliferating), $n = 14$ (senescent) and $n = 17$ (senescent + BAI1) cells analysed, representative of 2 independent experiments. **b**, The mRNA levels of *IL6* and *IL8* in proliferating and senescent (IR) human fibroblasts with or without BAI1 treatment. $n = 3$ independent experiments. **c**, The experimental scheme (top). Bottom, rotarod latency in vehicle-treated ($n = 13$) and BAI1-treated ($n = 17$) aged mice. **d**,**e**, The average time spent on the pole (**d**) and forelimb grip strength (the number of trials required to remain hanging for a total of 90 s; percentage success is shown in green) (**e**) in vehicle-treated ($n = 7$) or BAI1-treated ($n = 8$) mice. **f**, The frailty index of mice at 0, 2 and 4 months after treatment with vehicle ($n = 14$) or BAI1 ($n = 15$). The linear regression of the mean frailty index at each timepoint is shown. **g**, Representative μCT images of bone microarchitecture at the lumbar spine and femur of vehicle- and BAI1-treated mice. **h**–**j**, Quantification of μCT-derived trabecular number (Tb.N; per mm)

(**h**) and trabecular separation (Tb.Sp; mm) (**i**) and bone volume fraction (BV/TV; percentage) (**j**). **k**, The mRNA expression of SASP genes was assessed using qPCR with reverse transcription (RT–qPCR) in the femur of vehicle- or BAI1-treated mice. Values are the fold change (FC) compared with the vehicle. For **g**–**k**, $n = 6$ (vehicle) and $n = 10$ (BAI1-treated) mice. **l**, Single-nucleus suspensions from vehicle-treated and BAI1-treated aged mice were prepared from whole brains for RNA-seq analysis. The *t*-distributed stochastic neighbour embedding (*t*-SNE) plots indicate the separation of different cell populations. **m**, BAI1 reduced the fraction of p16[INK4a]-expressing cells across cell populations. **n**, BAI1 significantly reduced the expression of the SenMayo gene set in oligodendrocytes and microglia. Two vehicle-treated and two BAI1-treated mice were pooled for analysis. Data are mean ± s.e.m. Statistical significance was assessed using one-way ANOVA followed by Tukey's multiple-comparison test (**a** and **b**), two-sided Student's unpaired *t*-tests (**c**, **h**–**k** and **n**) and two-way ANOVA followed by Sidak's multiple-comparison test (**d**).

for mitochondrial fission using an alternative method, we examined whether treatment with carbonyl cyanide *m*-chlorophenyl hydrazine (CCCP), which causes extensive mitochondrial fission, exacerbated mtDNA release and the SASP in senescent cells. Indeed, senescent cells treated with CCCP showed increased mitochondrial fragmentation, frequency of cytosolic mtDNA foci and mRNA expression of SASP factors (Supplementary Fig. 8a–d). These data demonstrate that mitochondrial dynamics regulate miMOMP-induced SASP in senescent cells (Supplementary Fig. 8e).

## Inhibiting MOMP improves healthspan

We next assessed whether mtDNA release and the SASP could be suppressed using pharmacological inhibitors of MOMP. We investigated the small-molecule BAX inhibitor BAI1, which inhibits conformational events in BAX activation preventing BAX mitochondrial translocation and oligomerization[29–31]. Aligning with published data, BAI1 treatment had a protective effect against BH3-mimetic induced cell-death in *BAK*-knockout cells, but not in *BAX*-knockout cells, consistent with

a BAX-specific inhibitory effect (Extended Data Fig. 9a,b). We next investigated whether BAI1 impacted senescence. BAI1 was effective at preventing mtDNA release, BAX activation and the SASP in MRC5 and IMR90 senescent human fibroblasts (Fig. 6a,b and Extended Data Fig. 9c–f). Senescent cells treated with BAI1 were also more resistant to ABT263-induced cell death (Extended Data Fig. 9g). Further supporting that the anti-inflammatory effect of BAI1 in senescence was due to BAX inhibition, we found that inflammation induced by transfection with herring testes DNA in human fibroblasts occurred independently of BAX and BAK and was not affected by BAI1 (Extended Data Fig. 9h–j). Finally, treatment of senescent fibroblasts with eltrombopag, an FDA-approved drug that directly inhibits BAX through a different mechanism from BAI1[32], was also effective at suppressing the expression of several SASP factors in senescent fibroblasts (Extended Data Fig. 9k–n).

These results prompted us to investigate pharmacological inhibition of miMOMP as a therapeutic approach targeting senescent cells during ageing. The prevalence of frailty-related characteristics increases with ageing. To investigate whether MOMP inhibition can improve frailty phenotypes during ageing, we treated aged (20 months old) mice with BAI1 for 3 months (Fig. 6c). We found that treatment with BAI1 ameliorated age-related decline in neuromuscular coordination as demonstrated by a significant increase in rotarod latency (Fig. 6c) and improved performance in the pole test, in which BAI1-treated animals were able to maintain balance for a significantly longer duration on a raised rod (Fig. 6d and Extended Data Fig. 10a). Moreover, BAI1 treatment improved forelimb grip strength in old mice and delayed the progression of age-associated frailty symptoms[33] (Fig. 6e,f). Notably, BAI1 treatment improved the healthspan of aged mice without affecting the lifespan (Extended Data Fig. 10b). We also performed a cytokine array in the plasma from vehicle-treated and BAI1-treated mice and, although not statistically significant, we found tendencies for decreased levels of several circulating SASP factors, including IL-1α, IL-6, MCP-1 and TNF (Extended Data Fig. 10c).

Given the improvements in musculoskeletal phenotypes observed in BAI1-treated aged mice, and that senescent cells and the SASP have been shown to have a role in age-related bone loss[34], we performed micro-computed tomography (μCT) analysis to investigate the effects of BAX inhibition on bone microarchitecture (Fig. 6g). BAI1 treatment effectively improved spine and femur trabecular bone microarchitecture (Fig. 6h), such that animals treated with the BAX inhibitor had a significantly increased spine and femur trabecular number and decreased trabecular separation (Fig. 6g–i), as well as a higher bone volume fraction in the femur (Fig. 6j). Analysis of femur cortical bone showed that BAI1-treated animals have increased femur cortical thickness and polar moment of inertia, which is a measure of the bone's resistance to torsion, although these differences were not statistically significant (Extended Data Fig. 10d). Consistent with the hypothesis that miMOMP is a driver of the SASP, we found that BAI1 treatment was effective at significantly reducing the mRNA expression of several pro-inflammatory SASP factors in the bone (Fig. 6k) without affecting genes that are responsible for senescence-associated cell-cycle arrest, such as *Cdkn2a* (encoding p16^Ink4a), *Cdkn1a* (encoding p21) and *Trp53* (encoding p53) (Extended Data Fig. 10e).

We also found evidence for reduced expression of some inflammatory factors in whole brains from aged mice that were treated with BAI1, including significant reductions in *Il6*, *Mmp13* and *Cxcl14* (Extended Data Fig. 10f). Brain pharmacokinetics analysis revealed that BAI1 penetrated the blood–brain barrier in aged mice, reaching concentrations of nearly 1,000 ng per g after 24 h (Supplementary Fig. 9). To elucidate the effects of BAI1 treatment, we conducted single-cell RNA-seq (scRNA-seq), which demonstrated a significant decrease in the frequency of p16[INK4a]-positive cells across all brain cell populations after BAI1 treatment (Fig. 6l,m). Moreover, analysis revealed that BAI1 treatment led to a significant decrease in the senescence gene panel

SenMayo[35] in microglia and oligodendrocytes in which senescence markers have been shown to increase in the ageing brain[36] (Fig. 6n). Given our previous observations indicating that miMOMP is a main driver of the SASP, it is conceivable that the decrease in p16[INK4a]-positive brain cells mediated by BAI1 could be linked to an inhibition of paracrine senescence[37]. These findings highlight the potential of targeting miMOMP as a therapeutic strategy for mitigating inflammation and cellular senescence in the ageing brain.

Mitochondrial dysfunction is a hallmark of ageing and cellular senescence and has been shown to regulate the SASP[2,19]. Our work demonstrates that senescent cells, despite reported resistance to apoptosis[38,39], display features of mitochondrial apoptotic stress without cell death[12]. Our results indicate that BAX and BAK macropores in a small subset of mitochondria are responsible for leakage of TFAM-bound mtDNA nucleoids that contribute to the SASP. Despite the release of mtDNA into the cytoplasm and activation of the cGAS–STING pathway, caspase activation during apoptosis allows this mode of cell death to be largely immunologically silent[20,21]. However, we found that caspase activation due to miMOMP, a process that is dependent on APAF1, is insufficient to dampen cGAS–STING signalling and activation of the SASP in senescent cells.

Several reports indicate that nucleus-derived CCFs and activation of retrotransposons have a role in senescence and the SASP by activating the DNA-sensing cGAS–STING pathway[13,14,40]. We demonstrate that mtDNA is present in the cytosol of senescent cells and can also engage this pathway. We found that pharmacological inhibition of miMOMP (through BAX inhibition) inhibits the SASP and improves various parameters of healthspan. Although inhibitors have potential off-target effects, we found that genetic inhibition of miMOMP also inhibits the SASP in vivo.

miMOMP in senescent cells is probably underpinned by several factors. Senescent cells often display high apoptotic priming[41]. Considering this, we have previously shown a correlation between apoptotic priming and miMOMP[28]. Our recent research has shown that mitochondrial dysfunction, as observed in senescent cells, serves as a mitochondrial intrinsic signal to prime selective MOMP[28]. Finally, we found that mitochondrial fission promotes miMOMP-mediated mtDNA release and the SASP. Activated BAX inhibits mitochondrial fusion[42] most likely limiting the propagation of miMOMP to other mitochondria.

In summary, MOMP is often essential for apoptotic cell death, a terminal cell fate that is considered to be independent from cellular senescence and immunologically silent. Our findings indicate that miMOMP occurs during cellular senescence and can drive the SASP through the release of mtDNA into the cytosol. Importantly, we show that inhibition of miMOMP may be a therapeutic target to counteract age-associated sterile inflammation and improve healthspan.

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

# Methods

## Cell culture and treatments

Human embryonic lung MRC5 fibroblasts (ATCC) and IMR90 fibroblasts (ATCC) were grown in Dulbecco's modified Eagle's medium (Sigma-Aldrich, D5796) supplemented with 10% heat-inactivated fetal bovine serum (FBS), 100 U ml$^{-1}$ penicillin, 100 µg ml$^{-1}$ streptomycin and 2 mM L-glutamine and maintained at 37 °C under 5% CO$_2$. MRC5 fibroblasts were cultured in atmospheric oxygen conditions and IMR90 fibroblasts were cultured under low-oxygen (3%) conditions.

HEK293T cells (ATCC) were used for lentiviral transduction and were cultured in DMEM as described above and further supplemented with 1% non-essential amino acids (Sigma-Aldrich, M7145), 500 µg ml$^{-1}$ G418 antibiotic (Sigma-Aldrich, A1720) and 1 mM sodium pyruvate (Sigma-Aldrich, S8636).

Parental and Rho$^0$ osteosarcoma 143B cells were grown in DMEM as described above with further supplementation using 5% non-essential amino acids and 25 µg ml$^{-1}$ of uridine.

MAFs were isolated from ear clippings and cultured in DMEM/F12 (Thermo Fisher Scientific, 12634010) supplemented with 10% heat-inactivated FBS, 100 µg ml$^{-1}$ streptomycin, 100 U ml$^{-1}$ penicillin (Sigma-Aldrich, P4333) and 2 mM L-glutamine (Sigma-Aldrich, G3126), and maintained at 37 °C under low-oxygen conditions (3% oxygen) with 5% CO$_2$.

WT and *Tfam$^{+/-}$* MEFs were generated from embryonic day 12.5–14.5 mouse embryos and cultured in DMEM (Corning, 10-013-CV) supplemented with 10% FBS (Atlanta Biological). The *Tfam$^{+/-}$* mice were originally derived from *Tfam$^{flox}$* mice obtained from N. Chandel and generated as described previously[24].

Human osteosarcoma U2OS cells (ATCC) were grown in DMEM with 10% FBS, 2 mM L-glutamine, 1 mM sodium pyruvate and 5 mM 2-β-mercaptoethanol. Cells were maintained at 37 °C with 5% CO$_2$.

Stress-induced senescence was achieved by exposing cells to X-ray irradiation at 10 Gy (MAFs) or 20 Gy (human fibroblasts). Replicative senescence was achieved by serially passaging cells until they reached their replicative potential and performed less than 0.5 population doublings for at least 4 weeks. Senescence was confirmed by the presence of p16 and p21, SA-β-Gal positivity, and the absence of proliferation markers Ki-67 or EdU incorporation. OIS was achieved in ER-RAS-IMR90 fibroblasts by treating cells with 100 nM 4-hydroxy-tamoxifen (4-OHT). 4-OHT was maintained in the culture medium until the cells were collected.

For induction of miMOMP, proliferating cells were treated with 2.5 µM ABT-737 (Abcam, ab141336) for 9 and 23 days. Treatment was refreshed every 48–72 h. For inhibition of the MPTP, cells were irradiated with 20 Gy X-ray irradiation and treated with 1 µM cyclosporin A (Sigma-Aldrich, SML1575) for 10 days. For BAX inhibition, MRC5 fibroblasts were irradiated with 20 Gy X-ray irradiation and treated with BAX inhibitor (BAI1) (Adooq Biosciences, A15335) at the indicated concentrations. Cyclosporin A and BAI1 were added straight after irradiation and maintained in the cell culture medium for 10 days (refreshed every 48–72 h).

For cytotoxicity analysis, cells were treated with ABT-263 at the indicated concentrations for 24 h before assessment of cytotoxicity.

For therapy-induced senescence, cells were treated with either 250 nM of doxorubicin or 50 µM etoposide for 24 h. After 24 h, the culture medium was refreshed. Cells were maintained in culture and collected at days 10 and 8 after treatment, respectively.

X-ray-irradiated cells were treated with 10 µM eltrombopag (provided by E. Gavathiotis) for 10 days (treatment was added immediately after irradiation). The treatment was refreshed every 48–72 h. Cells were collected at 10 days after irradiation for analysis.

To induce mitochondrial fragmentation, cells were treated with 12.5 µM CCCP at days 2 and 3 after irradiation. At day 4, the cell culture medium was refreshed, and the cells were maintained in culture until day 10 after irradiation. Cells were collected at day 10 for analysis.

For BAX inhibition, U2OS cells were pre-incubated for 1–2 h with 2.5 µM BAI1 before apoptosis was induced. For induction of apoptosis, cells were treated with 10 µM ABT-737 (Abcam, ab141336) and 2 µM S63845.

The plasmids pFU-GEV16 and pF5XUAS-UL12.5, containing HSV-1 UL12.5, were a gift from G. Hacker. MRC5-UL12.5 cells were generated as described previously[43]. In brief, MRC5 fibroblasts were first transfected with the pFU-GEV16 construct (expression vector that contains the transcriptional activator). Cells were selected with hygromycin and were then infected with lentivirus containing the UL12.5 sequence (tamoxifen inducible) followed by puromycin selection. For induction of UL12.5, cells were treated with 100 nM 4-hydroxytamoxifen for 48 h. Mitochondrial DNA was assessed by qPCR using the Absolute Human Mitochondrial DNA Copy Number Quantification qPCR Kit (ScienCell, 8948) according to the manufacturer's instructions.

All of the cell lines used have been regularly tested for mycoplasma contamination. The cell lines have not been authenticated.

## Parkin-mediated mitochondria clearance

Parkin-mediated widespread mitochondrial clearance was performed as described previously[2,23]. In brief, proliferating or irradiated Parkin-expressing IMR90 fibroblasts were treated with 12.5 µM CCCP (Sigma-Aldrich, C2759) (3 days after irradiation) for 48 h (CCCP was refreshed every 24 h). Mitochondria-depleted cells were then transfected with isolated mitochondrial DNA (as described below) at 7 days after irradiation and collected at 10 days after irradiation.

Parkin-expressing IMR90 fibroblasts were transfected with 15 µg of isolated mitochondrial DNA, 7 days after irradiation using DharmaFECT kb DNA transfection reagent (Horizon, T-2006-01), according to the manufacturer's instructions.

## Subcellular fractionation and mitochondrial DNA extraction

For cytosolic fraction analysis, a total of $7 \times 10^6$ cells was collected and centrifuged at 900g for 5 min. The supernatant was discarded, cells were resuspended in PBS and then divided into two 1.5 ml Eppendorf tubes. After another centrifugation at 600g for 5 min, the pellet from one tube was frozen and considered as the whole-cell fraction. The pellet from the other tube was incubated in 500 µl of buffer 1 (150 mM NaCl, 50 mM HEPES pH 7.4, 25 µg ml$^{-1}$ digitonin (Sigma-Aldrich, D141)) for 10 min at room temperature. Cells were centrifuged at 150g at 4 °C. The supernatant was next centrifuged twice at 150g at 4 °C and once at 17,000g for 10 min, obtaining the cytosolic fraction. Extraction of DNA from the whole-cell fraction was performed using the DNeasy Blood & Tissue Kit (Qiagen, 69504) according to the manufacturer's instructions. The cytosolic fraction was cleaned up using the Qiaquick Nucleotide Removal Kit (Qiagen, 28115) according to manufacturer's instructions and the DNA concentration was measured using the Nanodrop ND-1000 Spectrophotometer.

For the mitochondrial-enriched fraction, followed by a rinse in ice-cold PBS, cells were collected by scraping the flask with 5 ml of ice-cold PBS. Cells were centrifuged at 800g for 5 min at 4 °C and resuspended in mitochondrial isolation solution (MIS) (20 mM HEPES-KOH pH 7, 220 mM mannitol, 70 mM sucrose, 1 mM EDTA, 0.5 mM PMSF, 2 mM DTT). The samples were transferred to a glass homogenizer and cells were broken open using 60 strokes. The homogenate was centrifuged at 800g for 5 min at 4 °C. The supernatant was further centrifuged at 800g for 5 min at 4 °C. An aliquot of the supernatant was collected and stored as the whole-cell extract. The remaining was centrifuged at 16,100g for 10 min at 4 °C. The supernatant was collected as the cytosolic fraction. The pellet containing mitochondria was resuspended in 1 ml of MIS and centrifuged again at 16,100g for 10 min at 4 °C. This step was repeated, and the resulting pellet was resuspended in 100 µl of MIS. For mitochondrial DNA extraction, the mitochondrial pellet was centrifuged at 16,100g for 10 min at 4 °C and the pellet was resuspended in 200 µl PBS. DNA extraction was performed using the DNeasy

Blood & Tissue Kit (Qiagen, 69504) according to the manufacturer's instructions.

## Seahorse analysis

Cellular oxygen consumption rate was measured using Agilent Seahorse XFe96 Analyzer, according to manufacturer's instructions. The cell culture medium was replaced with unbuffered basic medium, 45 mg l$^{-1}$ dextrose, 110 mg l$^{-1}$ sodium pyruvate (Sigma-Aldrich, S8636), 4 mM L-glutamine (Sigma-Aldrich, G3126). The following compounds were added to test mitochondrial activity: 0.5 μM oligomycin, 2.5 μM FCCP, 0.5 μM rotenone with 2.5 μM antimycin A. The resulting oxygen consumption rate values were normalized to cell numbers quantified after the assay using an automated cell counter.

## CRISPR–CAS9-based genome editing

The following plasmids were used: LentiCRISPR v2 hBAK (Addgene, 129579), LentiCRISPR v2 hBAX (Addgene, 129580), LentiCRISPR v2-puro (Addgene, 52961), hMFN2 CRISPR (sgRNA 3194; VectorBuilder, VB900133-9722dcw), *MFN2* shRNA gene set (Horizon, RHS4533-EG9927), Lentiviral pLKO.1 Empty Vector Control (Horizon, RHS4080), *APAF1* CRISPR, *CGAS* CRISPR, LentiCRISPR v2-Blasti.

The following sequences were used to create CRISPR–Cas9-mediated deletion of *STING*, *APAF1* and *CGAS*: hTMEM173_1 5′-GCAAGCATCCAAGTGAAGGG-3′; hTMEM173_2 5′-CGGGCCGACCGCATTTGGGA-3′; APAF1 5′-ACAGCCTGCCATTCCATGTA-3′; CGAS 5′-AAAGTAATATGCACGAGTGT-3′.

For lentiviral transduction, HEK293FT cells were transfected with the plasmids above together with the packaging and envelope plasmids VSVG and Gag-Pol (Sigma-Aldrich) using Lipofectamine 3000 (Invitrogen, L3000015) according to the manufacturer's instructions. Then, 2 days later, the supernatant from the transfected HEK293FT cells containing viral particles was filtered using a 0.45 μm pore PVDF filter, mixed with 10 μg ml$^{-1}$ polybrene and used to infect the cells of interest. After infection, cells were selected for successful CRISPR–Cas9 deletion using the following antibiotics: 1 μg ml$^{-1}$ puromycin (for BAX, BAK and cGAS) or blasticidin 10 μg ml$^{-1}$ (for APAF1 and STING).

## Cytokine array

Detection of cytokines and chemokines in the cell culture supernatant and mouse plasma was performed by Eve Technologies. The following assays were used: Human Cytokine/Chemokine 41-Plex Discovery Assay (HD41) and Mouse Cytokine/Chemokine 31-Ples Discovery Assay Array (MD31), respectively.

Confirmatory ELISAs were performed using the following kits: Human IL-6 DuoSet ELISA (R&D Systems, DY206), Human IL-8 DuoSet ELISA (R&D Systems, DY208) and Human CXCL10/IP-10 DuoSet ELISA, according to manufacturer's instructions. The optical density at 450 nm was determined using the Multiskan FC microplate reader (Thermo Fisher Scientific) and corrected by subtracting the readings at 540 nm.

## Sen-β-Gal activity

Cells grown on coverslips were fixed in 0.2% glutaraldehyde in 2% PFA in PBS for 5 min. Sen-β-Gal staining solution (150 mM sodium chloride, 2 mM magnesium chloride, 40 mM citric acid, 12 mM sodium phosphate pH 6.0, 1 mg ml$^{-1}$ 5-bromo-4-chloro-3-inolyl-β-D-galactosidase (X-gal), 5 mM potassium hexacyanoferrate(II) trihydrate, 5 mM potassium hexacyanoferrate(III) trihydrate) (pH 6.0) was applied and incubated overnight at 37 °C in the dark overnight. Cells were washed in PBS three times and were then mounted onto glass microscope slides using ProLong Gold Antifade Mountant with DAPI (Invitrogen).

## Western blotting

Cells were lysed in lysis buffer (150 mM NaCl, 1% NP40, 0.5% sodium deoxycholate, 0.1% SDS, 50 mM Tris pH 7.4, 1× phosphatase and protease inhibitors cocktail in H$_2$O) and the protein concentration was determined using the Bio-Rad protein assay (Bio-Rad, reagent A, 500-0113; reagent B, 500-0114; reagent C, 500-0115). Equal amounts of protein (at least 15 μg) from each sample were resolved on Tris-glycine gels and samples were then blotted onto a 0.45 μm polyvinylidene difluoride (PVDF) membrane (Millipore) using Trans-Blot SD Semi-Dry Transfer Cells (Bio-Rad). Membranes were blocked with PBS-Tween blocking buffer (5% milk powder, 0.05% Tween-20 in PBS) and then incubated with primary antibodies at 4 °C overnight (a list of the antibodies used is provided in Supplementary Table 2). After washes in distilled water, the membranes were incubated with a peroxidase-conjugated secondary antibody for 1 h at room temperature. The membranes were then incubated with either Clarity ECL Western Blot Substrate (Bio-Ras, 170–5060) or the KwikQuant Western blot detection kit (Kindle Bioscience, R1100) according to manufacturer's instructions, and visualized using the LAS4000 (Fujifilm) or KwikQuant Imager (Kindle Bioscience, D1001) system (uncropped western blots are shown in Supplementary Fig. 1). The signal intensity of protein bands was analysed using ImageJ.

## FPLC

Cells grown in 150 cm$^2$ flasks were trypsinized and pooled to obtain sufficient material for the assay. After PBS washes, cells were centrifuged at 900$g$ for 5 min. Cell pellets were then lysed using CHAPS lysis buffer (1% (w/v) CHAPS, 20 mM HEPES at pH 7.4, 150 mM NaCl, 1% (v/v) glycerol, 1 mM PMSF, 10 μg ml$^{-1}$ leupeptin, 10 μg ml$^{-1}$ pepstatin, 100 mM NaF, 10 mM sodium pyrophosphate, 1 mM sodium vanadate and 20 nM microcystin) for 30 min at 4 °C. The samples were diluted to contain 10 mg ml$^{-1}$ of protein and 200 μl was injected onto a Superdex S200 size-exclusion column. Twenty 500 μl fractions were collected. Protein precipitation using trichloroacetic acid (TCA) was then performed. In brief, the samples were incubated with one-tenth sample volume of 10% Triton X-100 and one-fifth sample volume of 100% ice-cold TCA for 20 min on ice. The samples were then centrifuged for 5 min at 800$g$ at 4 °C, the supernatant was discarded and the pellet was washed once with 1 ml ice-cold 10% TCA and twice with 1 ml acetone at −20 °C. The pellets were left to air dry at room temperature and were then solubilized in sample buffer (4× Laemmli Sample Buffer, Bio-Rad, 1610747) with 1% 2-mercaptoethanol (Bio-Rad, 1610710). Protein was separated using 4–20% gradient acrylamide Tris-Glycine gel generated using Gradient Former (Bio-Rad, Model 230, 165–2700). Protein was transferred to BioTrace NT nitrocellulose membrane (Pall Corporation, 66485) and immunoblotted according to the method described above.

## RNA-seq analysis

Sequencing libraries were made from poly(A) RNA, as recommended by Illumina, and sequenced using either the Illumina GAIIX or a NextSeq 500 sequencer. RNA-seq paired-end reads were assessed for quality using the FastQC algorithm, then aligned to the human genome using the splice-aware aligner STAR with a two-pass alignment pipeline. Reference splice junctions were provided by a reference transcriptome from the Gencode GRCh38 (hg38) build. BigWig files were generated using DeepTools. Raw read counts per gene were calculated using htseq-count. The read count matrix was then used for differential expression analysis with the linear modelling tool DESeq2. Significantly changing expression was defined as a false-discovery-rate-corrected $P \leq 0.05$. Fragments per kilobase of transcript per million mapped reads (FPKM) values were generated using Cufflinks. Gene Ontology analysis was performed using Gene Set enrichment Analysis (GSEA) and Ingenuity Pathway Analysis (IPA) software.

## Mouse models and treatments

All of the animal experiments were performed according to protocols approved by the Institutional Animal Care and Use Committee (IACUC) at Mayo Clinic, unless specified otherwise. Male and female aged wild-type C57BL/6 mice (aged 18–20 months) were acquired

from the National Institute on Aging (NIA) and were maintained in a pathogen-free facility under a 12 h–12 h light–dark cycle at 23–24 °C with free access to regular chow and water. The mice were housed in same-sex cages in groups of 3–5. The animals were randomly assigned into the vehicle or treatment group. Mice were injected intraperitoneally with 10 mg per kg of BAI1 (Tocris Bioscience, 2160) three times a week for 3 months, at which point the animals were euthanized and tissues were collected for analysis. For the lifespan study, BAI1 injections were administered three times weekly until death. Frailty assessment was conducted every 2 months because these measurements are non-invasive. The mice were euthanized and considered to be dead if they met humane end points. Survival was assessed by right-censored Kaplan–Meier curve analysis using the log-rank test.

$Bak^{-/-};Bax^{fl/fl}$ mice (mixed background; male and female) were donated by O. Sansom. Mice were monitored daily and kept in conventional animal facilities. Experiments conducted with $Bak^{-/-};Bax^{fl/fl}$ mice were performed under UK Home Office license and approved by the University of Glasgow Animal Welfare and Ethical Review Board. Mice were genotyped by Transnetyx. AAV-Cre virus in 100 µl PBS (AAV8. TBG.PI.Cre.rBG, UPenn Vector Core, AV-8-PV1091) was delivered by tail-vein injection ($2 \times 10^{11}$ plaque-forming units (PFU) per mouse) in 8-week-old mice. One week after injection, mice were irradiated with 4 Gy. Mice were euthanized 6 days later, and the livers were collected in 10% neutral-buffered formalin.

Aged (17–20 months) $Bax^{fl/fl}Bak^{-/-}$ (mixed background; male and female) were administered either AAV9-CAG-eGFP or AAV9-CAG-iCre/ eGFP virus (in 100 µl PBS; Vector BioLabs) through tail injection ($2 \times 10^{12}$ PFU per mouse). The mice were euthanized 3 weeks after injection, and tissues were collected for analysis.

The animals were randomly assigned numbers at weaning. Once assigned to groups, the genotype or treatment group was not linked to the numbers until data analysis after completion of all studies. Group size estimates were based on power analyses and previous experiences.

Investigators were blinded to allocation during experiments and outcome assessments, and data were collected and analysed in a blinded manner.

### Rotarod tests

Assessment of maximal walking speed and latency was performed using an accelerating rotarod system (Ugo Basile, Rota Rod 47650). Mice were trained on the rotarod for 3 consecutive days before the test day. Training consisted of mice remaining on a rotarod at speeds of 4, 6 and 8 rpm for 200 s on days 1, 2 and 3, respectively. On the test day, mice were placed onto the rotating cylinder, which increased in speed from 4 to 40 rpm over a 200 s interval. The speed and latency at which a mouse fell off the cylinder were recorded. The results were the average of three trials.

### Neuromuscular coordination analysis

Assessment of neuromuscular coordination was performed using the tightrope test[44]. Mice were placed onto a horizontal bar, which was 1.5 cm in diameter and situated 60 cm off the ground. The time that the mice were able to spend on top of the bar was recorded. A trial was deemed to be successful if a mouse could remain on top of the bar for 60 s without falling. Each mouse was given five trials with a 30 s rest between trials.

### Forelimb grip strength analysis (hanging test)

Assessment of forelimb grip strength was performed by allowing the mice to grip a suspended wire coat hanger, which was 2 mm in diameter and 30 cm in length, by using their forelimbs. The time that a mouse was able to hang from the wire following grip was recorded, and each mouse was given 10 attempts (with a 20 s rest in between) up to a total of 90 s. Success was defined as being able to hang for a total sum of 90 s. The trial was defined as failure if the animal fell from the wire.

### Frailty measurements

Frailty was assessed using a 30-parameter index based on a previous study[33]. For each parameter, the mice were given a score of 0, 0.5 or 1 corresponding to absence, mild or severe phenotype, respectively. The body weight was recorded, and the surface body temperature was measured using an infrared temperature probe. For dystonia assessment, a score of 1–4 was given, where a score of 1 was equivalent to clasping with one limb whereas a score of 4 was given if the animal showed clasping with all four limbs.

### Skeletal imaging

All bone imaging and analysis was performed in a blinded manner. Quantitative ex vivo analyses of bone microarchitecture of the lumbar vertebrae ($L_5$) and femur (proximal metaphysis/mid-shaft diaphysis) were performed using a µCT system (Skyscan 1276 Scanner, Bruker). The scan settings were as follows: 55 kV, 200 µA, 10 µm voxel size, 0.4° rotation steps for 360° and 4 frames average imaging with a 0.25 mm A1 filter. Skyscan NRecon software was used to reconstruct scans and for post-alignment and beam hardening corrections. µCT parameters were derived using the manufacturer's protocols for Bruker CtAN software, which permits assessments of trabecular and cortical bone parameters. The trabecular bone volume fraction (BV/TV; percentage) was assessed at the lumbar spine (200 slices) and proximal metaphysis (100 slices) of the femur. Furthermore, at the proximal metaphysis and mid-diaphysis (50 slices) of the femur, the cortical thickness (Ct.Th; mm), endocortical circumference (E.C; mm), periosteal circumference (P.C; mm), and cortical porosity (Ct.Po, percentage) were assessed. Moreover, an estimate of bone torsional strength (that is, polar moment of inertia; mm⁴) was derived.

### qPCR

For bone, osteocyte-enriched cell samples were generated as described previously[45]. The samples were homogenized in QIAzol Lysis Reagent (Qiagen) and immediately stored at −80 °C. Total RNA was extracted using QIAzol Lysis Reagent followed by purification using RNeasy Mini Columns (Qiagen). For the liver, a section measuring approximately 3 mm³ stabilized in RNAlater (Qiagen) was used. RNA from the liver and cells was extracted using the RNAeasy Mini Kit (Qiagen, 74106) according to the manufacturer's instructions (an additional DNase I treatment was performed in the liver to remove genomic DNA contamination). Complementary DNAs were synthesized using the High-Capacity cDNA Reverse Transcription Kit (Thermo Fisher Scientific, 4368814) according to the manufacturer's instructions. qPCR was performed using either Power SYBR Green PCR Master Mix (Invitrogen, 4367659) in a C100TM Thermal Cycler (Bio-Rad), ToughMix Perfecta (PerfeCTa qPCR ToughMix, QuantaBio, 95112-250) using the CFX96TM Real-Time System (Bio-Rad), Brilliant III Ultra-Fast SYBR Green qPCR Master Mix (Agilent Technologies) (for liver samples) or with the ABI Prism 7900HT Real Time System (Applied Biosystems) using SYBR green (Qiagen) as the detection method (for bone). mRNA levels were calculated using the $2^{-\Delta\Delta C_t}$ method and normalized to a housekeeping gene. For 143b and mtDNA-depleted 143b Rho⁰ osteosarcoma cells, DNA was extracted using the GeneJET Genomic DNA purification kit (Thermo Fisher Scientific) according to the manufacturer's instructions. DNA was quantified using the Nanodrop and stored at −20 °C. qPCR was performed using Brilliant III SYBR Green q-PCR Master Mix (Agilent Technologies) using 20 ng genomic DNA. A list of the primers used is provided in Supplementary Tables 3 and 4.

### Immunocytochemistry

Cells grown on coverslips were fixed using 2% paraformaldehyde in PBS for 10 min. Cells were washed in PBS and then permeabilized in PBG-Triton (PBS, 0.4% fish skin gelatin, 0.5% BSA, 0.5% Triton X-100) for 45 min. Subsequently, cells were incubated with primary antibody

overnight at 4 °C. After PBS washes, secondary antibodies were applied and incubated for 45 min at room temperature. Coverslips were mounted onto glass microscope slides with ProLong Gold Antifade Mountant with DAPI (Invitrogen). A list of the antibodies used is provided in Supplementary Table 1.

For mitochondrial proteins, cells were fixed in 0.02% glutaraldehyde in 4% paraformaldehyde for 10 min, washed in PBS and permeabilized in 0.5% Triton X-100 in PBS for 10 min. Cells were blocked in 5% normal goat serum (NGS) for 1 h. Incubation with primary and secondary antibodies was performed as described above. Labeling of mitochondria using CellLight Mitochondria-RFP BacMam 2.0 (Thermo Fisher, C10601) was performed following manufacturer's instructions.

## Immunohistochemistry

Formalin-fixed paraffin-embedded tissue sections (5 μm) were deparaffinized in xylene (3 times for 3 min each) and hydrated using 100% ethanol (twice for 5 min), 70% ethanol (5 min) and distilled water (twice for 3 min each). Antigen retrieval was done by heating the sections to 98 °C in citrate buffer pH 6.0 (Agilent-Dako, S236984) for 30 min. The slides were allowed to cool down for 30 min and were then rinsed in TBS twice for 5 min. Tissue sections were blocked with NGS (Agilent-Dako, X090710-8) for 30 min at room temperature followed by primary antibody incubation overnight at 4 °C. The sections were washed in TBS and incubated with peroxidase block (Agilent-Dako, K500711) for 30 min at room temperature. HRP-conjugated secondary antibody was then applied for 30 min at room temperature, followed by TBS washes and DAB incubation for 5–10 min. The sections were counterstained with haematoxylin according to a standard procedure. A list of the antibodies used is provided in Supplementary Table 1.

## Immuno-FISH

Formalin-fixed paraffin-embedded tissue sections were deparaffinized in 100% Histoclear, hydrated through a graded ethanol series of 100, 90 and 70% ethanol (twice for 5 min each), and washed twice for 5 min in distilled water. Antigen retrieval was performed by placing the sections in 0.01 M citrate buffer (pH 6.0) and heating it until boiling for 10 min. The sections were allowed to cool to room temperature and then washed in distilled water for 5 min. Blocking was then performed using normal goat serum (1:60) in BSA/PBS for 30 min followed by overnight incubation with rabbit monoclonal anti-γH2AX antibodies (1:400, 9718; Cell Signaling) at 4 °C. After three PBS washes, the tissues were incubated with a goat anti-rabbit biotinylated secondary antibody (1:200, PK-6101; Vector Labs) for 30 min at room temperature. The sections were then washed three times in PBS and incubated with fluorescein avidin DCS (1:500, A-2011; Vector Labs) for 30 min at room temperature. The tissues were then washed three times in PBS and incubated in 4% paraformaldehyde in PBS for 20 min for cross-linking. After three PBS washes, the sections were dehydrated in graded cold ethanol solutions (70, 90, 100%) for 3 min each and were then allowed to air dry. Next, 10 μl of PNA hybridization mix (70% deionized formamide (Sigma-Aldrich), 20 mM MgCl$_2$, 1 M Tris pH 7.2, 5% blocking reagent (Roche) containing 2.5 μg ml$^{-1}$ Cy-3-labelled telomere-specific (CCCTAA) peptide nucleic acid probe (PANAGENE)) was added to sections and denaturation was allowed to occur for 10 min at 80 °C. The sections were then incubated in PNA hybridization mix for 2 h at room temperature in the dark to allow hybridization to occur. Tissues were washed in 70% formamide in 2× SSC for 10 min, followed by one wash in 2× SSC for 10 minutes and a PBS wash for 10 min. Tissues were mounted using ProLong Gold Antifade Mountant with DAPI (Invitrogen) and imaged using in-depth z stacking (a minimum of 40 optical slices with a ×63 objective).

## Immunogold electron microscopy

Cells were fixed in 0.1% glutaraldehyde and 4% paraformaldehyde in 0.1 mol l$^{-1}$ phosphate buffer for 2 h and were then collected and centrifuged at 900g for 5 min. For cryoprotection, cells were incubated with 2.3 mol l$^{-1}$ sucrose in 0.1 mol l$^{-1}$ phosphate buffer overnight and then frozen in liquid nitrogen. Thin cryosections (60 nm) were cut using the Leica cryo-microtome. The sections were incubated with primary antibodies (1:20) at 4 °C overnight. The sections were then incubated with a 10 nm anti-mouse IgG gold secondary antibody (Sigma-Aldrich, G7652) for 2 h at room temperature. After washes, the sections were fixed in 1% glutaraldehyde and embedded in 2% methyl cellulose solution containing 0.3% uranyl acetate. The samples were imaged using the Jeol 1200 electron microscope (Mayo Clinic Core Microscopy Facility) operating at 60 to 80 kV.

## Lactate dehydrogenase cytotoxicity assay

Cytotoxicity was assessed using the lactate dehydrogenase assay (Abcam, ab65939) according to the manufacturer's instructions. In brief, cells were grown in 24-well plates and treated as indicated. On the day on which the assay was performed, 50 μl of cell culture medium was mixed with 50 μl of LDH reaction mix and pipetted into a 96-well plate. The absorbance was measured using a plate reader with a 450 nm filter.

## Incucyte

U2OS cells were seeded at approximately 20,000 cells per well in a 24-well plate. Cell death assays were performed using Incucyte S3 software (Sartorius) and Sytox Green at 30 nM as cell death readout, which is taken up into cells after cell death. Cell death was induced using ABT-737 and S63845 at 10 μM and 2 μM, respectively. Images (four images per well) were taken every hour for 24 h at ×10 magnification. Analysis was performed using Incucyte S3 2022 software with Sytox count per well normalized to cell confluency.

## HT-DNA transfection

Proliferating pLenti or *Bax*$^{-/-}$*Bak*$^{-/-}$ cells were transfected with herring testes DNA (HT-DNA; 1 μg ml$^{-1}$; Sigma-Aldrich, D6898) using Lipofectamine 3000 Reagent (Thermo Fisher Scientific, L3000008). To assess the specificity of BAI-1, cells were treated with either DMSO or BAI-1 (2.5 μM) 24 h before transfection. Cells were collected for analysis 20 h after transfection with HT-DNA.

## Determination of activated BAX interactome in mitochondria from senescent cells

**Preparation of tissue cultures and mitochondria isolation.** Cellular senescence was induced with 200 nM doxorubicin for 24 h and developed for 9 days. A small fraction of proliferative and senescent cells was pelleted and kept for MS analysis (total cell control samples). The remaining cells were resuspended in mitochondria isolation buffer (MIB: 20 mM HEPES pH 7.6, 220 mM mannitol, 70 mM sucrose, 1 mM EDTA) and homogenized with a Dounce homogenizer followed by centrifugation at 850g for 5 min, 4 °C. Crude mitochondria were pelleted at 10,000g for 5 min at 4 °C and washed once more. A total of 50 μg of mitochondria was kept for further MS analysis (crude mitochondria control samples).

**Co-immunoprecipitation of activated BAX interactors.** A total of 400 μg of mitochondria was resuspended in ice-cold co-IP buffer (20 mM HEPES pH 7.6, 220 mM mannitol, 70 mM sucrose, 1 mM EDTA, 1% CHAPS (w/v), protease inhibitor cocktail), lysed on ice for 30 min and cleared from insoluble material at 10,000g, 30 min, 4 °C. A total of 1 μg of anti-BAX6A7 antibody (NBP1-28566, Novus Biologicals) was immobilized on Protein G-coupled Dynabeads (Invitrogen). The beads were washed twice with co-IP buffer and combined with cleared mitochondrial extracts. Activated BAX complexes were immunoprecipitated overnight at 4 °C with gentle rocking. The beads were washed three times with ice-cold co-IP buffer and twice with CHAPS-free co-IP buffer.

**On-bead protein digestion.** The beads were resuspended in the 2 M urea, 5 mM dithiothreitol, 50 mM triethylammonium bicarbonate

and 50 ng trypsin (Promega), and incubated for 30 min at room temperature with gentle agitation. The samples were mixed with 40 mM chloroacetamide and incubated for 30 min at room temperature with gentle agitation. The supernatants with eluted proteins were collected. A total of 50 ng of LysC (Promega) was added to protein eluates and supplemented with 100 ng of trypsin. Proteins were digested overnight at 37 °C with vigorous shaking. Peptides were desalted on the Affinisep SPE-Disks-C18 and separated by liquid chromatography.

**Processing of the control samples.** Cell and mitochondrial pellets were resuspended in 8 M urea, 50 mM triethylammonium bicarbonate and protease inhibitor cocktail. Chromatin was degraded in a water bath sonicator (10 min, cycle 30/30 s), followed by benzonase HC nuclease treatment. Insoluble material was removed by centrifugation at 20,000*g*, 4 °C. A total of 50 µg of proteins was reduced for 1 h in 5 mM dithiothreitol and incubated with 40 mM chloroacetamide for 30 min. The samples were subjected to sequential LysC and trypsin digestion (1:75 enzyme to substrate ratio, overnight, 37 °C). Peptides were desalted on the Affinisep SPE-Disks-C18 and separated by liquid chromatography.

**MS and sample analysis.** The measurements were performed at the Proteomic Core Facility of IMol Polish Academy of Sciences using the Dionex UltiMate 3000 nano-LC system coupled to a Q-Exactive HF-X through an EASY-Spray ion source (Thermo Fisher Scientific). The data were processed with MaxQuant 1.6.17.0, and the peptides were identified from the MS/MS spectra searched against human reference proteome UP00000564 using the built-in Andromeda search engine. The data were subjected to non-quantitative analysis.

### Brain pharmacokinetic analysis

Aged (>16 months old) C57BL/6 J male mice were fasted overnight and water was available ad libitum. The mice had access to certified rodent diet ad libitum 4 h after dosing. The mice were housed in a controlled environment, under the following target conditions: temperature, 20–26 °C; relative humidity, 40 to 70%. The temperature and relative humidity were monitored daily. An electronic time-controlled lighting system was used to provide a 12 h–12 h light–dark cycle. Three mice for each indicated timepoint were administered BAI1(10 mg per kg) in 3.5% DMSO, 14% PEG-400, 75% gamma-cyclodextrin (20%) in 0.1 M citrate buffer pH 6, 7.5% 0.1 M citrate buffer pH 6 through the intraperitoneal route. The mice were euthanized and brain tissue was collected at 0 h, 1 h, 2 h, 4 h, 8 h and 24 h. Sample collection at 0 h was to collect brain tissue immediately after dose administration. Brain homogenates were prepared by homogenizing tissue with 4 volumes (w/v) of homogenizing solution (MeOH/15 mM PBS 1:2) and analysed for BAI1 concentration using liquid chromatography coupled with tandem MS bioanalysis.

### Microscopy

Imaging was performed using confocal microscopes (SP8 Leica and LSM780 Zeiss) and super-resolution microscopes (confocal microscopy using the AiryScan type detector LSM800 Zeiss AiryScan;and SIM using the Zeiss Elyra PS.1 Super Resolution system).

Analysis of extramitochondrial DNA foci and mitochondria positive for BAX(6A7) was performed manually using ImageJ.

For the visualization of cytosolic Cyt *c*, CLSM images were processed for deconvolution using AutoQuant X3 Deconvolution. The Pearson's *R* value was subsequently determined for each cell using the Coloc2 ImageJ plugin.

For the 3D visualization of mitochondrial leakage, Imaris 9.6 Image Visualization and Analysis Software was used.

Images for miMOMP analysis were acquired from cells labelled with antibodies against Cyt *c* and TOM20 by 3D imaging using *z*-stacks on the Leica SP8 Confocal microscope. Approximately 30 *z*-stacks were acquired for each coverslip imaged. After imaging, micrographs were processed using the Huygens Deconvolution Software. The level of colocalization was assessed using the colocalization tool within the Huygens Deconvolution Software.

### Single-cell gene expression flex assay

Fixed, single-cell suspensions were prepared from flash-frozen mouse brain tissue using the Chromium Next GEM Single Cell Fixed RNA Preparation kit (10x Genomics) according to the user guide (CG000553, Rev B). Minced tissues were fixed for 24 h at 4 °C. Cell dissociation was performed using the gentleMACS Octo Dissociator (Miltenyi). Libraries were constructed using the single-plex Chromium Fixed RNA Profiling kit (CG000477, Rev C). Probes were incubated for 20 h at 42 °C and 9 cycles were performed for the indexing PCR. Libraries were sequenced on the NextSeq500 (Illumina) system.

### Statistical analysis

GraphPad Prism v.9.0 was used for statistical analysis; the results were considered to be statistically significant when $P \leq 0.05$. For normally distributed data, the differences between two groups were tested for statistical significance using an independent-sample two-tailed *t*-tests. For data that were normally distributed and when there was more than one group, one-way ANOVA was used, with Tukey's comparison post hoc test. Where data were not normally distributed, Mann–Whitney *U*-tests were used to determine statistical significance.

### Ethics statement

All animal experiments were performed according to protocols approved by the Institutional Animal Care and Use Committee (IACUC) at Mayo Clinic.

### Reporting summary

Further information on research design is available in the Nature Portfolio Reporting Summary linked to this article.

## Data availability

The RNA-seq datasets generated and analysed during this study are available at the Gene Expression Omnibus (GSE196610 and GSE235225). The MS proteomics data have been deposited at the ProteomeXchange Consortium through the PRIDE partner repository under dataset identifier PXD040018. Source data are provided with this paper.

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

**Acknowledgements** This work was funded by NIH grants R01AG068048 (to J.F.P.), UG3CA268103 (to J.F.P.), P01 AG062413 (to S.K., J.N.F., N.K.L. and J.F.P.), P01 AG073084 and R01 AR069876 (to G.S.S.), F31 AG062099 (to A.G.S.), R01 AG068182-01 (to D.J.), R01 CA225996 (to S.H.K. and H.D.), R01 DK128552 (to J.N.F.), R01 AG071861-01 (to X.L. and P.D.A.), P01 AG073084 (to P.D.A. and X.L.), R01 AG076515 (to S.K.), U54 AG079754 (to S.K.), R01 AG061875 and RF1 AG070391 (to T.Y.H.), R01AG071861 (to P.D.A.) and R33AG61456-4 and R01AG064165; Department of Defense grant W81XWH-20-1-0792 (to E.G.); U54 AG79754 (to S.K. and N.K.L.); Cancer Research UK grants C40872/A2014, DRCNPG-Jun22\100011 (to S.W.G.T.); the Ted Nash Long Life Foundation (to J.F.P. and D.J.); The Glenn Foundation For Medical Research (to J.F.P. and N.K.L.); Hevolution/AFAR (to D.J.); a Robert and Arlene Kogod Center on Aging Career Development Award (to S.V.); a fellowship from Coordenação de Aperfeiçoamento de Pessoal de Nível Superior—Brasil (Capes)—Finance Code 001 and by the Universidade Federal do Rio de Janeiro (to C.A.M.); Cancer Research UK grant DRCPFA-Nov22/100001 and Wellcome grant 203105/Z/16/Z0 to L.C.G., BBSRC PhD studentship BB/R506345/1 (to G.K. and V.I.K.); UKRI cross-council Newcastle University Centre for Ageing and Vitality Ph.D. studentship MR/L016354/1 (to J.C.) and CRUK Awards C18342/A23390 and DRCRPG-Nov22/100007 and MRC MR/R023026/1 (to D.A.M.). G.S.S. holds the Audrey Geisel Chair in Biomedical Science. MS interactome analysis was performed at the Proteomic Core Facility of IMol Polish Academy of Sciences and supported by ReMedy International Research Agenda (Foundation of Polish Science, MAB/2017/2) and EMBO Installation Grant 5040-2022 (to K.S.). scRNA-seq was performed by the Sanford Burnham Prebys Genomics Core. G.S.S. holds the Audrey Geisel

Chair in Biomedical Science and is supported by P30AG068635 as director of the San Diego Nathan Shock Center. We thank G. Nelson and the members of the Newcastle University Bioimaging Unit for technical support.

**Author contributions** S.V. was involved in mouse and cell culture experiments, analysis of end points, conceptualization, interpretation and editing the manuscript. H.S. and J.C. were involved in cell culture experiments, analysis of end points, conceptualization and interpretation of data. H.M., M.G.V., E.H.-Y., A.B.L., G.K., M.E., C.A.M. and A.G.S. were involved in cell culture experiments and analysis of end points. J.S.R., C.C., J.M.E.-N., L.S.G. and N.P. were involved in animal experiments and analysis of end points. R.R. conducted Seahorse analysis. J.N.F. and S.K. were involved in conceptualization and functional assessment of bone function. H.D. conducted analyses of BAX oligomerization through FPLC under the supervision of S.H.K. K.S. conceptualized and supervised BAX6A7 co-IP and subsequent proteomics analysis. D.J. oversaw animal experiments and analysis of end points. R.A.P., H.H. and T.Y.H. performed scRNA-seq analysis of the mouse brain. E.Z. conducted pharmacological analyses of BAI1 in the brain under the supervision of E.G. D.S. and X.L. conducted analyses of RNA-seq datasets. L.C.G. and D.A.M. were involved in the supervision of J.C. N.K.L. was involved in the supervision of H.S. V.I.K. oversaw cell culture and microscopy experiments performed by H.S. and G.K. G.S.S. supervised cell culture studies involving $Tfam^{+/-}$ fibroblasts performed by A.G.S. Z.D. provided the *CGAS* shRNA lentiviral construct. P.D.A. oversaw RNA-seq analyses. J.F.P. and S.W.G.T. conceived the idea, oversaw the study and wrote the manuscript. All of the authors contributed to reviewing or editing the manuscript.

**Competing interests** V.I.K. is a scientific advisor for Longaevus Technologies. S.W.G.T. consults for Exo Therapeutics. The other authors declare no competing interests.

**Additional information**
**Correspondence and requests for materials** should be addressed to Stephen W. G. Tait or João F. Passos.

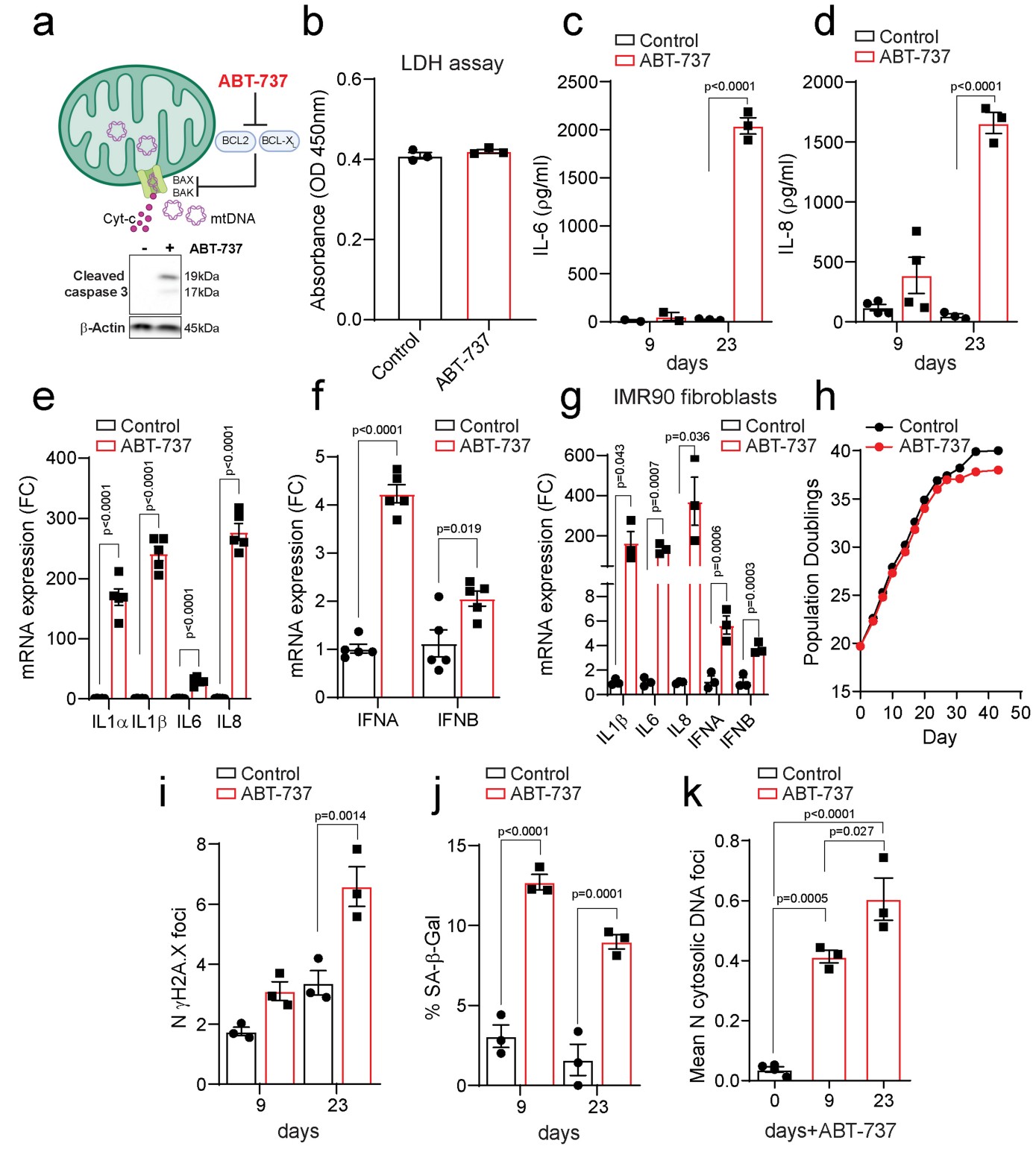

**Extended Data Fig. 1 | ABT-737 treatment induces miMOMP and drives a SASP-like response.** (**a**) Scheme representing the mechanism by which ABT-737 induces miMOMP. Below, representative Western blot showing cleaved caspase 3 in proliferating and ABT-737- treated MRC5 fibroblasts. (**b**) Absorbance values (at 450 nm) as a measure for lactate dehydrogenase (LDH) release from proliferating MRC5 fibroblasts treated with vehicle or ABT-737 for 72 h (n = 3 independent experiments), showing no difference in cell death. Levels of secreted (**c**) IL-6 (n = 2 and n = 3 independent experiments, 9 and 23 days, respectively) and (**d**) IL-8 in control and MRC5 fibroblasts treated with ABT-737 for 9 (n = 4 independent experiments) and 23 days (n = 3 independent experiments). Quantification of mRNA expression levels of interleukins (IL-1α,β, IL 6 and IL-8), interferon genes (IFN-α and β) in MRC5 (n = 5 independent experiments) (**e-f**) and IMR90 fibroblasts (n = 3 independent experiments) (**g**) 23 days after ABT-737 treatment. (**h**) Graph showing population doublings of control and ABT-737-treated MRC5 fibroblasts. Quantification of (**i**) mean number of γH2AX foci, (**j**) percentage of Sen-β-Gal-positive cells and (**k**) mean number of DNA nucleoids located outside of the mitochondrial network in control and MRC5 fibroblasts treated with ABT-737 for 9 and 23 days (n = 3 independent experiments; n = 4 for day 0). Data are mean ± S.E.M. Statistical significance was assessed using one-way ANOVA followed by Tukey's multiple comparison test (k), two-sided Student's unpaired t-test (b, e-g), two-way ANOVA followed by Sidak's multiple comparison test (c, d, i, j).

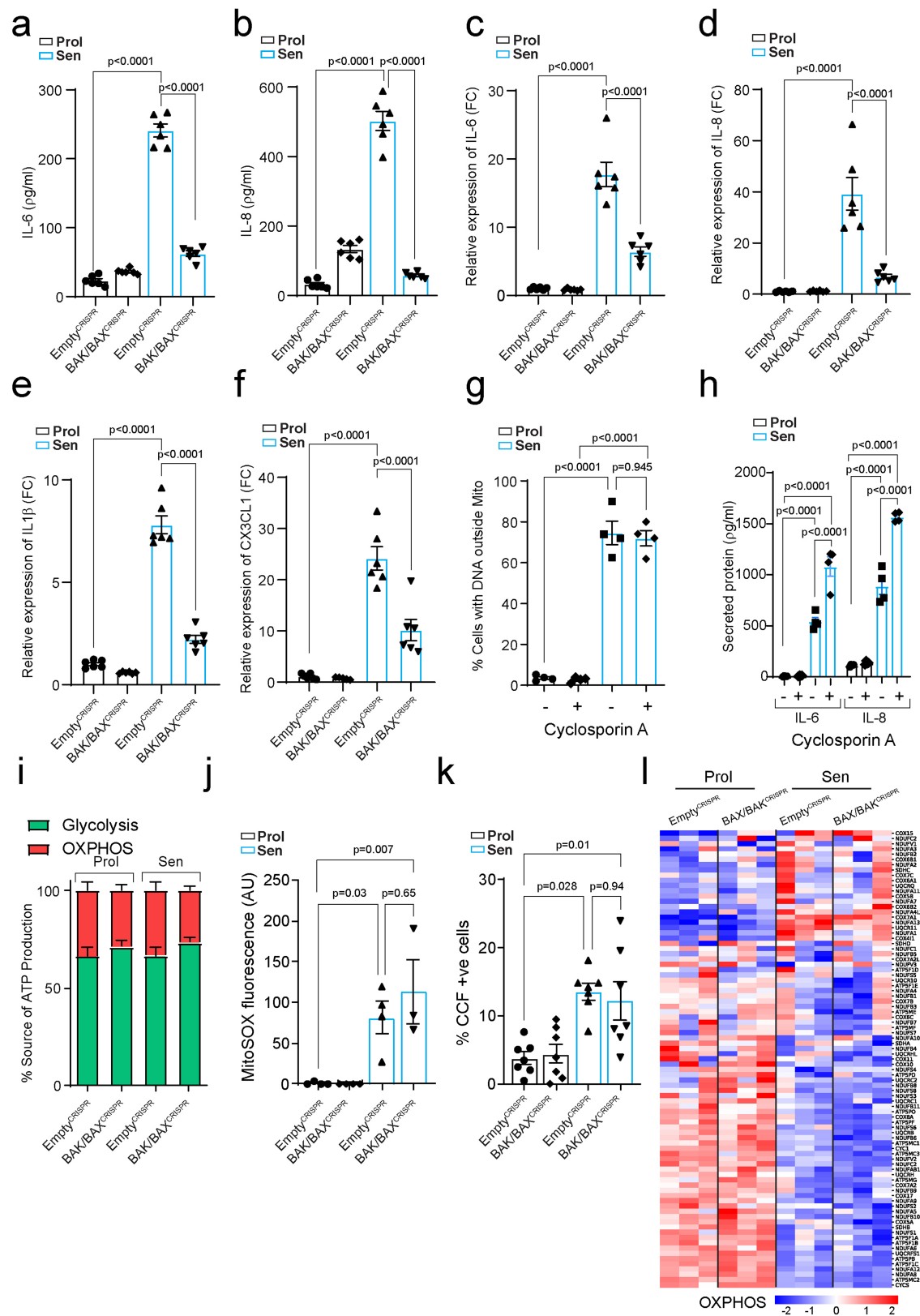

**Extended Data Fig. 2** | See next page for caption.

**Extended Data Fig. 2 | BAX and BAK mitochondrial pores regulate the SASP without the involvement of the MPTP.** Levels of secreted (**a**) IL-6 and (**b**) IL-8 in proliferating and senescent (IR) EV and BAX/BAK-/- cells (n = 6 independent experiments). Quantification of mRNA levels of (**c**) IL-6, (**d**) IL-8, (**e**) IL1β and (**f**) CX3CL1. Data are expressed as fold change to proliferating EV cells (n = 6 independent experiments). (**g**) Percentage of cells containing DNA foci located outside of mitochondrial network, and (**h**) Levels of secreted IL-6 and IL-8 in proliferating and senescent cells treated with DMSO or 1 μM Cyclosporin A (n = 4 independent experiments). (**i**) Graph showing the energy production by glycolysis or oxidative phosphorylation normalized to total ATP production and expressed as a percentage (n = 5 independent experiments; n = 4 Sen BAX/BAK$^{CRISPR}$). (**j**) Quantification of MitoSOX fluorescence intensity in proliferating and senescent EV and BAX/BAK$^{-/-}$ cells (n = 4 independent experiments for Prol and Sen EV and Prol BAX/BAK$^{-/-}$ cells; n = 3 independent experiment for Sen BAX/BAK$^{-/-}$ cells). (**k**) Quantification of the percentage of EV and BAX/BAK$^{-/-}$ cells displaying Cytoplasmic chromatin Fragments (CCF) (n = 7 independent experiments). (**l**) Column clustered heatmap of OXPHOS genes that are differentially expressed in senescent cells and are not changed by BAX/BAK deletion. The colour intensity represents column Z-score, where red and blue indicate high and low expression, respectively. Data are mean ± S.E.M. Statistical significance was assessed using one-way ANOVA followed by Tukey's multiple comparison test (a-k).

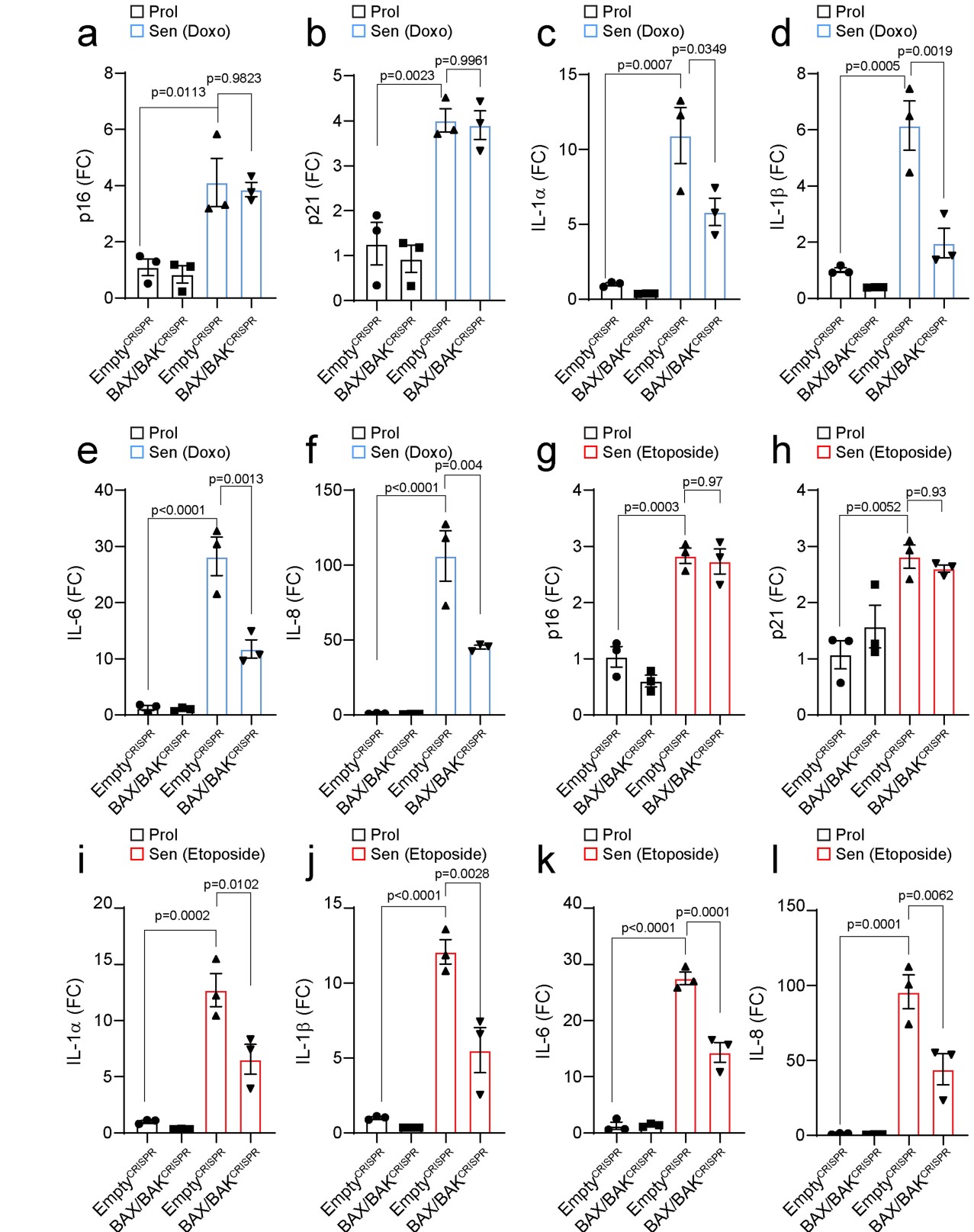

**Extended Data Fig. 3 | BAX and BAK deletion inhibits the SASP in therapy-induced senescent cells.** Quantification of mRNA expression levels of (**a**) p16[INK4A], (**b**) p21 and SASP factors (**c**) IL-1α, (**d**) IL-1β, (**e**) IL-6 and (**f**) IL-8 in proliferating and doxorubicin-induced senescent (n = 3 independent experiments) BAX/BAK[−/−] MRC5 human fibroblasts. Quantification of mRNA levels of (**g**) p16[INK4A], (**h**) p21, (**i**) IL-1α, (**j**) IL-1β, (**k**) IL-6 and (**l**) IL-8 in proliferating and etoposide-induced senescent (n = 3 independent experiments) BAX/BAK[−/−] MRC5 human fibroblasts. Data are expressed as fold change to proliferating Empty[CRISPR] cells. Data are mean ± S.E.M. Statistical significance was assessed using one-way ANOVA followed by Tukey's multiple comparison test (a-l).

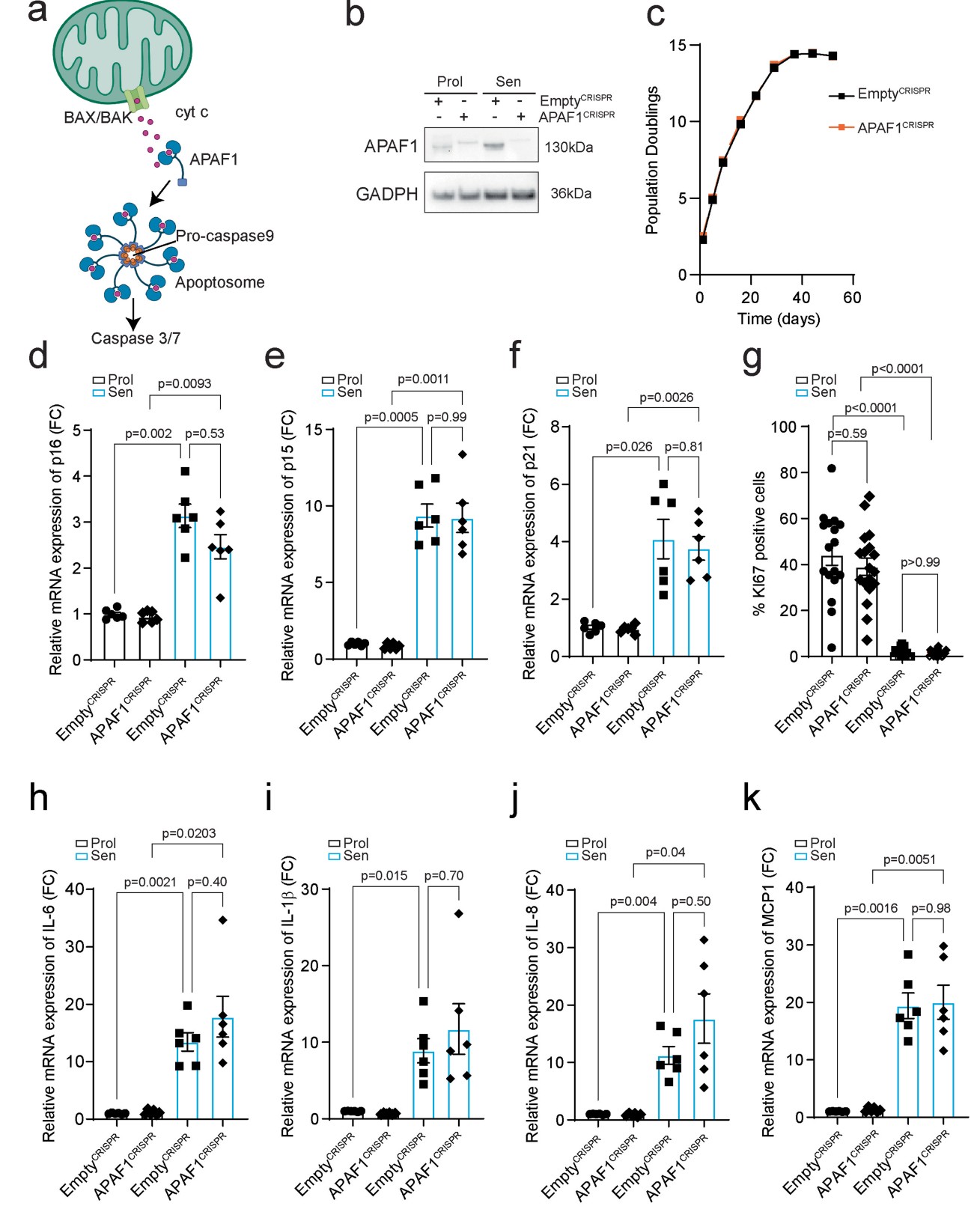

**Extended Data Fig. 4 | Caspase activation is not involved in SASP regulation in senescent cells.** (**a**) Scheme showing the mechanism of miMOMP-induced caspase activation. (**b**) Western blot showing successful CRISPR/Cas9 deletion of APAF1 in proliferating and senescent (IR) MRC5 human fibroblasts. Representative of n = 2 experiments. (**c**) Graph showing population doublings of EV and APAF1-deficient cells. Data are representative of n = 2 independent experiments. Quantification of mRNA levels of (**d**) p16INK4A, (**e**) p15, (**f**) p21 (n = 6 independent experiments). (**g**) Quantification of the percentage of EV and APAF1 deficient cells positive for Ki67. Each dot represents the average per image (n = 3 technical replicates; 6 images taken per replicate). Quantification of mRNA levels of (**h**) IL-6, (**i**) IL-1β, (**j**) IL-8, (**k**) MCP1 in proliferating and senescent (IR) EV and APAF1-deficient cells (n = 6 independent experiments). Data are mean ± S.E.M. Statistical significance was assessed using one-way ANOVA followed by Tukey's multiple comparison test (d-k). For gel source data (b), see Supplementary Fig. 1.

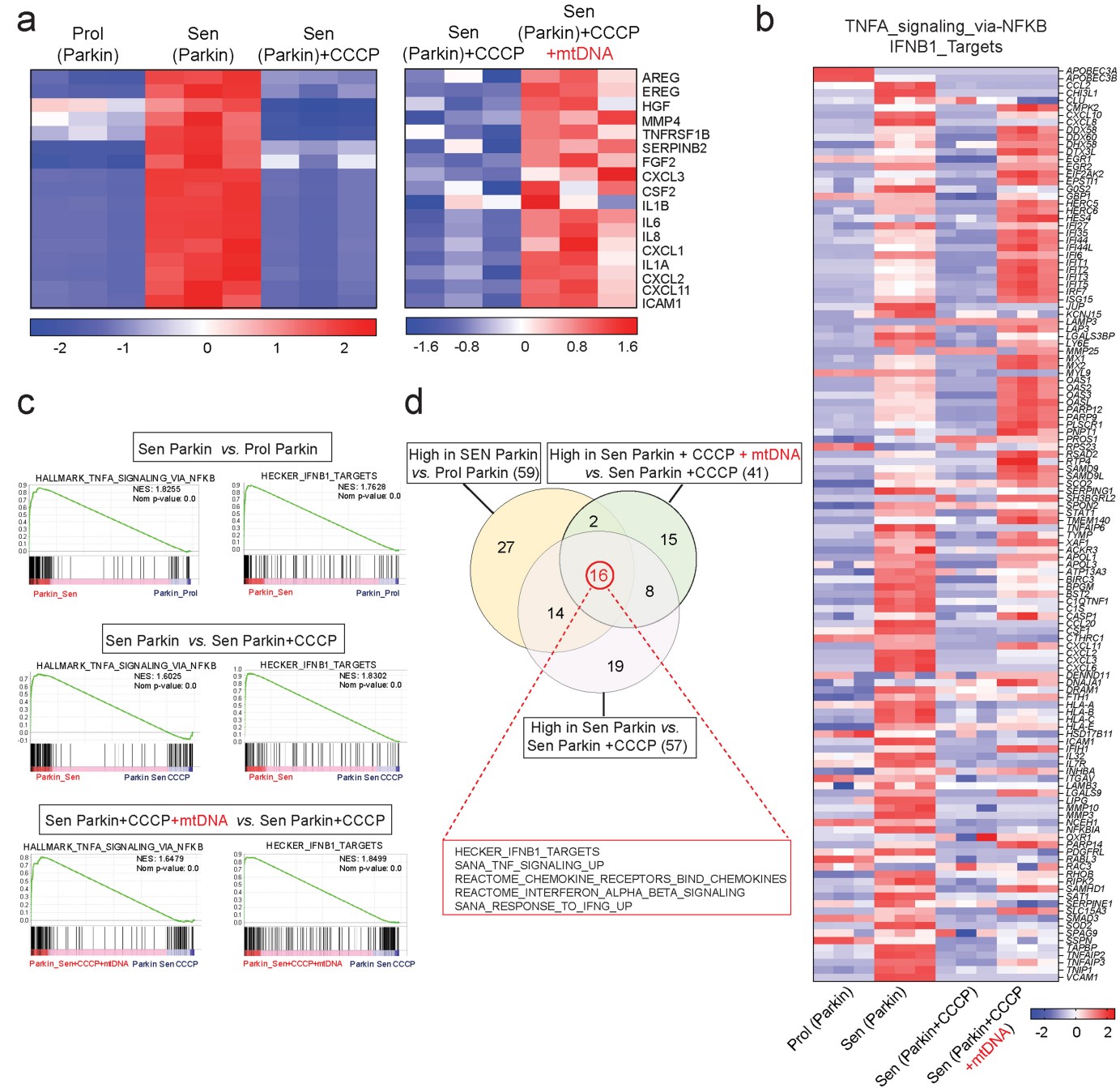

**Extended Data Fig. 5 | Widespread mitochondrial clearance suppresses inflammation during senescence, while reintroduction of mtDNA restored it.** (**a**) Column clustered heatmap of SASP genes that are differentially expressed in senescent IMR90 fibroblasts, down-regulated upon mitochondria clearance and are rescued by mtDNA transfection. The colour intensity represents column Z-score, where red and blue indicate high and low expression, respectively. (**b**) Gene members of two hallmark signatures, TNFA-signalling via NFKB (M5890) and Hecker IFNB1 targets (M3010), were upregulated in senescent cells and downregulated after CCCP. Subsequent addition of mtDNA rescued this phenotype. (**c**) The GSEA plots for Parkin Sen vs. Parkin Prol display an enrichment for the TNFA-signalling via NF-κB (M5890) and Hecker IFNB1 targets (M3010) pathways in the senescent cell population. Addition of mtDNA led to a significant enrichment compared to CCCP alone in senescent cells. (**d**) Venn diagram depicting a substantial overlap of enriched pathways in all three conditions (FWER p-value < 0.25).

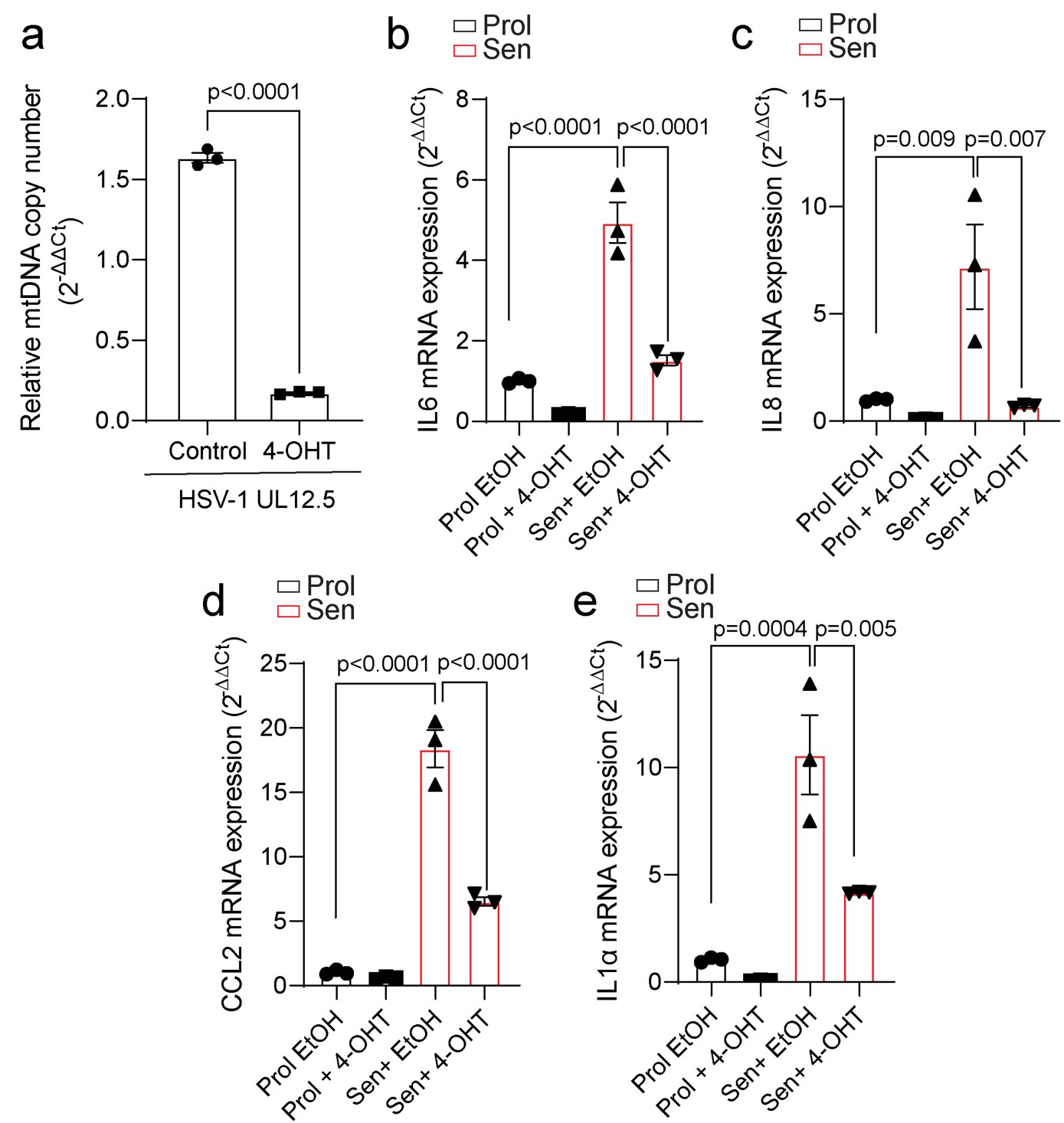

**Extended Data Fig. 6 | Expression of a tamoxifen-inducible viral DNase (HSV-1 UL12.5) that specifically targets mitochondria, results in depletion of mtDNA and suppresses the SASP. a)** Relative mtDNA copy number of MRC5-UL12.5 fibroblasts with and without tamoxifen treatment (100 nM, 48 h). **(b-e)** mRNA expression levels of SASP genes, IL6, IL8, Ccl2 and IL-1α in proliferating and senescent (IR) MRC5-UL12.5 fibroblasts with and without tamoxifen treatment (48 h). Data are mean ± S.E.M. of n = 3 independent experiments. Statistical significance was assessed using two-sided Student's unpaired t-test (a), one-way ANOVA followed by Tukey's multiple comparison test (b-e).

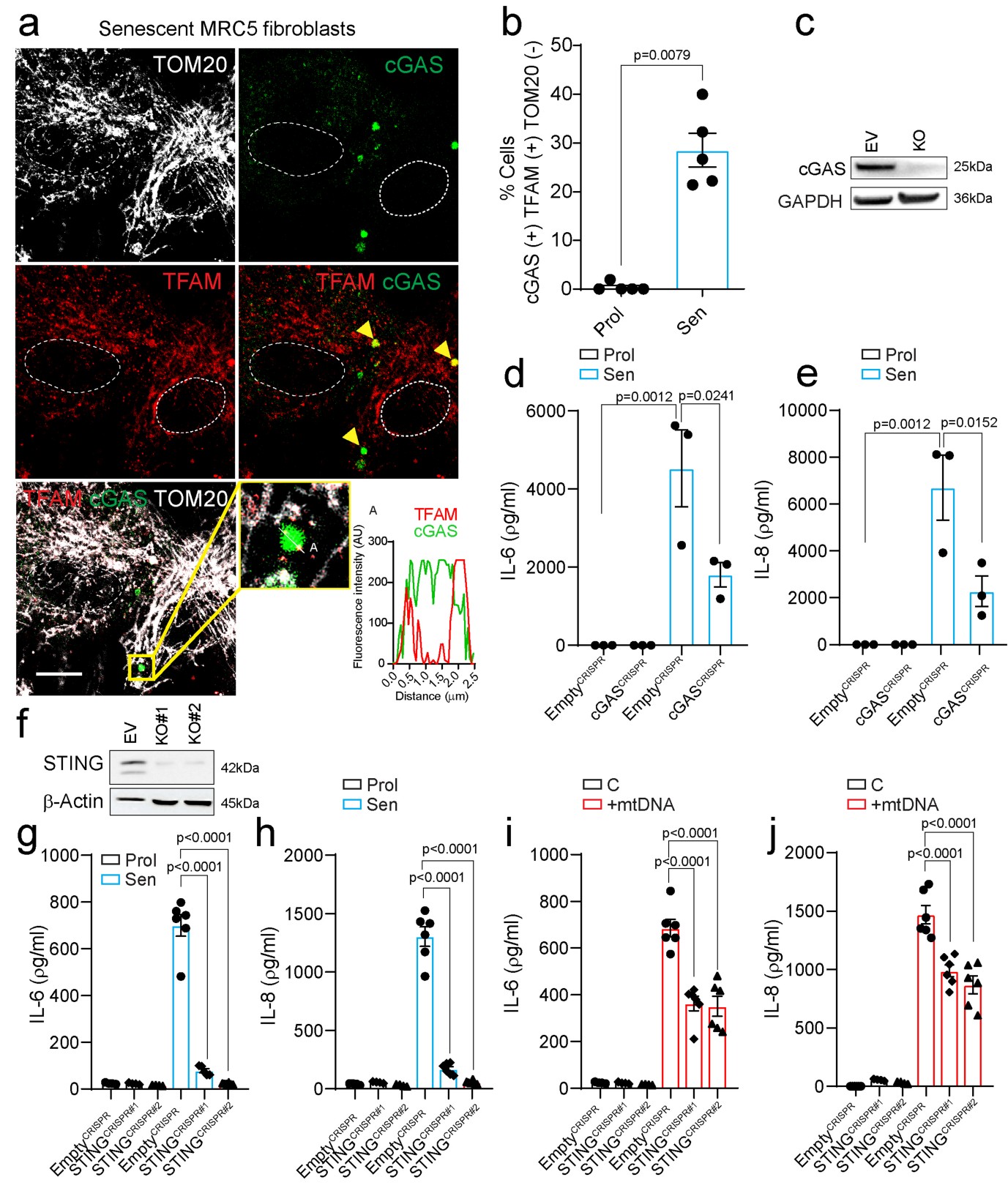

**Extended Data Fig. 7** | See next page for caption.

**Extended Data Fig. 7 | Cytosolic mtDNA acts via the cGAS–STING pathway to drive the SASP in senescent cells.** (**a**) Representative immunofluorescence image of cGAS–GFP fusion protein (green) co-localizing with TFAM (red) in senescent cells. Magnification at the bottom shows cGAS–GFP reporter colocalizing with TFAM foci in the absence of TOM20 (white) (scale bar is 20 µm). Graph represents quantification of cGAS and TFAM signals in selected linear region indicated in a. (**b**) Quantification of the percentage of proliferating and senescent cells (IR) containing cGAS co-localizing with cytosolic TFAM foci (n = 5 independent experiments). (**c**) Western blot showing the level of cGAS upon CRISPR/Cas9-mediated cGAS deletion. Representative blot of n = 1 independent experiment. Secreted levels of (**d**) IL-6 and (**e**) IL-8 in proliferating and senescent EV and cGAS-deficient MRC5 human fibroblasts (n = 3 independent experiments). (**f**) Representative Western blot showing the level of STING upon CRISPR/Cas9-mediated STING deletion using two different gRNAs. Blot is representative of n = 2 independent experiments. Secreted levels of (**g**) IL-6 and (**h**) IL-8 in proliferating (n = 4 independent experiments) and senescent (n = 6 independent experiments) EV and STING-deficient MRC5 human fibroblasts. Secreted levels of (**i**) IL-6 and (**j**) IL-8 in EV and STING-deficient MRC5 human fibroblasts upon mtDNA transfection (n = 4 for control, n = 6 independent experiments for +mtDNA condition). Data are mean ± S.E.M. Statistical significance was assessed using two-sided Student's unpaired t-test (b), one-way ANOVA followed by Tukey's multiple comparison test (d, e, g-j).

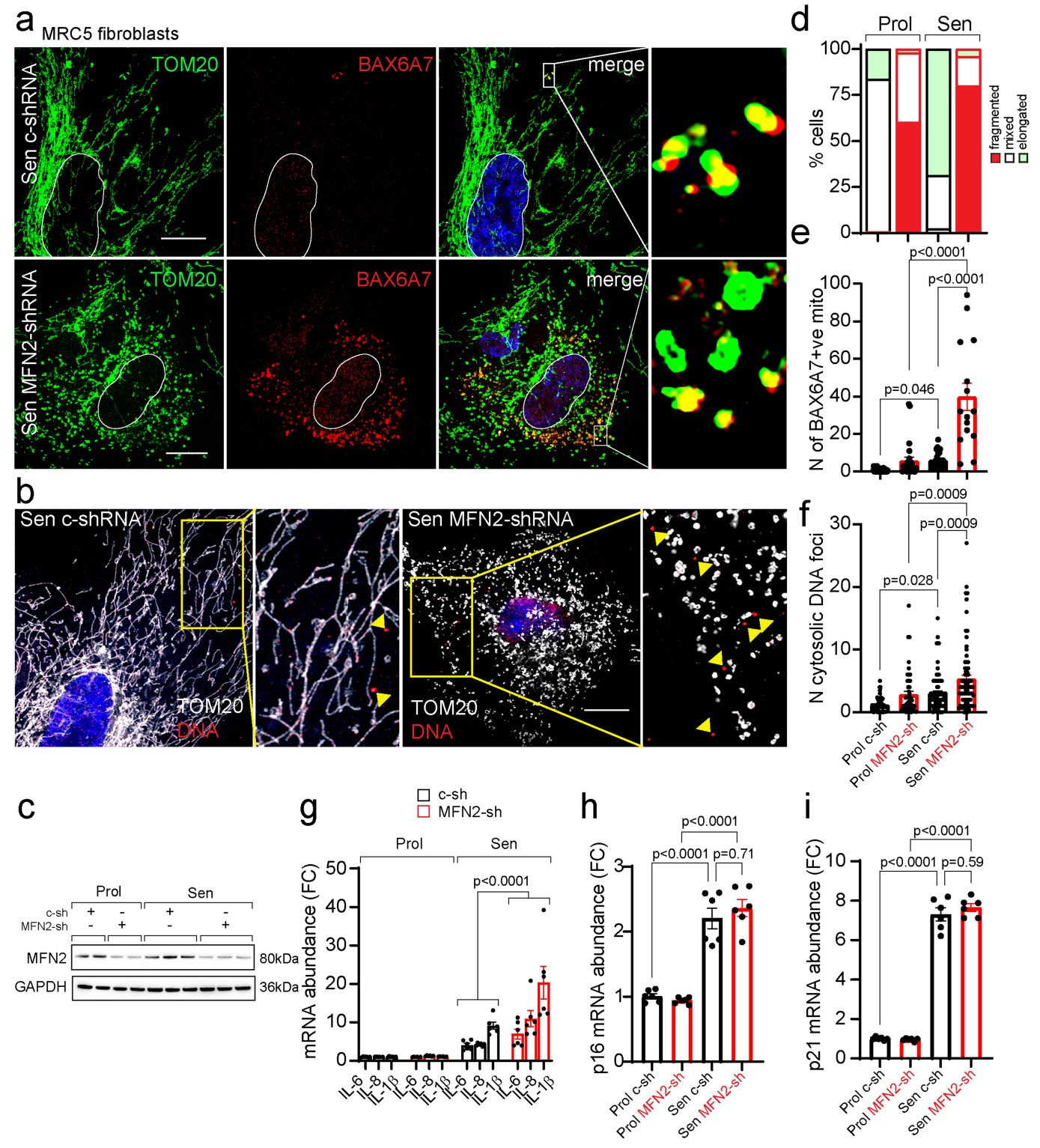

**Extended Data Fig. 8** | See next page for caption.

**Extended Data Fig. 8 | MFN2 deficiency exacerbates intracellular mtDNA release and the SASP in senescent MRC5 fibroblasts.** (**a**) Representative immunofluorescence images of TOM20 (green) and activated BAX (BAX6A7) (red) in senescent (IR) control (c-shRNA) and shRNA-mediated MFN2 knockout MRC5 fibroblasts (scale bar is 20 μm). Magnifications show BAX6A7 co-localizing with fragmented mitochondria. (**b**) Representative immunofluorescence images of TOM20 (white) and DNA (red) in control and shRNA-mediated MFN2 knockout MRC5 human fibroblasts (scale bar is 20 μm). Magnification shows DNA foci located outside of TOM20. Images are representative of n = 3 independent experiments (a, b). (c) Western blot showing the protein level of MFN2 following shRNA-mediated deletion of MFN2 in proliferating and senescent MRC5 fibroblasts. Quantification of (**d**) the percentage of cells containing fragmented, mixed, and elongated mitochondria (n = 3 independent experiments), (**e**) the number of BAX6A7- positive mitochondria in proliferating and senescent control and MFN2-shRNA MRC5 fibroblasts (n = 74 Prol, n = 26 Prol (sh-MFN2), n = 29 Sen (sh-C), n = 15 Sen (sh-MFN2) cells analysed over 2 independent experiments). Data are mean ± S.E.M. (**f**) Number of DNA foci located outside of TOM20 in proliferating and senescent control and MFN2-shRNA MRC5 fibroblasts. Data are mean of n = 3 independent experiments ± S.E.M. (n = 50 Prol, n = 49 Prol (sh-MFN2), n = 78 Sen (sh-C), n = 76 Sen (sh-MFN2) cells). Quantification of mRNA expression level of (**g**) indicated SASP factors, (**h**) p16INK4A, and (**i**) p21 in proliferating and senescent control- and MFN2- shRNA MRC5 fibroblasts. Data are mean of n = 6 independent experiments ± S.E.M. Statistical significance was assessed using one-way ANOVA followed by Tukey's multiple comparison test (e, f, h, i), two-way ANOVA followed by Tukey's multiple comparison test (g). For gel source data (c), see Supplementary Fig. 1.

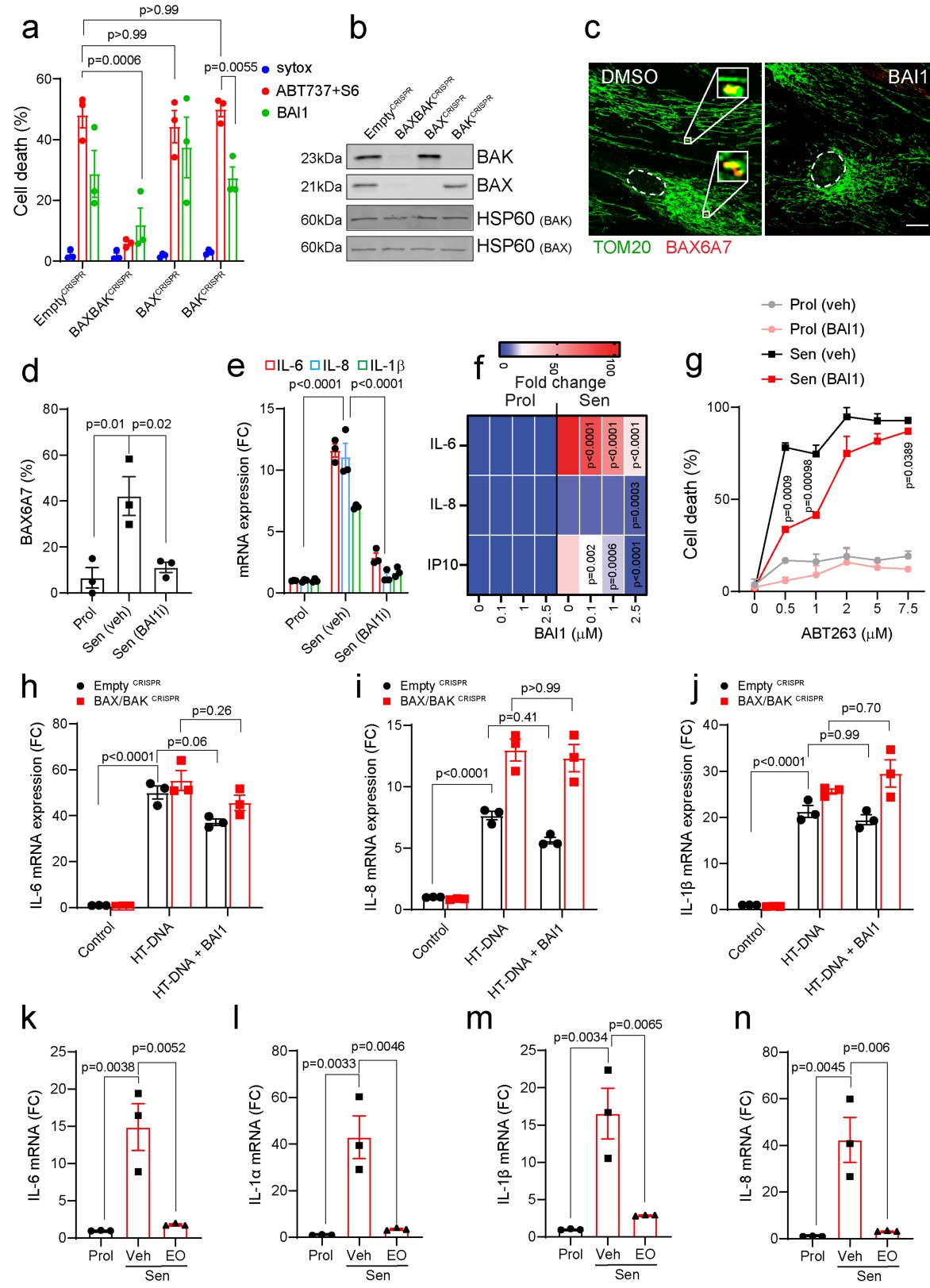

**Extended Data Fig. 9 |** See next page for caption.

**Extended Data Fig. 9 | BAI1 suppresses miMOMP and SASP in senescent cells and does not inhibit DNA-induced inflammation.** (**a**) Percentage cell death following ABT737 + S6 induced cell death at hour 8 post apoptosis induction in presence of vehicle control or BAI1. Data are representative of n = 3 independent experiments. (**b**) Western blot showing successful CRISPR/Cas9-mediated deletion of BAX/BAK and single BAX and BAK in U2OS cells. HSP60 was used as loading control. Image is representative of two separate blots. (**c**) Representative image of TOM20 (green) and BAX6A7 (red) in senescent (IR) MRC5 fibroblasts treated with DMSO or BAI1 (scale bar is 25 μm). (**d**) Percentage of MRC5 fibroblasts containing BAX6A7-positive mitochondria following BAI1 treatment (n = 3 independent experiments). (**e**) mRNA expression levels of the indicated SASP genes in BAI1-treated senescent MRC5 fibroblasts (n = 3 independent experiments). (**f**) Heatmap showing secreted levels of IL-6, IL-8 and IP-10 in proliferating and senescent MRC5 fibroblasts treated with different concentrations of BAI1. Values are shown as fold change to proliferating controls. Data are mean of n = 4 independent experiments. (**g**) Percentage death of vehicle- or BAI1- treated proliferating and senescent cells upon ABT263 treatment at the concentrations indicated. Data are mean of n = 3 technical replicates ± S.E.M. mRNA levels of (**h**) IL-6, (**i**) IL-8, and (**j**) IL-1β in control (EmptyCRISPR) and BAX/BAK-/- MRC5 fibroblasts treated with Herring testes DNA (HT-DNA) with or without BAI1 (n = 3 independent experiments). mRNA expression levels of (**k**) IL-6, (**l**) IL-1α, (**m**) IL-1β, and (**n**) IL-8 in senescent cells treated with eltrombopagan (EO) (n = 3 independent experiments). Data are mean ± S.E.M. Statistical significance was assessed using one-way ANOVA followed by Tukey's multiple comparison test (d, f, k-n), two-way ANOVA followed by Sidak's multiple comparisons test (h-j) or Tukey's multiple comparison test (a, e, g). For gel source data (b), see Supplementary Fig. 1.

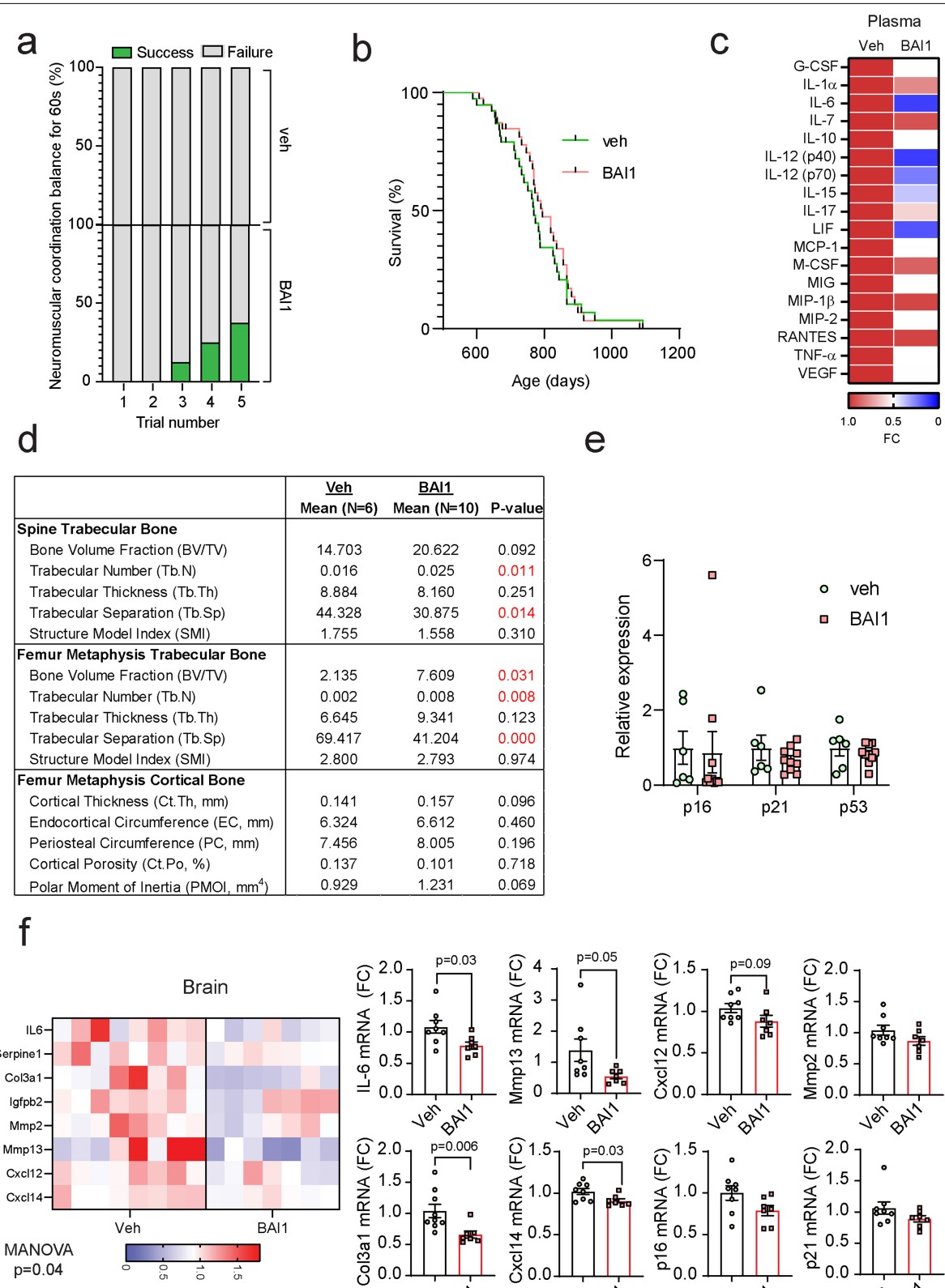

**Extended Data Fig.10 | Treatment with BAI1 improves healthspan and reduces inflammation but it does not affect lifespan of aged mice.** (**a**) Neuromuscular coordination shown as a percentage number of successful attempts (green) to remain on a straight rod for 60 s (n = 7 vehicle and n = 8 BAI1-treated mice). (**b**) Kaplan-Meier survival curves of animals treated with vehicle (n = 38) or Bax inhibitor (n = 39) from 18–20 months old until death. (**c**) Heatmap showing levels of cytokines found in plasma from mice treated with vehicle or BAI1. Values are shown as fold change compared to vehicle-treated animals. Red denotes high expression and blue indicates low expression.

**d**) Table summarizing μCT-derived parameters obtained for the spine and femur from vehicle (n = 6) and BAI1-treated mice (n = 10). **e**) mRNA expression of p16, p21 and p53 in control (n = 6) and BAI-1 treated mice (n = 10). **f**) (left) Heatmap showing mRNA expression of the indicated SASP factors in the brain from aged animals treated with BAI1 (p = 0.04). (right) Graphs showing quantification of levels of mRNA of the indicated genes in brains from aged mice treated with BAI1 (n = 8 vehicle- and n = 7 BAI1-treated mice). Data are mean ± S.E.M. Statistical significance was assessed using two-sided Student's unpaired t-test (d, f), MANOVA (heatmap in f).

# Reporting Summary

## Statistics

For all statistical analyses, confirm that the following items are present in the figure legend, table legend, main text, or Methods section.

| n/a | Confirmed | |
|---|---|---|
| ☐ | ☒ | The exact sample size (*n*) for each experimental group/condition, given as a discrete number and unit of measurement |
| ☐ | ☒ | A statement on whether measurements were taken from distinct samples or whether the same sample was measured repeatedly |
| ☐ | ☒ | The statistical test(s) used AND whether they are one- or two-sided<br>*Only common tests should be described solely by name; describe more complex techniques in the Methods section.* |
| ☐ | ☒ | A description of all covariates tested |
| ☐ | ☒ | A description of any assumptions or corrections, such as tests of normality and adjustment for multiple comparisons |
| ☐ | ☒ | A full description of the statistical parameters including central tendency (e.g. means) or other basic estimates (e.g. regression coefficient) AND variation (e.g. standard deviation) or associated estimates of uncertainty (e.g. confidence intervals) |
| ☐ | ☒ | For null hypothesis testing, the test statistic (e.g. *F*, *t*, *r*) with confidence intervals, effect sizes, degrees of freedom and *P* value noted<br>*Give P values as exact values whenever suitable.* |
| ☒ | ☐ | For Bayesian analysis, information on the choice of priors and Markov chain Monte Carlo settings |
| ☒ | ☐ | For hierarchical and complex designs, identification of the appropriate level for tests and full reporting of outcomes |
| ☒ | ☐ | Estimates of effect sizes (e.g. Cohen's *d*, Pearson's *r*), indicating how they were calculated |

*Our web collection on statistics for biologists contains articles on many of the points above.*

## Software and code

Policy information about availability of computer code

| | |
|---|---|
| Data collection | No software was used |
| Data analysis | GraphPad Prism v9 |

For manuscripts utilizing custom algorithms or software that are central to the research but not yet described in published literature, software must be made available to editors and reviewers. We strongly encourage code deposition in a community repository (e.g. GitHub). See the Nature Portfolio guidelines for submitting code & software for further information.

## Data

Policy information about availability of data

All manuscripts must include a data availability statement. This statement should provide the following information, where applicable:

- Accession codes, unique identifiers, or web links for publicly available datasets
- A description of any restrictions on data availability
- For clinical datasets or third party data, please ensure that the statement adheres to our policy

The RNA-seq datasets generated and analyzed during the current study are available in the GEO repository GSE196610 and GSE235225. The mass spectrometry proteomics data have been deposited to the ProteomeXchange Consortium via the PRIDE partner repository with the dataset identifier PXD040018.

# Field-specific reporting

Please select the one below that is the best fit for your research. If you are not sure, read the appropriate sections before making your selection.

☒ Life sciences  ☐ Behavioural & social sciences  ☐ Ecological, evolutionary & environmental sciences

For a reference copy of the document with all sections, see nature.com/documents/nr-reporting-summary-flat.pdf

# Life sciences study design

All studies must disclose on these points even when the disclosure is negative.

| | |
|---|---|
| Sample size | For in vivo mouse studies, we have projected group size estimates based on power analyses and previous experiences. |
| Data exclusions | no data was excluded |
| Replication | we tested our hypotheses using multiple approaches when feasible to enhance scientific rigor and reproducibility. in vitro all experiments were performed at least 3 times independently. |
| Randomization | Animals were randomly assigned numbers at weaning. Once assigned to groups, the genotype was not be linked to the numbers until data analysis following completion of all studies. |
| Blinding | Investigators were blinded to allocation during experiments and outcome assessments, and data was collected and analyzed in a blinded fashion. |

# Reporting for specific materials, systems and methods

We require information from authors about some types of materials, experimental systems and methods used in many studies. Here, indicate whether each material, system or method listed is relevant to your study. If you are not sure if a list item applies to your research, read the appropriate section before selecting a response.

### Materials & experimental systems

| n/a | Involved in the study |
|---|---|
| ☐ | ☒ Antibodies |
| ☐ | ☒ Eukaryotic cell lines |
| ☒ | ☐ Palaeontology and archaeology |
| ☐ | ☒ Animals and other organisms |
| ☒ | ☐ Human research participants |
| ☒ | ☐ Clinical data |
| ☒ | ☐ Dual use research of concern |

### Methods

| n/a | Involved in the study |
|---|---|
| ☒ | ☐ ChIP-seq |
| ☒ | ☐ Flow cytometry |
| ☒ | ☐ MRI-based neuroimaging |

## Antibodies

| | |
|---|---|
| Antibodies used | Anti-TOMM20 rabbit polyclonal antibody; Millipore Sigma; HPA011562<br>Anti-Cytochrome c mouse monoclonal antibody; BioLegend; 612301<br>Anti-Bax(6A7) mouse monoclonal antibody; Santa Cruz; sc-23959<br>Anti-Phospho-Histone H2A.X (Ser139) (20E3) rabbit monoclonal antibody; Cell Signaling; 9718<br>Anti-DNA mouse monoclonal antibody; Millipore Sigma; CBL186<br>Anti-TFAM rabbit monoclonal antibody; Cell Signaling; 8076S<br>Anti-p16 (E6H4) mouse monoclonal antibody; Roche; 705-4713<br>Anti p21 Waf1 Cip1 (12D1) rabbit monoclonal antibody; Cell Signaling; 2947S<br>Anti-Ki67 rabbit polyclonal antibody; Abcam; ab15580<br>Anti-Bax (D3R2M) rabbit monoclonal antibody; Cell Signaling; 14796<br>Anti-Cleaved Caspase-3 (Asp175) Antibody; Cell Signaling; 9661<br>Goat anti-Rabbit IgG (H+L), Superclonal™ Recombinant Secondary Antibody, Alexa Fluor 647; Thermo Fisher Scientific; A-27040<br>Goat anti-Rabbit IgG (H+L) Cross-Adsorbed Secondary Antibody, Alexa Fluor 488; Thermo Fisher Scientific; A-11008<br>Goat anti-Rabbit IgG (H+L) Cross-Adsorbed Secondary Antibody, Alexa Fluor 594; Thermo Fisher Scientific; A-11012<br>Goat anti-Mouse IgG (H+L) Highly Cross-Adsorbed Secondary Antibody, Alexa Fluor 594; Thermo Fisher Scientific; A-11032<br>Goat anti-Mouse IgG (H+L) Cross-Adsorbed Secondary Antibody, Alexa Fluor 647; Thermo Fisher Scientific; A-21235<br>Goat anti-Mouse IgG (H+L), Superclonal™ Recombinant Secondary Antibody, Alexa Fluor 488; Thermo Fisher Scientific; A-28175<br>Anti-cytochrome c (D18C7) rabbit monoclonal antibody; Cell Signaling; 11940S<br>Anti-UQCR2 rabbit monoclonal antibody; Abcam; ab103616<br>Anti-Cleaved Caspase 3 (D175 hAIE) rabbit monoclonal antibody; Cell Signaling; 9664S<br>Anti-β-actin mouse monoclonal antibody; Abcam; ab8226<br>Anti-TFAM rabbit monoclonal antibody; Cell Signaling; 8076S |

Anti-Bak (D4E4) rabbit monoclonal antibody; Cell Signaling; 12105
Anti-Bax (D2E11) rabbit monoclonal antibody; Cell Signaling; 5023
Anti-Bax (6A7) mouse monoclonal antibody; Santa Cruz; sc-23959
Anti- -Tubulin rabbit polyclonal antibody; Cell Signaling; 2146S
Anti-Lamin B1 rabbit polyclonal antibody; Abcam; ab16048
Anti-NDUFB8 mouse monoclonal antibody; Abcam; ab110242
Anti-GAPDH rabbit monoclonal antibody; Cell Signaling; 5174S
Anti-cGAS (D1D3G) rabbit monoclonal antibody; Cell Signaling; 15102
Anti-STING (D2P2F) rabbit monoclonal antibody; Cell Signaling; 13647S
Anti-APAF1 rabbit monoclonal antibody; Cell Signaling; 8723S
Anti-HMGB1 rabbit monoclonal antibody; Cell Signaling; 6893S
Goat Anti-Rabbit HRP Conjugated; Sigma Aldrich; A0545
Goat Anti-Mouse HRP Conjugated; Sigma Aldrich; A2554

Validation

All antibodies used were purchased from commercial sources and reputable vendors (eg. Abcam, Cell Signaling). We selected them based on cross-reactivity with mouse and human (depending on our research needs) and carefully evaluated the validation data provided by both the vendors and literature. In several instances, we performed further validation of antibodies by examining the molecular weight of the band by Western blotting and Knocking out protein of interest. Antibodies were aliquoted and stored as recommended by the manufacturer to minimize freeze thaw cycles and associated loss in performance.

# Eukaryotic cell lines

Policy information about cell lines

Cell line source(s)

IMR90 and MRC5 human fibroblasts were acquired from ATCC

Authentication

none of the cells have been authenticated

Mycoplasma contamination

all cell lines used have been regularly tested for Mycoplasm

Commonly misidentified lines
(See ICLAC register)

N/A

# Animals and other organisms

Policy information about studies involving animals; ARRIVE guidelines recommended for reporting animal research

Laboratory animals

Information regarding experiments involving laboratory mice is in methods section.

Wild animals

N/A

Field-collected samples

N/A

Ethics oversight

All animal experiments were performed according to protocols approved by the Institutional Animal Care and Use Committee (IACUC) at Mayo Clinic.

Note that full information on the approval of the study protocol must also be provided in the manuscript.

