## [Peer Review File · Nature]

Manuscript Title: Apoptotic stress causes mtDNA release during senescence and drives the SASP

Reviewer Comments & Author Rebuttals

Reviewer Reports on the Initial Version:

Referees' comments:

Referee #1 (Remarks to the Author):

This manuscript reports that senescence induced by low dose radiation, ER-RAS, and replicative aging, induces a minimal outer membrane permeabilization of the mitochondria, which then releases mtDNA into the cytosol and activates inflammation via the cGAS STING pathway. This is an interesting extension of prior reports from this group and others that mild induction of pro-apoptotic stimuli induces Bax to release mtDNA into the cytosol and induce cGAS STING activation. The authors further show in some experiments, but not all, that senescence itself does not depend on mtDNA release into the cytosol, but the SACP signature of inflammation does. How senescence mediates miMOMP is a key missing issue. Animal studies are thought by the authors to indicate that pharmacological inhibition of Bax in vivo inhibits inflammation, bone volume and several motor phenotypes.

Major points:

1) Much of the data is repetitive of the group's prior EMBO J. paper and other prior literature. In Figures 1, 2, 3 and 5 the only real novelty is that senescence can induce what has been found for other stresses. What is missing is identifying some novel mechanism into how senescence induces Bax to form pores in a subset of mitochondria and how those select mitochondria are denoted for miMOMP.

2) What senescent feature induces miMOMP and what constrains cytosolic DNA foci to a handful of miMOMPs per cell? Can one titrate the duration of senescence upwards to increase the number of DNA foci per cell? At what point does miMOMP shift to more MOMP and apoptosis? Or is there some mechanistic set point that maintains the miMOMP level at a few mitochondria per cell and thus avoiding apoptosis and also avoiding no MOMP and thus no SACP. Some insight into how that minimal level of MOMP is maintained/restrained or regulated is needed.

3) In Figure 4 it is unclear why the authors targeted liver specific viral Cre in Bax floxed and Bak -/- mice using immunostaining to semiquantitatively assess the degree of Bax knock out. Why not more simply and rigorously use Bak and BaxBak knock out mice? Most importantly, how do wild type mice cytokine profiles compare? Does loss of Bak and Bax decrease cytokine expression relative to wild type mice? Maybe the increase in caspase positive cells seen in Bak deficient cells in Figure 4f would be much higher in wild type mice indicative of apoptosis and also indicating apoptosis occurred in Bak deficient cells just below the detectability range using immunostaining – but not below the level of detection of the more sensitive qPCR detection of cytokine expression downstream of apoptosis.

4) In Figure 5 Parkin experiments assess the role of mitochondria in senescence and SACP. However, these are not clean experiments as the authors seek to assess the role of mitochondrial DNA not the pleiotropic roles of mitochondria in SACP (transfecting mtDNA or almost any DNA is already known to activate STING to induce cytokines so not very informative). Cells completely lacking mitochondria are deficient in several necessary pathways such as iron sulfur cluster

formation. Iron sulfur containing enzymes are needed for nuclear DNA repair so cells lacking mitochondria may have excessive nuclear DNA damage following irradiation and induce more senescence (which was seen) unrelated to mtDNA. Cells completely lacking mitochondria do not survive long term regardless of irradiation and as shown here by the authors have accelerated senescence induction. The key experiment is to use true Rho0 cells, that do survive and do make iron sulfur clusters but lack mtDNA. This is touched on in Figure 5 o,p. However, substantial IL-8 is still induced in the Rho0 cells arguing against mtDNA as the essential activator. Some recent reports use short term mtDNA "depletion" with ethidium bromide (and perhaps in this report – it is not stated how the Rho0 cells were made or where they were obtained or how Rho0 status was assessed). Full Rho0 cells stably negative for mtDNA need to be used and this may resolve the problem of substantial IL-8 and some IL-6 still induced by senescence in Rho0 cells. That full mtDNA depletion is achieved after many passages of Rho0 cells after release from ethidium bromide is needed to assess full Rho0 status by qPCR. If the authors did use true Rho0 cells – something other than mtDNA is inducing SACP during senescence – and this does not match the full depletion of SACP in, for example, the Bak/Bax deficient mice (Fig. 4) – unless SACP is induced by apoptosis or some other process. In this important experiment (Fig. 5 o,p) how was senescence induced and how was it's induction assessed? Is mtDNA required for senescence? Presumably not if these cells are true Rho0 and the cells become senescent. But this contrasts with the implication of Suppl. Fig. 1 which seems to link a correlation between miMOMP and senescence induction in the title and in Supplemental Figure 5b – see below.

5) TFAM heterozygous MEFs were previously reported to release mtDNA and activate STING. Here the authors show they reach replicative senescence sooner than WT MEFs. Is this related to mtDNA release or STING activation? If so, this does not match Figure 3e-l where the authors make the point that mtDNA release inhibits SACP, but not senescence itself. In order to yield some interpretation of Supplemental Figure 5b it would be important to either inhibit STING pharmacologically or knock out STING in these TFAM het cells to see if this reverses the more rapid senescence phenotype. If miMOMP drives senescence then how do Rho0 cells undergo senescence (Fig. 5o,p)? It would be important to compare the rate of senescence in Rho0 and the matched control cells following senescence induction as used in Fig. 5o,p.

6) The Bax pore is now known to be a large and nebulous arc or disk ranging in size. Use of a poorly substantiated compound reported in 2003 (BCB) to inhibit Bax in vivo is not sufficient evidence for the authors claims. There is no evidence presented that BCB is inhibiting inflammation via inhibiting Bax pores as opposed to a myriad of off target effects. Does BCB inhibit inflammation in Bax KO mice? Most important, does Bcl-xL overexpression prevent SACP in vivo?

Minor points:

1) During apoptosis following MOMP cytochrome c is released from mitochondria and is faintly and evenly dispersed in the cytosol. What are the bright foci of cytochrome c in Fig 1A? Why are these foci not yellow as in most of the perinuclear mitochondria? It is hard to see any green staining in the high mag images that does not have some red staining too. Maybe these bright foci of cytochrome c artificially make the other red cytochrome c within the green mitochondria more difficult to visualize. Please show high mag examples of cytochrome c release where bright foci of cytochrome c are not present as the current images are not convincing evidence of miMOMP.

2) Please show representative images for the counting of cGAS TFAM colocalization in Fig. 5g. Is Fig. 5g representing Pearson's colocalization index? If not, please add this.

3) Figure 5m,n could be deleted to save space because this is already well known.

4) It is stated on page 6 that ABT-737 "does not induce any cell-death ..." citing Supplemental Figure 1a. This is not shown in Supplemental Figure 1a or anywhere in Supplemental Figure 1 – and needs to be added.

Referee #2 (Remarks to the Author):

In this interesting paper, Chapman et al. examine the role of mitochondria in the phenomenon of senescence-associated secretory phenotype (SASP). They find that limited mitochondrial outer-membrane permeabilization (MOMP), dependent on Bax and Bak, occurs in senescent cells, and that the resultant release of mtDNA and cGAS/STING activation is required for the SASP. In vivo, they further show that Bax and Bak in the liver are required for the induction of SASP in a model of liver senescence. These experiments and results, and their interpretation are overall very solid and convincing. They go on to demonstrate that a putative Bax inhibitor prevents SASP in vitro and has striking neuroprotective effects in age-association neurological decline. While these data are compelling, I have some issues with these experiments, as described below.

1. The authors use a piperazine derivative of 2-propanol as a "Bax inhibitor," based on a paper from 2003. Whether this compound is indeed specific for Bax or for MOMP is by no means known, and indeed, there is evidence that propanol derivatives can be neuroprotective in other settings (although mechanism and specificity are an issue here as well). Since the effects they show are interesting, I suggest that some effort be made to determine if these effects are likely to be on target. First, we need to know if this is indeed Bax specific, which can be readily done using cells that are deficient in Bax or Bak, induced to undergo apoptosis (it should be noted that should the compound block death in both settings, this is fine, but does suggest that it should not be referred to as a "Bax inhibitor."

2. Secondly, it would be important to know if the inhibitor does indeed prevent miMOMP in senescent cells. Note that since the inhibitor blocks SASP in cells that are already senescent (apparently—as suggested in the legend of Figure 6b) this would suggest that continued release of mtDNA is necessary for the SASP. If the compound is added prior to induction of senescence (in 6a and 6b) this should be noted. In any case, we need to know if the inhibitor blocks miMOMP, as suggested by the experiment in 6a.

3. Finally, it is critical to determine if the inhibitor blocks the production/secretion of SASP cytokines induced by means other than senescence (e.g., TLR engagement). If so, this would call into question the interpretation of their results.

4. The authors should discuss the findings on propanol derivatives as neuroprotective agents in settings that might be unlikely to relate to senescence. At least, the possibility that their inhibitor is acting in some other way than via the pathway they have found should be presented as a possibility.

5. On a different note, the experiments on the induction of senescence using low dose BH3 mimetics is interesting, as is the associated gamma H2AX staining. It is possible that this phenotype is dependent on caspase-activated nuclease, which they can easily test in their APAF1-deficient cells (which undergo senescence and SASP). I do not think that testing this idea is essential for the conclusions of the paper, but would be interesting to specialists. I suggest that this experiment (induction of g-H2AX by BH3 mimetics in the presence or absence of APAF1) is optional.

Referee #3 (Remarks to the Author):

In this work, Chapman et al. report that senescent cells display minority MOMP in a subset of mitochondria generating Bax/Bak pores, mtDNA release, and inflammatory cytokine production through cGAS-STING activation. The authors show the importance of Bax/Bak for inflammatory

cytokine production in a model of irradiation-induced stress in vivo and report improvements in physical function and bone homeostasis in aged mice. These findings provide novel information on how senescent cells regulate their inflammatory phenotypes and will be of interest to readers from the field of aging, senescence, and innate immunity. I am generally positive about the manuscript but would like to see additional data to support the generalization of the findings in other settings of senescence and thorough documentation of reduced inflammation in aged mice receiving Bak inhibitor treatments.

Major:

1) Mitochondrial dysfunction in senescence and aging is well-documented, but a precise molecular understanding of how this connects to inflammation is missing. The primary conclusion from this work is that minority MOMP is responsible for mtDNA release and senescence-induced cytokine secretion via the cGAS-STING pathway.

There are many ways to trigger senescence in vitro. It would add to the manuscript to demonstrate Bax/Bak-dependent in chemotherapy-induced senescence - a medically relevant context of senescence.

2) The authors show multiple independent data to document health improvements in aged mice treated with a BAK inhibitor. According to their model, this outcome should strictly depend on decreased markers of inflammation, an aspect that appears underdeveloped in the current manuscript. Ideally, I would like to see an assessment of inflammatory cytokines in another organ than bone.

Minor:

1) Prior work has reported that TFAM deficiency induces senescence, triggers inflammaging, and accelerates features of aging-associated decline in mice (PMID: 32439659). This publication should be cited in this context.

2) References 5 and 24 are identical.

3) The authors should measure BAX and BAK protein levels in senescent cells versus naïve cells.

4) Related to Supplementary Fig. 1: One might have expected that the treatment of cells with a miMOMP inducer would trigger an immediate immune response by releasing mtDNA. Are type I IFN response genes also upregulated in a delayed fashion?

Author Rebuttals to Initial Comments:

We thank the reviewers for the constructive comments, and we believe the manuscript is greatly improved as a result of addressing them.

We aimed to respond to all the comments by conducting additional experiments. Our revised manuscript has now 6 main figures and 16 additional supplementary datasets, which not only support our central hypothesis but also provide additional mechanistic insights into how sub-lethal apoptotic stress is regulated in senescent cells.

Referee #1 (Remarks to the Author):

This manuscript reports that senescence induced by low dose radiation, ER-RAS, and replicative aging, induces a minimal outer membrane permeabilization of the mitochondria, which then releases mtDNA into the cytosol and activates inflammation via the cGAS STING pathway. This is an interesting extension of prior reports from this group and others that mild induction of pro-apoptotic stimuli induces Bax to release mtDNA into the cytosol and induce cGAS STING activation. The authors further show in some experiments, but not all, that senescence itself does not depend on mtDNA release into the cytosol, but the SACP signature of inflammation does. How senescence mediates miMOMP is a key missing issue. Animal studies are thought by the authors to indicate that pharmacological inhibition of Bax in vivo inhibits inflammation, bone volume and several motor phenotypes.

Major points:

1) Much of the data is repetitive of the group's prior EMBO J. paper and other prior literature. In Figures 1, 2, 3 and 5 the only real novelty is that senescence can induce what has been found for other stresses. What is missing is identifying some novel mechanism into how senescence induces Bax to form pores in a subset of mitochondria and how those select mitochondria are denoted for miMOMP.

Response: We appreciate the reviewer's comment. The textbook view of apoptosis and senescence is that they are distinct processes, often engaged in response to similar stressors. *Our study describes a direct connection between apoptotic signaling and senescence, namely that minority MOMP is a key promotor of the pro-inflammatory senescence associated secretory phenotype (SASP) – we emphasize this key point in the revised manuscript.*

As the reviewer highlights, in our initial submission what regulates minority MOMP in senescent cells wasn't clear. To investigate this, in new experiments we isolated activated BAX (using 6A7 mAb against activated BAX) from proliferating and senescent cells followed by mass spec analysis. Consistent with our dataset, activated BAX was only isolated from senescent cells. Importantly, proteins involved in mitochondrial dynamics were found to co-associate with activated BAX (**Extended Data Figure 10**). In new data, we also noted that mitochondria displaying minority MOMP were fragmented (**Extended Data Figure 11**). From these findings, we investigated a role for mitochondrial dynamics in regulating minority MOMP induced SASP in senescent cells. We found that enforced mitochondrial fission (MFN-2 CRISPR/CAS9, shRNA-mediated knockout or CCCP treatment), promotes minority MOMP leading to leakage of mtDNA and the SASP. Mitochondrial dysfunction has been reported by us and others to occur in senescent cells¹⁻³, and in a recent study⁴ we found in cancer cells that mitochondrial dysfunction serves as a mitochondrial-intrinsic signal to promote minority MOMP, both by recruiting BAX and segregating dysfunctional mitochondria. This supports our new data describing *a key role for mitochondrial dynamics and dysfunction in regulating minority MOMP and the SASP in senescent cells* (see scheme in **Extended Data Figure 14 e**).

2) What senescent feature induces miMOMP and what constrains cytosolic DNA foci to a handful of miMOMPs *per cell*?

Response: An important driver of minority MOMP in senescent cells (alongside mitochondrial dynamics and dysfunction) is most likely increased apoptotic priming associated with senescence. We have previously shown that increased apoptotic priming promotes minority MOMP. In line with previous reports^{5,6}, in new data we find that senescent cells are more sensitive to BH3-mimetic treatment than their proliferating counterparts (**Extended Data Figure 15g**) – consistent with increased apoptotic priming. Stemming from our earlier comment, what constrains minority MOMP is likely a combination of factors: 1) fragmentation/fission of mitochondria that are primed to undergo minority MOMP (consistent with our new data), thus limiting MOMP to individual mitochondria 2) intrinsic mitochondrial dysfunction signals minority MOMP on individual mitochondria, consistent with findings from us (and others) of mitochondrial dysfunction during senescence^{1-3,7} and that enforced mitochondrial dysfunction enhances the SASP.

Can one titrate the duration of senescence upwards to increase the number of DNA foci per cell? At what point does miMOMP shift to more MOMP and apoptosis? Or is there some mechanistic set point that maintains the miMOMP level at a few mitochondria per cell and thus avoiding apoptosis and also avoiding no MOMP and thus no SACP. Some insight into how that minimal level of MOMP is maintained/restrained or regulated is needed.

Response: We see no correlation between the duration of senescence and extent of miMOMP. In new experiments, we have analyzed kinetically the presence of cytosolic DNA foci after senescence induction by X-ray irradiation (IR). We found that cytosolic leakage of mtDNA reaches a peak at 3 days after IR and remains stable up to 10 days (see reviewer **Figure 2**). Similarly, we start observing BAX6A7 positive mitochondria between 2-3 days after IR (which coincides with the presence of cytosolic DNA foci), however this number does not change up to 10 days.

Senescence is an extremely stable phenotype as demonstrated in human fibroblasts and other cell-types- from our own experience and that of others, one can keep senescent fibroblasts alive, (provided we refresh their media) almost indefinitely without considerable cell-death. In our lab, we have kept senescent cells in culture for up to 2 years without signs of cell-death (unpublished observation) and other earlier reports have revealed that senescent cells can be kept for months and years in culture without cell-death^{8,9}.

The reviewer raises an important question, when does miMOMP switch to extensive MOMP and cell death? Our data (and that of others) argues that miMOMP does not progress to extensive MOMP/cell death based on a failure to observe cells displaying miMOMP undergoing cell death even after prolonged or increased levels of sub-lethal stress and absence of a gradual increase in MOMP (it either appears limited <5% of mitochondria, or extensive/complete >95% of mitochondria). Emphasising an earlier point, mitochondrial intrinsic apoptotic priming promotes MOMP only on dysfunctional mitochondria. Finally, activated BAX has been shown to prevent mitochondrial fusion, this will serve as an additional mechanism to prevent propagation of minority MOMP¹⁰.

Reviewer Figure 1 - Representative images of TOM20 DNA in MRC5 fibroblasts at different time points after 20 Gy X-ray irradiation (IR). Quantification of extramitochondrial DNA foci at different time points after IR.

3) In Figure 4 it is unclear why the authors targeted liver specific viral Cre in Bax floxed and Bak $-/-$ mice using immunostaining to semiquantitatively assess the degree of Bax knock out. Why not more simply and rigorously use Bak and BaxBak knock out mice?

Response: We took a conditional (liver-specific) approach to delete BAX in BAK null animals for two reasons: i) to determine the effect of BAX deletion was hepatocyte-autonomous, given that we and others have detected senescent markers in these cells ¹¹⁻¹³; 2) whole-body deletion of BAX and BAK is most often embryonic lethal ¹⁴. In the revised version we outline this rationale. In new expts, we also injected AAV9-CAG-iCre to delete BAX in aged mice (20 months). In this case, we analyzed both liver and bone where we observed a significant reduction in BAX expression. In both organs, we saw that deficiency in BAX and BAK significantly reduced several SASP factors known to increase during aging (**Figure 4** and **Extended Data Figure 6**).

Most importantly, how do wild type mice cytokine profiles compare? Does loss of Bak and Bax decrease cytokine expression relative to wild type mice?

Response: In new data, we compared expression of SASP factors between age-matched wild-type, $Bak^{-/-}Bax^{fl/fl}$ and $BakBax^{-/-}$. Using two-way ANOVA (Tukey's multiple comparisons test), we found no significant differences between aged wild-type and $Bak^{-/-}Bax^{fl/fl}$ ($P=0.9790$) but found a significant decrease between aged wt and $BakBax^{-/-}$ ($p=0.0025$) and between aged $Bak^{-/-}Bax^{fl/fl}$ and $BakBax^{-/-}$ ($p=0.0086$) (**Figure 4e**).

Maybe the increase in caspase positive cells seen in Bak deficient cells in Figure 4f would be much higher in wild type mice indicative of apoptosis and also indicating apoptosis occurred in Bak deficient cells just below the detectability range using immunostaining – but not below the level of detection of the more sensitive qPCR detection of cytokine expression downstream of apoptosis.

Response: We agree with the reviewer that it is a possibility we should consider that the attenuation of inflammation we observe in $BaxBak^{-/-}$ mice could be a consequence of reduced apoptosis *in vivo*, particularly if immunostaining is not a sensitive enough method. While we do not discard the possibility (and note this possibility in the revised version), we found that

when we analyze the aging liver, we and others have found evidence for increased expression of SASP factors (not shown) and increased senescence-associated markers such as p21, p16^{Ink4a}, telomere-associated foci (TAF) and Senescence-associated distension of satellites (SADS) (see reviewer **Figure 2**). % of TAF and SADS-positive cells reaches around 15% in aged mice and is mostly observed in hepatocytes. Analysing with an additional apoptotic marker (TUNEL staining) in young and aged mice, we find low frequencies of TUNEL positive cells (below 0.4% and not in hepatocytes) and no age-dependent increase, consistent with our earlier data. Additionally, we and others have found that genetic removal of p16^{Ink4a} senescent positive cells or treatment with senolytic drugs reduces inflammation in the liver^{11,12} (consistent with the notion that senescent cells via the SASP are drivers of age-dependent inflammation).

Reviewer Figure 2 – % of senescence-associated markers a) Telomere-associated foci (TAF); b) Senescence-associated distension of satellites (SADS) in 3,12-,15- and 24.5-month-old mouse liver; mRNA abundance of c) p21 and d) p16^{Ink4a} in 3- and 20-month-old mouse liver; % of TUNEL positive cells in 3 and 20 month old mouse liver.

4) In Figure 5 Parkin experiments assess the role of mitochondria in senescence and SACP. However, these are not clean experiments as the authors seek to assess the role of mitochondrial DNA not the pleiotropic roles of mitochondria in SACP (transfecting mtDNA or almost any DNA is already known to activate STING to induce cytokines so not very informative).

Response: We agree with the reviewer for the reasons stated that this expt. in itself is insufficient to conclude a role for mtDNA in the SASP. As suggested by the reviewer, this is complemented by additional expts. using rho zero cells.

Cells completely lacking mitochondria are deficient in several necessary pathways such as iron sulfur cluster formation. Iron sulfur containing enzymes are needed for nuclear DNA repair so cells lacking mitochondria may have excessive nuclear DNA damage following irradiation and induce more senescence (which was seen) unrelated to mtDNA. Cells completely lacking mitochondria do not survive long term regardless of irradiation and as shown here by the authors have accelerated senescence induction. The key experiment is to use true Rho0 cells, that do survive and do make iron sulfur clusters but lack mtDNA. This is touched on in Figure 5 o,p. However, substantial IL-8 is still induced in the Rho0 cells arguing against mtDNA as the essential activator. Some recent reports use short term mtDNA “depletion” with ethidium bromide (and perhaps in this report – it is not stated how the Rho0 cells were made or where they were obtained or how Rho0 status was assessed). Full Rho0 cells stably negative for mtDNA need to be used and this may resolve the problem of substantial IL-8 and some IL-6 still induced by senescence in Rho0 cells. That full mtDNA depletion is achieved

after many passages of Rho0 cells after release from ethidium bromide is needed to assess full Rho0 status by qPCR.

Response: We thank the reviewer for raising this point and apologise for incomplete description of the Rho zero cells used here, we used 143B osteosarcoma Rho zero cells previously published¹⁵. In new data., we confirm that these cells are indeed rho zero by qPCR for mtDNA (**Figure 5g**). We should also clarify that the Parkin-mediated clearance model is extremely stable. It is true that in cancer and immortalized cells, cells without mitochondria can survive only for a relatively short period of times¹⁶. However, we and other groups have performed Parkin-mediated mitochondrial clearance in different primary cells and can keep them alive for at least 30 days after treatment with CCCP^{1,7,17}. The reviewer is correct that cells without mitochondria can still become senescent with increased expression of cyclin-dependent kinase inhibitors p21 and p16^{INK4A}, however, they do not express a SASP as shown by us and independently by Peter Adams' lab⁷.

If the authors did use true Rho0 cells – something other than mtDNA is inducing SACP during senescence – and this does not match the full depletion of SACP in, for example, the Bak/Bax deficient mice (Fig. 4) – unless SACP is induced by apoptosis or some other process. In this important experiment (Fig. 5 o,p) how was senescence induced and how was it's induction assessed? Is mtDNA required for senescence? Presumably not if these cells are true Rho0 and the cells become senescent. But this contrasts with the implication of Suppl. Fig. 1 which seems to link a correlation between miMOMP and senescence induction in the title and in Supplemental Figure 5b – see below.

Response: In the setting of Rho zero cells, the reviewer rightly notes continued IL-8 expression independent of mtDNA in senescent cells. Potentially, two main reasons underlie this, minority MOMP can activate inflammatory signalling independent of mtDNA for instance via NF-kB signaling¹⁸ and/or there are additional mechanisms driving the SASP (for instance CCFs which can drive senescence via cGAS-STING reported by others^{7,19}). Our data also shows that rho(0) cells despite a reduced SASP, show a senescent-growth arrest characterized by increased senescence-associated markers p16^{INK4A} and p21. Interestingly, we find that p21 induction is stronger in rho(0)s than parentals (**Reviewer Figure 3**). This is consistent with other reports showing that that rho(0) cells undergo a senescent-associated cell cycle arrest with increased expression of CDKis after exposure to different stressors²⁰.

Reviewer Figure 3 - Both parental and rho(0) cells show increased mRNA expression of p21 and p16^{INK4A} when senescence is induced by x-ray irradiation. Induction of p21 is stronger in rho(0) cells.

We discuss these possibilities in the revised manuscript. minority MOMP itself is not required for senescence induction, demonstrated by induction of senescence in BAX/BAK deleted cells, we apologise for the confusion here and have clarified this in the revised version. The induction of senescence by low-dose BH3-mimetic treatment likely relates to minority MOMP

caspase-dependent DNA-damage initiating senescence that we and others have reported^{21,22}.

Altogether, our data does not indicate that miMOMP and cytosolic mtDNA leakage are key determinants of the SASP but not of senescence arrest. Uncoupling the SASP from the cell-cycle arrest has been a major goal of the senescence field. Given that a chronic SASP is a driver of age-related pathology- senomorphic therapies aim to inhibit the SASP, while preserving the tumor suppressive component of senescence – our data argue that targeting miMOMP induced SASP may be one route to achieve this goal.

5) TFAM heterozygous MEFs were previously reported to release mtDNA and activate STING. Here the authors show they reach replicative senescence sooner than WT MEFs. Is this related to mtDNA release or STING activation? If so, this does not match Figure 3e-l where the authors make the point that mtDNA release inhibits SACP, but not senescence itself. In order to yield some interpretation of Supplemental Figure 5b it would be important to either inhibit STING pharmacologically or knock out STING in these TFAM het cells to see if this reverses the more rapid senescence phenotype. If miMOMP drives senescence then how do Rho0 cells undergo senescence (Fig. 5o,p)? It would be important to compare the rate of senescence in Rho0 and the matched control cells following senescence induction as used in Fig. 5o,p.

Response: We thank the reviewer for suggesting a key experiment that further supports these conclusions. As suggested, we treated TFAM^{+/-} MEFs with STING inhibitor SN011²³. We found that SN011 did not prevent the senescence-associated growth arrest in TFAM^{+/-} MEFs, but significantly reduced the expression of pro-inflammatory factors associated with senescence. It is likely that TFAM deficiency leads to premature senescence due to other changes in mitochondrial function apart from its role in preventing mtDNA cytosolic leakage (**Extended Data Figure 8i-n**). As we clarify above, collectively our data strongly supports the model that mtDNA contributes to the SASP but not the senescence-arrest *per se*.

6) The Bax pore is now known to be a large and nebulous arc or disk ranging in size. Use of a poorly substantiated compound reported in 2003 (BCB) to inhibit Bax *in vivo* is not sufficient evidence for the authors claims. There is no evidence presented that BCB is inhibiting inflammation via inhibiting Bax pores as opposed to a myriad of off target effects. Does BCB inhibit inflammation in Bax KO mice? Most important, does Bcl-xL overexpression prevent SACP *in vivo*?

Response: We appreciate the points raised by the reviewer, since the initial submission we noted that BCB was more rigorously characterised and renamed as a BAX activation inhibitor (BAI) in a study by Gavathiotis and colleagues²⁴. In new data, using combined BH3-mimetic (ABT-737/S683) that kill in a BAX/BAK dependent manner, we found that BAI could indeed inhibit death only in BAK deficient cells, consistent with a BAX specific inhibitory effect. We also found that treatment with BAI prevents activated BAX and miMOMP in senescent cells (**Extended Data Figure 15c &d**) and increases resistance to senolysis induced by ABT-263 (**Extended Data Figure 15g**). Importantly, BAI does not impact STING dependent inflammation induced by DNA-transfection (arguing against a non-specific anti-inflammatory effect). We include this data in the revised version (**Extended Data Figure 15h-j**). *In vivo* suppression of SASP by BAI could be through off-target effects of course, we acknowledge this possibility in the revised manuscript highlighting that genetic inhibition of minority MOMP through two different approaches: hepatocyte deletion in the original ms (**Figure 4 a-b**) and adenoviral Cre in aged mice in the revised ms. (**Figure 4c-i**) inhibits SASP. We are unaware of inducible BCL-xL overexpressing transgenic mice (to specifically express XL upon senescence or in aged animals *in vivo*), we consider BAX/BAK deletion that we have applied here a more definitive way to prevent minority MOMP than BCL-xL expression.

Minor points:

1) During apoptosis following MOMP cytochrome c is released from mitochondria and is faintly and evenly dispersed in the cytosol. What are the bright foci of cytochrome c in Fig 1A? Why are these foci not yellow as in most of the perinuclear mitochondria? It is hard to see any green staining in the high mag images that does not have some red staining too. Maybe these bright foci of cytochrome c artificially make the other red cytochrome c within the green mitochondria more difficult to visualize. Please show high mag examples of cytochrome c release where bright foci of cytochrome c are not present as the current images are not convincing evidence of miMOMP.

Response: We thank the reviewer for pointing this out, we now include representative images revealing minority MOMP, without bright cytochrome c foci (See **Figure 1a** and **Extended Data Figure 11c**).

2) Please show representative images for the counting of cGAS TFAM colocalization in Fig. 5g. Is Fig. 5g representing Pearson's colocalization index? If not, please add this.

Response: These are now added, please see **Extended Data Figure 9**.

3) Figure 5 m,n could be deleted to save space because this is already well known.

Response: We agree with the reviewer, these are now in **Extended Data Figure 9**. We have kept them in the MS since we believe this to be an important control to demonstrate the involvement of cGAS/STING pathway as a driver of the SASP in our experimental system.

4) It is stated on page 6 that ABT-737 “does not induce any cell-death ...” citing Supplemental Figure 1a. This is not shown in Supplemental Figure 1a or anywhere in Supplemental Figure 1 – and needs to be added.

Response: We apologise for the oversight, this is now added, **Extended Data Figure 1**.

Referee #2 (Remarks to the Author):

In this interesting paper, Chapman et al. examine the role of mitochondria in the phenomenon of senescence-associated secretory phenotype (SASP). They find that limited mitochondrial outer-membrane permeabilization (MOMP), dependent on Bax and Bak, occurs in senescent cells, and that the resultant release of mtDNA and cGAS/STING activation is required for the SASP. In vivo, they further show that Bax and Bak in the liver are required for the induction of SASP in a model of liver senescence. These experiments and results, and their interpretation are overall very solid and convincing. They go on to demonstrate that a putative Bax inhibitor prevents SASP in vitro and has striking neuroprotective effects in age-associated neurological decline. While these data are compelling, I have some issues with these experiments, as described below.

Response: We appreciate the reviewer's positive and constructive critique. Our responses to individual comments are detailed below.

1. The authors use a piperazine derivative of 2-propanol as a “Bax inhibitor,” based on a paper from 2003. Whether this compound is indeed specific for Bax or for MOMP is by no means known, and indeed, there is evidence that propanol derivatives can be neuroprotective in other settings (although mechanism and specificity are an issue here as well). Since the effects they

show are interesting, I suggest that some effort be made to determine if these effects are likely to be on target. First, we need to know if this is indeed Bax specific, which can be readily done using cells that are deficient in Bax or Bak, induced to undergo apoptosis (it should be noted that should the compound block death in both settings, this is fine, but does suggest that it should not be referred to as a “Bax inhibitor.”

Response: We agree with the referee, following initial submission, we noted BCB has been more rigorously characterised as a BAX inhibitor (renamed BAX activation inhibitor – BAI) by Gavathiotis and colleagues ²⁴. They showed that BAI inhibits conformational events in BAX activation that prevent BAX mitochondrial translocation and oligomerization.

In the revised ms. we tested if this inhibitor could prevent cell death using combined BH3-mimetic treatment (S83/ABT-737) in U2OS cells singly deficient for BAX, BAK or BAX/BAK deleted (please see **Extended Data Figure 15 a-b**). As expected, only combined BAX/BAK deletion prevented BH3-mimetic induced cell death. Importantly, BAI addition inhibited cell-death only in BAK^{-/-} cells, consistent with a Bax-specific inhibitory effect.

2. Secondly, it would be important to know if the inhibitor does indeed prevent miMOMP in senescent cells. Note that since the inhibitor blocks SASP in cells that are already senescent (apparently—as suggested in the legend of Figure 6b) this would suggest that continued release of mtDNA is necessary for the SASP. If the compound is added prior to induction of senescence (in 6a and 6b) this should be noted. In any case, we need to know if the inhibitor blocks miMOMP, as suggested by the experiment in 6a.

Response: The reviewer raises an excellent point, in new data (**Extended Data Figure 15 c and d**), we show that the inhibitor does indeed prevent BAX activation. As the reviewer highlights, our results imply that continued release of mtDNA is required to maintain the SASP. Additionally, we also found that senescent cells pre-treated with BAI1 were significantly less sensitive to treatment with BH3 mimetic ABT263.

3. Finally, it is critical to determine if the inhibitor blocks the production/secretion of SASP cytokines induced by means other than senescence (e.g., TLR engagement). If so, this would call into question the interpretation of their results.

Response: To test this (in effect a possible off-target effect), in new expts. we transfected herring testes (HT) DNA into cells (to activate cGAS-STING) and measured inflammatory cytokine production in the presence/absence of BAI. We found that inflammation induced by transfection with herring testes (HT) DNA in human fibroblasts occurred independently of BAX and BAK and was not affected by BAI1 (**Extended Data Figure 15h-j**), demonstrating that the inhibitor does not non-specifically inhibit cGAS-STING signaling *in vitro*.

4. The authors should discuss the findings on propranol derivatives as neuroprotective agents in settings that might be unlikely to relate to senescence. At least, the possibility that their inhibitor is acting in some other way than via the pathway they have found should be presented as a possibility.

Response: We agree that *in vivo* application of the inhibitor may have effects independent of senescence and we state the possible caveat in the revised version. We consider the genetic inhibition of minority MOMP through two different approaches to delete BAX (hepatocyte restricted in the original ms, adenoviral systemic delivery of Cre in the revised) in BAK null/BAX FL/FL mice, both inhibiting the inflammatory SASP a more definitive approach to determine the impact of minority MOMP on SASP *in vivo*.

5. On a different note, the experiments on the induction of senescence using low dose BH3 mimetics is interesting, as is the associated gamma H2AX staining. It is possible that this phenotype is dependent on caspase-activated nuclease, which they can easily test in their

APAF1-deficient cells (which undergo senescence and SASP). I do not think that testing this idea is essential for the conclusions of the paper but would be interesting to specialists. I suggest that this experiment (induction of g-H2AX by BH3 mimetics in the presence or absence of APAF1) is optional.

Response: We thank the reviewer for highlighting this, completely agree with their likely interpretation of this data – namely minority MOMP/caspase dependent DNA-damage could promote senescence. While we have not formally tested this idea, we note an earlier publication (now cited) supportive of this hypothesis where BH3-mimetic treatment led to DNA-damage and senescence in a caspase-dependent manner ²¹.

Referee #3 (Remarks to the Author):

In this work, Chapman et al. report that senescent cells display minority MOMP in a subset of mitochondria generating Bax/Bak pores, mtDNA release, and inflammatory cytokine production through cGAS-STING activation. The authors show the importance of Bax/Bak for inflammatory cytokine production in a model of irradiation-induced stress *in vivo* and report improvements in physical function and bone homeostasis in aged mice. These findings provide novel information on how senescent cells regulate their inflammatory phenotypes and will be of interest to readers from the field of aging, senescence, and innate immunity. I am generally positive about the manuscript but would like to see additional data to support the generalization of the findings in other settings of senescence and thorough documentation of reduced inflammation in aged mice receiving Bak inhibitor treatments.

Response: We very much appreciate the reviewer's positive appraisal and thank them for the constructive review. Our responses to specific comments follow:

Major: 1) Mitochondrial dysfunction in senescence and aging is well-documented, but a precise molecular understanding of how this connects to inflammation is missing. The primary conclusion from this work is that minority MOMP is responsible for mtDNA release and senescence-induced cytokine secretion via the cGAS-STING pathway.

There are many ways to trigger senescence *in vitro*. It would add to the manuscript to demonstrate Bax/Bak-dependent in chemotherapy-induced senescence - a medically relevant context of senescence.

Response: To address this important point in the revised ms. we now investigate the SASP in human fibroblasts (with or without BAX/BAK deletion to prevent MOMP) following treatment with two different chemotherapies – doxorubicin and etoposide. Both therapies induced senescence, independent of the presence of BAX/BAK however the SASP was largely blunted by BAX/BAK deletion, consistent with our dataset that apoptotic signalling leading to minority MOMP contributes to the SASP (see new data -**Extended Data Figure 4**). This may have important implications for the negative effects therapy induced-SASP have been reported to have in cancer patients.

2) The authors show multiple independent data to document health improvements in aged mice treated with a BAK inhibitor. According to their model, this outcome should strictly depend on decreased markers of inflammation, an aspect that appears underdeveloped in the current manuscript. Ideally, I would like to see an assessment of inflammatory cytokines in another organ than bone.

Response: Extending our analysis we find that applying the BAX inhibitor in aged animals also reduces inflammatory markers in the brain of aged mice, this data is now included (**please see Extended Data Figure 16**). In an additional new model, we used intravenous injection of adenoviral Cre into aged BAK null/BAX fl/fl mice, with the aim of deleting BAX in multiple

organs in aged animals. Phenocopying the BAX inhibitor work, we also found an inhibition of inflammation in aged bone (**Extended Data Figure 6d-f**). Alongside this, inflammatory markers in livers were also reduced following Bax and Bak deletion (**Figure 4**). Thus, through different approaches, inhibition in inflammatory markers in aged animals by preventing MOMP can be detected in multiple organs.

Minor:

1) Prior work has reported that TFAM deficiency induces senescence, triggers inflammaging, and accelerates features of aging-associated decline in mice (PMID: 32439659). This publication should be cited in this context.

Response: Thank you for highlighting this publication which we now cite.

2) References 5 and 24 are identical.

Response: Apologies, this has been corrected.

3) The authors should measure BAX and BAK protein levels in senescent cells versus naïve cells.

Response: This has now been included, see new **Extended Data Figure 2**.

4) Related to Extended Data Fig. 1: One might have expected that the treatment of cells with a miMOMP inducer would trigger an immediate immune response by releasing mtDNA. Are type I IFN response genes also upregulated in a delayed fashion?

Response: We analyzed interferon genes alpha and beta and show that they increase in a delayed fashion (see revised **Extended Data Figure 1 f and g**).

References:

- 1 Correia-Melo, C. *et al.* Mitochondria are required for pro-ageing features of the senescent phenotype. *Embo j* **35**, 724-742, doi:10.15252/embj.201592862 (2016).
- 2 Passos, J. F. *et al.* Feedback between p21 and reactive oxygen production is necessary for cell senescence. *Mol Syst Biol* **6**, 347 (2010).
- 3 Passos, J. F. *et al.* Mitochondrial Dysfunction Accounts for the Stochastic Heterogeneity In Telomere-Dependent Senescence. *PLoS Biology* **5**, e110 (2007).
- 4 Cao, K. *et al.* Mitochondrial dynamics regulate genome stability via control of caspase-dependent DNA damage. *Dev Cell* **57**, 1211-1225.e1216, doi:10.1016/j.devcel.2022.03.019 (2022).
- 5 Chang, J. *et al.* Clearance of senescent cells by ABT263 rejuvenates aged hematopoietic stem cells in mice. *Nat Med* **22**, 78-83, doi:10.1038/nm.4010 (2016).
- 6 Zhu, Y. *et al.* Identification of a novel senolytic agent, navitoclax, targeting the Bcl-2 family of anti-apoptotic factors. *Aging Cell* **15**, 428-435, doi:10.1111/accel.12445 (2016).
- 7 Vizioli, M. G. *et al.* Mitochondria-to-nucleus retrograde signaling drives formation of cytoplasmic chromatin and inflammation in senescence. *Genes & development* **34**, 428-445, doi:10.1101/gad.331272.119 (2020).
- 8 Matsumura, T., Zerrudo, Z. & Hayflick, L. Senescent human diploid cells in culture: survival, DNA synthesis and morphology. *J Gerontol* **34**, 328-334, doi:10.1093/geronj/34.3.328 (1979).
- 9 Pignolo, R. J., Rotenberg, M. O. & Cristofalo, V. J. Alterations in contact and density-dependent arrest state in senescent WI-38 cells. *In Vitro Cell Dev Biol Anim* **30a**, 471-476, doi:10.1007/bf02631316 (1994).

- 10 Hoppins, S. *et al.* The soluble form of Bax regulates mitochondrial fusion via MFN2
homotypic complexes. *Mol Cell* **41**, 150-160, doi:10.1016/j.molcel.2010.11.030 (2011).
- 11 Lagnado, A. *et al.* Neutrophils induce paracrine telomere dysfunction and senescence
in ROS-dependent manner. *Embo j* **40**, e106048, doi:10.15252/emj.2020106048
(2021).
- 12 Ogrodnik, M. *et al.* Cellular senescence drives age-dependent hepatic steatosis. *Nat
Commun* **8**, 15691, doi:10.1038/ncomms15691 (2017).
- 13 Hewitt, G. *et al.* Telomeres are favoured targets of a persistent DNA damage response
in ageing and stress-induced senescence. *Nat Commun* **3**, 708,
doi:10.1038/ncomms1708 (2012).
- 14 Lindsten, T. *et al.* The combined functions of proapoptotic Bcl-2 family members bak
and bax are essential for normal development of multiple tissues. *Mol Cell* **6**, 1389-
1399, doi:10.1016/s1097-2765(00)00136-2 (2000).
- 15 King, M. P. & Attardi, G. Human cells lacking mtDNA: repopulation with exogenous
mitochondria by complementation. *Science* **246**, 500-503,
doi:10.1126/science.2814477 (1989).
- 16 Tait, S. W. *et al.* Widespread mitochondrial depletion via mitophagy does not
compromise necroptosis. *Cell Rep* **5**, 878-885, doi:10.1016/j.celrep.2013.10.034
(2013).
- 17 Correia-Melo, C., Ichim, G., Tait, Stephen W. G. & Passos, J. F. Depletion of
mitochondria in mammalian cells through induction of widespread mitophagy. *Nature
Protocols* **12**, 183-194 (2017).
- 18 Giampazolias, E. *et al.* Mitochondrial permeabilization engages NF- κ B-dependent
anti-tumour activity under caspase deficiency. *Nat Cell Biol* **19**, 1116-1129,
doi:10.1038/ncb3596 (2017).
- 19 Dou, Z. *et al.* Cytoplasmic chromatin triggers inflammation in senescence and cancer.
Nature **550**, 402-406, doi:10.1038/nature24050 (2017).
- 20 Wiley, Christopher D. *et al.* Mitochondrial Dysfunction Induces Senescence with a
Distinct Secretory Phenotype. *Cell Metabolism* **23**, 303-314 (2016).
- 21 Song, J. H., Kandasamy, K., Zemskova, M., Lin, Y. W. & Kraft, A. S. The BH3 mimetic
ABT-737 induces cancer cell senescence. *Cancer Res* **71**, 506-515,
doi:10.1158/0008-5472.Can-10-1977 (2011).
- 22 Ichim, G. *et al.* Limited mitochondrial permeabilization causes DNA damage and
genomic instability in the absence of cell death. *Mol Cell* **57**, 860-872,
doi:10.1016/j.molcel.2015.01.018 (2015).
- 23 Hong, Z. *et al.* STING inhibitors target the cyclic dinucleotide binding pocket.
Proceedings of the National Academy of Sciences of the United States of America
118, doi:10.1073/pnas.2105465118 (2021).
- 24 Garner, T. P. *et al.* Small-molecule allosteric inhibitors of BAX. *Nat Chem Biol* **15**, 322-
330, doi:10.1038/s41589-018-0223-0 (2019).

Reviewer Reports on the First Revision:

Referees' comments:

Referee #1 (Remarks to the Author):

The authors have revised their manuscript and changed the text in some parts. Overall, the most important conclusions are not well substantiated by the data. Those issues are discussed below in context of my prior review comments and the author's rebuttal.

A) My prior comment #1) A major issue with the prior version of the manuscript, as I wrote before, was: "How senescence mediates miMOMP is a key missing issue."

The authors respond by including new data in the manuscript on mitochondrial dynamics and conclude in their rebuttal letter, "We found that enforced mitochondrial fission (MFN-2 CRISPR/CAS9, shRNA-mediated knockout or CCCP treatment), promotes minority MOMP leading to leakage of mtDNA and the SASP."

Although these new data are of moderate interest, they do not address how senescence mediates miMOMP, it only moves the issue of Bax function one step upstream, a step already reported in the literature. How senescence alters mitochondrial dynamics is now the key missing issue and needs to be resolved.

B) The authors do not resolve what restrains miMOMP during senescence in my prior point #2. They speculate in the rebuttal letter as follows, "what constrains minority MOMP is likely a combination of factors: 1) fragmentation/fission of mitochondria that are primed to undergo minority MOMP (consistent with our new data), thus limiting MOMP to individual mitochondria 2) intrinsic mitochondrial dysfunction signals minority MOMP on individual mitochondria, consistent with findings from us (and others) of mitochondrial dysfunction during senescence."

How could mitochondrial fragmentation limit miMOMP if the same process activates miMOMP? Perhaps the authors think that a brake on mitochondrial fragmentation during senescence limits the spread of miMOMP. How senescence sets that brake needs to be ascertained.

Prior point #6. The new citation added in the rebuttal letter for the compound used to inhibit Bax reflects a paper that shows cell free and tissue culture experiments. Many chemical inhibitors that work in tissue culture do not work in vivo owing to pharmacodynamic limitations, drug binding to blood proteins or off target effects on various tissues not able to be assessed in tissue culture. The in vivo data on BAI are not interpretable, may be misleading and cannot be justified by the authors simply adding a proviso about this possibility (From their rebuttal letter: "In vivo suppression of SASP by BAI could be through off-target effects of course, we acknowledge"). These data need to be deleted. Inducible Bcl-xL would be the correct way to address the in vivo aspects – and such mice can easily be made.

C) In response to my prior comment # 4, the authors correctly say in their rebuttal letter: "In the setting of Rho zero cells, the reviewer rightly notes continued IL-8 expression independent of mtDNA in senescent cells. Potentially, two main reasons underlie this, minority MOMP can activate inflammatory signalling independent of mtDNA for instance via NF- κ B signaling and/or there are additional mechanisms driving the SASP".

These data and the authors' conclusion above oppose the main point of the manuscript as stated by the authors in the abstract; "minority MOMP (miMOMP), depends on the formation of BAX and BAK macropores leading to the release of mitochondrial DNA (mtDNA) into the cytosol, which in turn activates the cGAS-STING pathway, a major regulator of the SASP.

Referee #2 (Remarks to the Author):

The authors have nicely addressed my concerns. I find that the conclusions of the paper are well supported by their data and provide novel insights into the nature of the SASP and its causes.

Referee #3 (Remarks to the Author):

The authors have constructively addressed the points that were raised. To provide evidence on the broad effect of BAI1 (BAX/BAK inhibition) in tissues other than the bone, inflammatory genes were measured in the brain of aged mice treated or not with BAI1 (Extended Data Fig. 16e, f). The effects of this intervention are moderate when compared to bone, begging the question of whether BAI1 is efficiently targeting BAX/BAK in the brain. At a minimum the aspect on brain penetrance should be added to the discussion section before publication or validated experimentally. Whether or not the inflammatory signals originate from within the brain is important.

In Extended Data Fig. 6e, for the figure panel related to the expression of Nfkb1 the y-axis is cut off. The y-axis should be displayed as in the other panels of Extended Data Fig. 6e.

The text related to Fig. 4g and h is misleading, as there are no significant changes in the abundance of CD45 positive immune cells.

Author Rebuttals to First Revision:

Point-by-point response to reviewers' comments:

Reviewer 1

“A) My prior comment #1) A major issue with the prior version of the manuscript, as I wrote before, was: “How senescence mediates miMOMP is a key missing issue.”

The authors respond by including new data in the manuscript on mitochondrial dynamics and conclude in their rebuttal letter, “We found that enforced mitochondrial fission (MFN-2 CRISPR/CAS9, shRNA-mediated knockout or CCCP treatment), promotes minority MOMP leading to leakage of mtDNA and the SASP.”

Although these new data are of moderate interest, they do not address how senescence mediates miMOMP, it only moves the issue of Bax function one step upstream, a step already reported in the literature. How senescence alters mitochondrial dynamics is now the key missing issue and needs to be resolved. “

Response: The revised version provides extensive new mechanistic insight as to why minority MOMP/sub-lethal apoptotic stress occurs in senescence cells – our apologies if this was not explicit.

These mechanistic aspects (re-iterated below) are now highlighted in the revised text (please see lines 457-460).

1) We found that the occurrence of minority MOMP correlates with apoptotic priming, cells with high apoptotic priming display minority MOMP, this is consistent with our earlier published data (Extended Figure 1 and 16g)¹. Senescent cells have been shown by us here (Extended Figure 15g) and many other labs²⁻⁶ as having high apoptotic priming, thus are inherently prone to engage minority MOMP.

2) We discovered that mitochondrial dynamics play a central role in enabling minority MOMP in senescent cells whereby fragmentation of mitochondria enables minority MOMP promoting mtDNA-dependent SASP, whereas fusion suppresses this effect.

We and others have shown that mitochondrial dynamics are altered in senescent cells⁷⁻⁹, this is likely due to many factors including expression of proteins involved in dynamics and/or disrupted mitochondrial function in senescent cells^{8,10-12}. The question as to why mitochondrial dynamics are altered in senescent cells, while undoubtedly interesting, is multifactorial, and not key to the central message of our paper linking sub-lethal apoptotic stress to the SASP.

“B) The authors do not resolve what restrains miMOMP during seescence in my prior point #2. They speculate in the rebuttal letter as follows, “what constrains minority MOMP is likely a combination of factors: 1) fragmentation/fission of mitochondria that are primed to undergo minority MOMP (consistent with our new data), thus limiting MOMP to individual mitochondria 2) intrinsic mitochondrial dysfunction signals minority MOMP on individual mitochondria, consistent with findings from us (and others) of mitochondrial dysfunction during senescence.”

How could mitochondrial fragmentation limit miMOMP if the same process activates miMOMP?”

Response: Our apologies, we didn't intend to convey the wrong message that fragmentation both promotes and inhibits minority MOMP. To clarify, fragmentation enables select mitochondria to permeabilize (i.e. minority MOMP), these permeabilized mitochondria can no longer fuse with intact mitochondria, thus fragmentation prevents minority MOMP from further propagation. We have revised the manuscript to clarify this point (lines 458-460).

“Perhaps the authors think that a brake on mitochondrial fragmentation during senescence limits the spread of miMOMP. How senescence sets that brake needs to be ascertained.”

Response: We don't think senescent cells need set a brake on mitochondrial fragmentation to limit the spread of miMOMP; once mitochondria have permeabilised they lose their ability to fuse with other mitochondria¹³, preventing minority MOMP from spreading to other mitochondria.

“Prior point #6. The new citation added in the rebuttal letter for the compound used to inhibit Bax reflects a paper that shows cell free and tissue culture experiments. Many chemical inhibitors that work in tissue culture do not work *in vivo* owing to pharmacodynamic limitations, drug binding to blood proteins or off-target effects on various tissues not able to be assessed in tissue culture. The *in vivo* data on BAI are not interpretable, may be misleading and cannot be justified by the authors simply adding a proviso about this possibility (From their rebuttal letter: “*In vivo* suppression of SASP by BAI could be through off-target effects of course, we acknowledge”). These data need to be deleted.”

Response: In the revised version we demonstrated that BAI effectively and selectively inhibits BAX *in vitro* (Extended Data Figure 15). BAI does work *in vivo*, phenocopying deletion of its target BAX, both causing suppression of the SASP – this strongly suggests that BAI is working *in vivo* through inhibition of BAX. We now cite (line 361) a recent paper in *Nature Cancer* has also demonstrated BAI inhibits BAX-dependent apoptosis *in vivo*¹⁴ demonstrating on-target inhibition. Our comment “*could be through off-target effects of course*” was in response to reviewer 2's (in our view correct) statement that 100% absolute determination of on-target effects by BAI (and indeed any drug) *in vivo* is very difficult – for clarity we have now removed this statement.

“Inducible Bcl-xL would be the correct way to address the *in vivo* aspects – and such mice can easily be made. “

Response: The gold-standard way to prevent MOMP is to delete BAX and BAK^{15,16}, hence we used this approach in our *in vivo* expts. in irradiation and ageing models of senescence (Figure 4). We are unclear why BAX/BAK deletion is considered inappropriate, with the reviewer suggesting, instead to generate/use a transgenic *in vivo* model of inducible BCL-xL expression. We note that BCL-XL can have additional non-apoptotic effects¹⁷⁻¹⁹, alongside no guarantee that sufficient BCLxL expression is achieved *in vivo* to completely block MOMP (in contrast to BAX/BAK deletion). Indeed, several reports show that senescent cells display increased expression of BCL-XL^{20,21}.

“C) In response to my prior comment # 4, the authors correctly say in their rebuttal letter: “In the setting of Rho zero cells, the reviewer rightly notes continued IL-8 expression independent of mtDNA in senescent cells. Potentially, two main reasons underlie this, minority MOMP can activate inflammatory signalling independent of mtDNA for instance via NF-kB signaling and/or there are additional mechanisms driving the SASP”.

These data and the authors' conclusion above oppose the main point of the manuscript as stated by the authors in the abstract; “minority MOMP (miMOMP), depends on the formation of BAX and

BAK macropores leading to the release of mitochondrial DNA (mtDNA) into the cytosol, which in turn activates the cGAS-STING pathway, a major regulator of the SASP.”

Response: Given the concerns from the reviewer, we applied an alternative approach (using mitochondrial targeting of a viral DNase), to generate mtDNA depleted cells. Consistent with our previous data, mtDNA reduction profoundly inhibited the SASP in senescent cells, implicating mtDNA in the SASP (Extended Figure 8a – e).

The quoted statement is not opposed to a main conclusion of our study - “minority MOMP (miMOMP), ... activates the cGAS-STING pathway, a major regulator of the SASP”. Major studies have shown various ways to activate the SASP, including alternate sources of DNA leading to cGAS-STING dependent SASP, as well as cGAS-STING independent contributions of NF-kB signaling to the SASP²²⁻²⁶. Therefore, it would be wrong to suggest the only way to engage the SASP is via minority MOMP.

Reviewer 2:

The authors have nicely addressed my concerns. I find that the conclusions of the paper are well supported by their data and provide novel insights into the nature of the SASP and it's causes.

Response: We thank the reviewer for their positive comments.

Reviewer 3:

The authors have constructively addressed the points that were raised. To provide evidence on the broad effect of BAI1 (BAX/BAK inhibition) in tissues other than the bone, inflammatory genes were measured in the brain of aged mice treated or not with BAI1 (Extended Data Fig. 16e, f). The effects of this intervention are moderate when compared to bone, begging the question of whether BAI1 is efficiently targeting BAX/BAK in the brain. At a minimum the aspect on brain penetrance should be added to the discussion section before publication or validated experimentally. Whether or not the inflammatory signals originate from within the brain is important.

Response: We thank the reviewer for raising this important point that we have addressed through collaboration with Dr. Evris Gavathiotis (Albert Einstein College of Medicine) finding that BAI1 is effectively targeted to the brain (penetrating the blood brain barrier) (please see Extended data Fig 18a and b).

Using single-cell RNAseq we analyzed, more comprehensively, brain cell populations following BAI1 treatment, finding a significant reduction in senescence and SASP components (defined by the gene set SenMayo²⁷, which is composed of 125 genes) in microglia and oligodendrocytes (Figure 6n) highlighting a contributory role of MOMP to the SASP in the aging brain.

In Extended Data Fig. 6e, for the figure panel related to the expression of Nfkb1 is the y-axis is cut off. The y-axis should be displayed as in the other panels of Extended Data Fig. 6e.

Response: This has been corrected.

The text related to Fig. 4g and h is misleading, as there are no significant changes in the abundance of CD45 positive immune cells.

Response: We apologize for this oversight and have corrected it.

References

- 1 Cao, K. *et al.* Mitochondrial dynamics regulate genome stability via control of caspase-dependent DNA damage. *Dev Cell* **57**, 1211-1225.e1216 (2022). <https://doi.org:10.1016/j.devcel.2022.03.019>
- 2 Shahbandi, A. *et al.* BH3 mimetics selectively eliminate chemotherapy-induced senescent cells and improve response in TP53 wild-type breast cancer. *Cell Death Differ* **27**, 3097-3116 (2020). <https://doi.org:10.1038/s41418-020-0564-6>
- 3 Rysanek, D. *et al.* Synergism of BCL-2 family inhibitors facilitates selective elimination of senescent cells. *Aging (Albany NY)* **14**, 6381-6414 (2022). <https://doi.org:10.18632/aging.204207>
- 4 Kohli, J. *et al.* Targeting anti-apoptotic pathways eliminates senescent melanocytes and leads to nevi regression. *Nat Commun* **13**, 7923 (2022). <https://doi.org:10.1038/s41467-022-35657-9>
- 5 Chang, J. *et al.* Clearance of senescent cells by ABT263 rejuvenates aged hematopoietic stem cells in mice. *Nat Med* **22**, 78-83 (2016). <https://doi.org:10.1038/nm.4010>
- 6 Yosef, R. *et al.* Directed elimination of senescent cells by inhibition of BCL-W and BCL-XL. *Nat Commun* **7**, 11190 (2016). <https://doi.org:10.1038/ncomms11190>
- 7 Dalle Pezze, P. *et al.* Dynamic modelling of pathways to cellular senescence reveals strategies for targeted interventions. *PLoS Comput Biol* **10**, e1003728 (2014). <https://doi.org:10.1371/journal.pcbi.1003728>
- 8 Yu, B. *et al.* Mitochondrial phosphatase PGAM5 modulates cellular senescence by regulating mitochondrial dynamics. *Nat Commun* **11**, 2549 (2020). <https://doi.org:10.1038/s41467-020-16312-7>
- 9 Jendrach, M. *et al.* Morpho-dynamic changes of mitochondria during ageing of human endothelial cells. *Mech Ageing Dev* **126**, 813-821 (2005). <https://doi.org:10.1016/j.mad.2005.03.002>
- 10 Correia-Melo, C. *et al.* Mitochondria are required for pro-ageing features of the senescent phenotype. *EMBO J* **35**, 724-742 (2016). <https://doi.org:10.15252/emboj.201592862>
- 11 Lee, S. *et al.* Mitochondrial Fission and Fusion Mediators, hFis1 and OPA1, Modulate Cellular Senescence*. *Journal of Biological Chemistry* **282**, 22977-22983 (2007). <https://doi.org:https://doi.org/10.1074/jbc.M700679200>
- 12 Park, Y.-Y. *et al.* Loss of MARCH5 mitochondrial E3 ubiquitin ligase induces cellular senescence through dynamin-related protein 1 and mitofusin 1. *Journal of Cell Science* **123**, 619-626 (2010). <https://doi.org:10.1242/jcs.061481>
- 13 Hoppins, S. *et al.* The soluble form of Bax regulates mitochondrial fusion via MFN2 homotypic complexes. *Mol Cell* **41**, 150-160 (2011). <https://doi.org:10.1016/j.molcel.2010.11.030>
- 14 Amgalan, D. *et al.* A small-molecule allosteric inhibitor of BAX protects against doxorubicin-induced cardiomyopathy. *Nat Cancer* **1**, 315-328 (2020). <https://doi.org:10.1038/s43018-020-0039-1>
- 15 Wei, M. C. *et al.* Proapoptotic BAX and BAK: a requisite gateway to mitochondrial dysfunction and death. *Science* **292**, 727-730 (2001). <https://doi.org:10.1126/science.1059108>
- 16 Lindsten, T. *et al.* The combined functions of proapoptotic Bcl-2 family members bak and bax are essential for normal development of multiple tissues. *Mol Cell* **6**, 1389-1399 (2000). [https://doi.org:10.1016/s1097-2765\(00\)00136-2](https://doi.org:10.1016/s1097-2765(00)00136-2)

- 17 Alavian, K. N. *et al.* Bcl-xL regulates metabolic efficiency of neurons through interaction with the mitochondrial F1FO ATP synthase. *Nat Cell Biol* **13**, 1224-1233 (2011). <https://doi.org:10.1038/ncb2330>
- 18 Maiuri, M. C. *et al.* Functional and physical interaction between Bcl-X(L) and a BH3-like domain in Beclin-1. *Embo j* **26**, 2527-2539 (2007). <https://doi.org:10.1038/sj.emboj.7601689>
- 19 Wiese, C., Pierce, A. J., Gauny, S. S., Jasin, M. & Kronenberg, A. Gene conversion is strongly induced in human cells by double-strand breaks and is modulated by the expression of BCL-x(L). *Cancer Res* **62**, 1279-1283 (2002).
- 20 Yosef, R. *et al.* Directed elimination of senescent cells by inhibition of BCL-W and BCL-XL. *Nat Commun* **7**, 11190 (2016). <https://doi.org:10.1038/ncomms11190>
- 21 Zhu, Y. *et al.* Identification of a novel senolytic agent, navitoclax, targeting the Bcl-2 family of anti-apoptotic factors. *Aging Cell* **15**, 428-435 (2016). <https://doi.org:10.1111/ace1.12445>
- 22 Dou, Z. *et al.* Cytoplasmic chromatin triggers inflammation in senescence and cancer. *Nature* **550**, 402-406 (2017). <https://doi.org:10.1038/nature24050>
- 23 Rodier, F. *et al.* Persistent DNA damage signalling triggers senescence-associated inflammatory cytokine secretion. *Nat Cell Biol* **11**, 973-979 (2009). <https://doi.org:10.1038/ncb1909>
- 24 Freund, A., Patil, C. K. & Campisi, J. p38MAPK is a novel DNA damage response-independent regulator of the senescence-associated secretory phenotype. *EMBO J* **30**, 1536-1548 (2011). <https://doi.org:10.1038/emboj.2011.69>
- 25 Glück, S. *et al.* Innate immune sensing of cytosolic chromatin fragments through cGAS promotes senescence. *Nat Cell Biol* **19**, 1061-1070 (2017). <https://doi.org:10.1038/ncb3586>
- 26 Kang, C. *et al.* The DNA damage response induces inflammation and senescence by inhibiting autophagy of GATA4. *Science* **349**, aaa5612 (2015). <https://doi.org:10.1126/science.aaa5612>
- 27 Saul, D. *et al.* A new gene set identifies senescent cells and predicts senescence-associated pathways across tissues. *Nat Commun* **13**, 4827 (2022). <https://doi.org:10.1038/s41467-022-32552-1>

Reviewer Reports on the Second Revision:

Referees' comments:

Referee #1 (Remarks to the Author):

The authors adequately addressed all my remaining concerns and I now support publication.

Referee #3 (Remarks to the Author):

The authors have adequately addressed the brain penetrance of BAI1. The new snRNA-seq data further support a role of BAI1 in impacting brain senescence. It appears as if in Fig. 6m (right panel) the scale is missing. The different levels of p16 positive cells can likely be explained by paracrine senescence. It would have been more informative to blot any inflammatory cytokine, given that this is the strongest effect of BAX/BAK inhibition. A short explanation in the text/discussion could potentially be useful.

Author Rebuttals to Second Revision:

Response to referees:

Referee #1 (Remarks to the Author):

The authors adequately addressed all my remaining concerns and I now support publication.

We thank the referee for their positive comments.

Referee #3 (Remarks to the Author):

The authors have adequately addressed the brain penetrance of BAI1. The new snRNA-seq data further support a role of BAI1 in impacting brain senescence. It appears as if in Fig. 6m (right panel) the scale is missing. The different levels of p16 positive cells can likely be explained by paracrine senescence. It would have been more informative to blot any inflammatory cytokine, given that this is the strongest effect of BAX/BAK inhibition. A short explanation in the text/discussion could potentially be useful.

The scale in figure 6 is now clearly presented. We agree with the referee's interpretation of our data, and discuss it, accordingly, see line 370.